# CONTEXT IS ENVIRONMENT

**Sharut Gupta**
Meta AI, MIT CSAIL
sharut@mit.edu

**Stefanie Jegelka**
MIT CSAIL
stefje@mit.edu

**David Lopez-Paz, Kartik Ahuja**
Meta AI
{dlp,kartikahuja}@meta.com

## ABSTRACT

Two lines of work are taking center stage in AI research. On the one hand, the community is making increasing efforts to build models that discard spurious correlations and generalize better in novel test environments. Unfortunately, a hard lesson so far is that no proposal convincingly outperforms a simple empirical risk minimization baseline. On the other hand, large language models (LLMs) have erupted as algorithms able to learn *in-context*, generalizing on-the-fly to the eclectic contextual circumstances that users enforce by prompting. We argue that *context is environment*, and posit that in-context learning holds the key to better domain generalization. Via extensive theory and experiments, we show that paying attention to context—unlabeled examples as they arrive—allows our proposed In-Context Risk Minimization (ICRM) algorithm to *zoom-in* on the test environment risk minimizer, leading to significant out-of-distribution performance improvements. Furthermore, training with context helps the model learn a better featurizer. From all of this, two messages are worth taking home: researchers in domain generalization should consider *environment as context*, and harness the adaptive power of in-context learning. Researchers in LLMs should consider *context as environment*, to better structure data towards generalization. Code is available at https://github.com/facebookresearch/ICRM.

## 1 INTRODUCTION

One key problem in AI research is to build systems that generalize across a wide range of test environments. In principle, these algorithms should discard spurious correlations present only in certain training environments, and capture invariant patterns appearing across conditions. For example, we would like to build self-driving systems that, while trained on certain weather conditions, levels of traffic, and driving rules, can perform satisfactorily in new circumstances. Unfortunately, this has proven challenging—for instance, these models often fail to drive in unseen weather conditions (Lechner et al., 2022), creating immediate hazards. Despite its importance, how to perform well beyond the distribution of the training data remains a burning question. In fact, major international conferences offer well-attended workshops dedicated to the issue (Wald et al., 2023), and news articles remind us of the profound societal impact of the failures of ML systems (Angwin et al., 2016).

Research efforts have so far led to domain generalization algorithms that fall into two broad categories. On the one hand, invariance proposals (Ganin et al., 2016; Peters et al., 2016; Arjovsky et al., 2019), illustrated in Figure 1a, discard all environment-specific information, thus removing excessive signal about the problem. On the other hand, marginal transfer proposals (Blanchard et al., 2011; Li et al., 2016a; Zhang et al., 2020; Bao and Karaletsos, 2023), illustrated in Figure 1b, summarize observed inputs in each environment as a coarse embedding, thus diluting important signal at the example level. So far, the bitter lesson is that no algorithm geared towards out-of-distribution generalization convincingly outperforms a simple empirical risk minimization (ERM) baseline, which pools the data from environments, when evaluated across standard real-world benchmarks (Gulrajani and Lopez-Paz, 2020; Gagnon-Audet et al., 2023; Yao et al., 2022). Has the generalization project hit a dead end?

In parallel, large language models (OpenAI, 2023; Touvron et al., 2023, LLMs) are taking the world by storm. LLMs are next-token predictors built with transformers (Vaswani et al., 2017) and trained on enormous amounts of natural language. One impressive capability of LLM systems is their ability to learn *in-context*, that is, to generalize on-the-fly to the eclectic circumstances that users enforce by prompting (Brown et al., 2020). For example, a trained LLM would complete the sequence

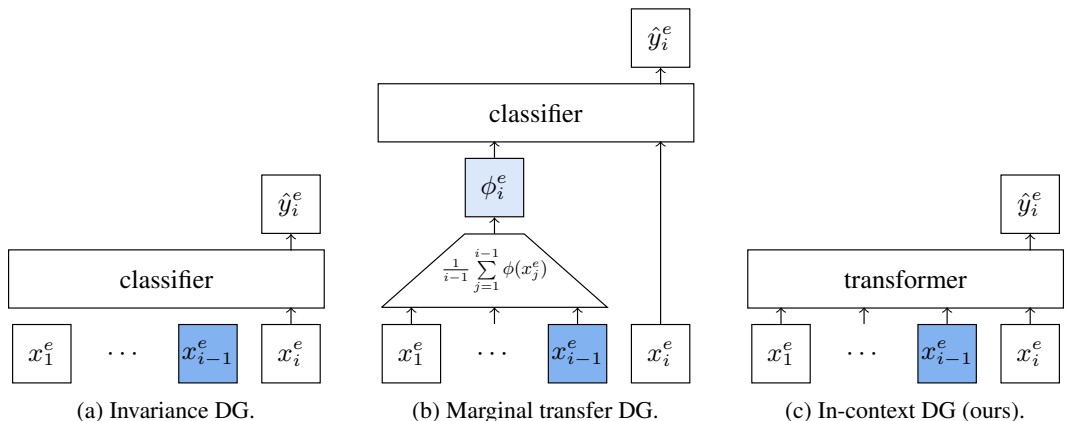

Figure 1: Three frameworks for domain generalization (DG), predicting the target $y_i^e$ from the input $x_i^e$ in test environment $e$. Depicted in blue, the last example $x_{i-1}^e$ contains relevant features for the current prediction. (a) Invariance DG discards all of the previously observed information from the test environment, removing too much predictive signal. (b) Marginal transfer DG summarizes all of the previously observed test inputs as a coarse embedding, diluting predictive signal found at the example level. (b) Our in-context DG directly observes all of the previous test inputs, allowing the search of "needle-in-the-haystack" signals, such as the relevant one, i.e., $x_{i-1}^e$.

"France-Paris Italy-Rome Spain-" with the sequence "Madrid," effectively learning, from the input itself, that the user is demanding a capital prediction task. When interacting with LLMs, one feels closer towards solving the puzzle of out-of-distribution (OOD) generalization. Could LLMs hold a key piece to the OOD puzzle?

This paper suggests a positive answer, establishing a strong parallel between the concept of *environment* in domain generalization, and the concept of *context* in next-token prediction. In fact, different environments describe varying contextual circumstances such as time, location, experimental intervention, and other background conditions. On the one hand, describing *environments as context* opens the door to using powerful next-token predictors off-the-shelf, with their adaptability to learn in-context, to address domain generalization problems. This allows us to move from coarse domain indices to fine and compositional contextual descriptions, helpful to amortize learning across similar environments. On the other hand, using *context as environment* can help LLM researchers to use various domain generalization methods such as distributionally robust optimization (Sagawa et al., 2019; Xie et al., 2023, DRO) across varying contexts.

Based on these insights, we propose a natural algorithm, *In-Context Risk Minimization (ICRM)*, illustrated in Figure 1c. Given examples $(x_i^e, y_i^e)$ from environment $e$, we propose to address *out*-of-distribution prediction as *in*-distribution context-based prediction, training a machine:

$$y_i^e \approx h(x_i^e; \underbrace{x_1^e, \ldots, x_{i-1}^e}_{\text{environment} \approx \text{context}}). \tag{1}$$

While the requested prediction $y_i^e$ concerns only the input $x_i^e$, the machine can now pay attention to the test experience so far, extracting relevant environment information from instance and distributional features. Our theoretical results show that such in-context learners can utilize context to *zoom-in* on the empirical risk minimizer of the test environment, achieving competitive out-of-distribution performance. Further, we show that the extended input-context feature space in ICRM can reveal invariances that ERM-based algorithms ignore. A standout feature of ICRM is its capability to improve feature learning through context-based training enabling ICRM to outperform counterparts even in the absence of context. Our extensive experiments demonstrate the efficacy of ICRM, and extensive ablations dissect and deepen our understanding of it.

We organize the rest of the exposition as follows. Section 2 reviews the fundamentals of domain generalization, centered around the concept of *environment*. Section 3 explains the basics of next-token prediction, with an emphasis on learning from *context*. Section 4 knits these threads together to propose ICRM, a framework to learn from contexts from multiple environments, supported by a

host of theory. Section 6 showcases the efficacy of our ideas in a variety of domain generalization benchmarks, and Section 7 closes the exposition with some topics for future discussion.

## 2 THE PROBLEM OF DOMAIN GENERALIZATION

The goal of domain generalization (DG) is to learn a predictor that performs well across a set of domains or environments $\mathcal{E}$ (Muandet et al., 2013). Environment indices $e \in \mathcal{E}$ list different versions of the data collection process—variations that may occur due to time, location, experimental interventions, changes in background conditions, and other contextual circumstances leading to distribution shifts (Arjovsky et al., 2019).

During training we have access to a collection of triplets $\mathcal{D} = \{(x_i, y_i, e_i)\}_{i=1}^n$. Each triplet contains a vector of features $x_i$, a target label $y_i$, and the index of the corresponding training environment $e_i \in \mathcal{E}_{\mathrm{tr}} \subset \mathcal{E}$. Each example $(x_i, y_i)$ is sampled independently from a joint distribution $P^e(X, Y)$. Using the dataset $\mathcal{D}$, we learn a predictor $h$ that maps features to labels, while minimizing the worst risk across the set of all environments $\mathcal{E}$:

$$h^* = \arg\min_h \max_{e \in \mathcal{E}} R^e(h), \tag{2}$$

where $R^e(h) = \mathbb{E}_{(X,Y) \sim P^e}[\ell(h(X), Y)]$ is the risk of the predictor $h$ in environment $e$, as measured by the expectation of the loss function $\ell$ with respect to the environment distribution $P^e$.

As one example, we could train a self-driving model $h$ to classify images $x_i$ into a label $y_i$ indicating the presence of a pedestrian. Each training example $(x_i, y_i)$ is hereby collected from one of the training cities $e_i \in \mathcal{E}_{\mathrm{tr}}$, with its own weather conditions. The goal of Equation (2) is to obtain a predictor that correctly classifies $x$ in new cities $e \in \mathcal{E}_{\mathrm{te}}$ observed during test time. This has proved to be challenging (Lechner et al., 2022), as predictors often exhibit penurious performance in unseen weather conditions.

Domain generalization is challenging because we do not have access to test environments during training time, rendering Equation (2) challenging to estimate. Therefore, to address the DG problem in practice, researchers have proposed various algorithms that make different assumptions about the invariances shared between $\mathcal{E}_{\mathrm{tr}}$ and $\mathcal{E}_{\mathrm{te}}$. In broad strokes, domain generalization algorithms fall in one of the two following categories. In the first category, domain generalization algorithms based on invariance (Muandet et al., 2013; Ganin et al., 2016; Peters et al., 2016; Arjovsky et al., 2019), illustrated in Figure 1a, regularize predictors $h(x_i^e)$ to not contain any information about the environment $e$. This however results in removing a lot of signal about the prediction task. In the second category, domain generalization algorithms based on marginal transfer (Blanchard et al., 2011; Li et al., 2016a; Zhang et al., 2020; Bao and Karaletsos, 2023) extract environment-specific information. These methods implement predictors $h(x_i^e, \phi_i^e)$, where $\phi_i^e = \frac{1}{i-1} \sum_{j=1}^{i-1} \phi(x_j^e)$ coarsely summarizes the environment $e$ in terms of previously observed instances. Different choices for $\phi$ include kernel functions (Blanchard et al., 2011, MTL), convolutional neural networks (Zhang et al., 2020, ARM), and patch embeddings (Bao and Karaletsos, 2023, Context-ViT). Alas, all of these alternatives in the second category dilute relevant features found in individual examples. For example, the size of the representation $\phi$ would have to grow linearly with the size of the training data to describe aspects corresponding to a small group of examples, such as extreme value statistics.

As a result, and despite all efforts, no proposal so far convincingly outperforms a simple empirical risk minimization baseline (Vapnik, 1998, ERM) across standard benchmarks (Gulrajani and Lopez-Paz, 2020; Gagnon-Audet et al., 2023; Yao et al., 2022). Effectively, ERM simply pools all training data together and seeks the *global* empirical risk minimizer:

$$h^\dagger = \arg\min_h \sum_{e \in \mathcal{E}_{\mathrm{tr}}} P(E = e) \cdot R^e(h). \tag{3}$$

Does the efficacy of ERM suggest that environmental information is useless? We argue that this is not the case. The key to our answer resides in a recently discovered emergent ability of next-token predictors, namely, in-context learning.

| paradigm | training data | testing data | estimates |
|---|---|---|---|
| ERM | $x, y$ | $x^{e'}$ | $P(Y \mid X)$ |
| IRM | $x, y, e$ | $x^{e'}$ | $P(Y \mid \phi^{\text{inv}}(X))$ |
| LLM | $z$ | $z_t$ and context $z_{j<t}$ | $P(Z_{t+1} \mid Z_t, \ldots, Z_1)$ |
| ICRM | $x, y, e$ | $x_t^{e'}$ and context $c_t^{e'} = (x_j^{e'})_{j<t}$ | $P(Y\mid X, C) \rightsquigarrow P^{e'}(Y \mid X)$ |

Table 1: Different learning paradigms discussed in this work, together with their training data and testing data formats, as well as the estimated predictors. In our ICRM, we amortize the current input $x^{e'}$ and its context $c^{e'}$, containing previously experienced unlabeled examples from the same environment $e'$, and "zoom-in" ($\rightsquigarrow$) to the appropriate local risk minimizer.

## 3 NEXT-TOKEN PREDICTORS AND IN-CONTEXT LEARNING

In next-token prediction, we aim to learn the conditional distribution

$$P(Z_{t+1} = z_{t+1} \mid Z_t = z_t, \ldots Z_1 = z_1), \tag{4}$$

describing the probability of observing the token $z_{t+1}$ after having observed the sequence of tokens $(z_1, \ldots, z_t)$. The quintessential next-token prediction task is language modeling (Bengio et al., 2000), where the sequence of tokens represents a snippet of natural language text. Most LLMs estimate Equation (4) via a transformer $z_{t+1} \approx h(z_t; z_{t-1}, \ldots, z_1)$ (Vaswani et al., 2017).

LLMs exhibit a certain ability, termed in-context learning (ICL), relevant to our interests. ICL is the ability to describe and learn about a learning problem from the sequence of tokens (typically labeled $(x, y)$ pairs) itself, called the context or prompt. As an example, trained LLM would complete the sequence "France-Paris Italy-Rome Spain-" with the sequence "Madrid," demonstrating its ability to infer from a few input-output pairs that the user is demanding a capital prediction task. To illustrate this ability for contexts without labels, consider the two following sequences:

$$\underbrace{\text{"You are talking to a teenager.}}_{\text{context } c_1} \; \underbrace{\text{Write a poem on gravitational fields."}}_{x_1}$$

$$\underbrace{\text{"You are talking to a Physics graduate.}}_{\text{context } c_2} \; \underbrace{\text{Write a poem on gravitational fields."}}_{x_2}$$

As widely observed, LLMs answer differently to these two sequences, producing two poems, say $y_1$ and $y_2$, each adapted to the assumed audience. While nothing unexpected is happening here at the sequence level—the model simply produces a high-likelihood continuation to each of the two prompts—we observe a degree of compositional generalization, because the LLM can provide different but correct answers to the same question $x_1 = x_2$ when presented under two contexts $c_1$ and $c_2$. By addressing the general task of *in*-distribution language modeling, LLMs can attain significant *out*-of-distribution abilities in a multitude of specific tasks—such as writing poems. ICL is reminiscent of meta-learning (Schmidhuber, 1987; Finn et al., 2017), yet it seamlessly accommodates to contexts without labels, and does not require updating the parameters of the model. ICL is also similar in spirit to test-time adaptation (Wang et al., 2020, TTA); however, TTA often requires updating model parameters with an externally hand-crafted objective.

Notably, ICL emerges without supervision. The training corpus does not contain any explicit division between questions and their context beyond the natural order of the words within each snippet of language in the training data. However, since we train the machine to produce an enormous amount of completions, some of which start with partially overlapping contexts, the predictor has the opportunity to amortize learning to a significant degree i.e. use the trained model to generalize across unseen distributions rather than explicitly optimizing a separate model for each distribution. While the machine may have never observed the context $\tilde{c}_1 = $ "You are now *speaking* to a teenager," its semantic similarity to $c_1$ above—plus other similar contexts where the word *speaking* appears— may endow OOD generalization. This is the desired ability to generalize over environments described in the previous section, which remained out of reach when using coarse domain indices.

## 4 ADAPTIVE DOMAIN GENERALIZATION VIA IN-CONTEXT LEARNING

Our exposition has so far laid out two threads. First, Section 2 motivated the need for domain generalization algorithms capable of extracting relevant environment-specific features, at both the example and distributional levels. To this end, we have argued to move away from coarse environment indices, and towards rich and amortizable descriptions shared in new circumstances. Second, Section 3 suggests understanding *context* as an opportunity to describe *environments* in precisely this manner. We now knit these threads together with a protocol to address domain generalization with in-context learners.

**In-Context Risk Minimization** (**ICRM**, Figure 1c):

- Collect a dataset of triplets $\mathcal{D} = \{(x_i, y_i, e_i)\}_{i=1}^n$ as described in Section 2. Initialize a context-based predictor $\hat{y} = h(x; c)$, tasked with predicting the label $y$ associated to the input $x$, as supported by the context $c$.

- During training, select $e \in \mathcal{E}_{\text{tr}}$ at random. Draw $t$ examples from this environment at random, construct one input sequence $(x_1^e, \ldots, x_t^e)$ and its associated target sequence $(y_1^e, \ldots, y_t^e)$. Update the context-based predictor to minimize the auto-regressive loss $\sum_{j=1}^t \ell(h(x_j^e; c_j^e), y_j^e)$, where the context is $c_j^e = (x_1^e, \ldots, x_{j-1}^e)$, for all $j = 2, \ldots, t$, and $c_1^e = \emptyset$.

- During test time, a sequence of inputs $(x_1^{e'}, \ldots, x_{t'}^{e'})$ arrives for prediction, one by one, all from the test environment $e' \in \mathcal{E}_{\text{te}}$. We predict $\hat{y}_j^{e'} = h(x_j^{e'}, c_j^{e'})$ for $x_j^{e'}$, where the context $c_j^{e'} = (x_1^{e'}, \ldots, x_{j-1}^{e'})$, for all $j = 2, \ldots, t'$, and $c_1^{e'} = \emptyset$.

A few critical remarks about the above proposal are in order. The idea of using contextual information to aid the prediction through attention mechanisms in transformers has been used in earlier works on neural processes (Kim et al., 2019; Nguyen and Grover, 2022), prior data fitted networks (Müller et al., 2021) and more recent works that study the mechanisms underlying in-context learning (Garg et al., 2022; Akyürek et al., 2022). These works leverage labeled contextual data. Our proposal embraces the challenge of domain generalization and only uses unlabeled data from the environments both at train and test time. The most natural way to construct contexts is to use past samples that appear in the natural order in which data was collected (e.g., video). Since existing DG datasets do not provide such a refined ordering, we build contexts using environment indices that are more readily available. The proposal also requires the data at test time to be sampled from the same or slowly changing environments. While the proposal is strongly inspired from LLMs in that both pay attention to current query and the contextual information, there are differences namely we predict the label of the input and not the next $x$ in the sequence.

Next, we develop theoretical guarantees on the behavior of ICRM. The results below concern the joint distribution of $\big((X_1, \cdots X_t), (Y_1, \ldots, Y_t), E\big)$, where each $X_j, Y_j$ is an independent draw from environment $E$ with distribution $P^E(X, Y)$. For query $X_j$, the context preceding it is $C_j = (X_1, \cdots, X_{j-1})$ and the environment underlying this context is $E$. To orient ourselves around these results, we recall three predictors featured in the exposition so far. First, the global empirical risk minimizer over the pooled training data, denoted by $h^\dagger$ in Equation (3), estimates $P(Y \mid X)$. Second, the environment risk minimizer estimates $P(Y \mid X, E)$. Third, our in-context risk minimizer, denoted by

$$\tilde{h} = \arg\min_h \sum_{j=1}^t \mathbb{E}_{(X_j, C_j, Y_j)}[\ell(h(X_j; C_j), Y_j)]. \tag{5}$$

estimates the conditional expectation $E(Y \mid X, C)$. The sequel focuses on the binary cross-entropy loss $\ell$. Our first result shows that, in the absence of context, ICRM *zooms-out* to behave conservatively.

**Proposition 1** (Zoom-out). *In the absence of context,* ICRM *behaves as the global empirical risk minimizer across the support of the training environments, i.e.,* $\tilde{h}(\cdot ; \emptyset) = h^\dagger(\cdot)$.

The above result is built on the insight that ICRM is Bayes optimal at all context lengths and ERM is Bayes optimal for context $c = \emptyset$. Having established the connection between ICRM and ERM in

the absence of any context, we now study the benefits of ICRM in the presence of sufficiently long contexts. The following result shows that, when provided with context from a training environment $e \in \mathcal{E}_{\text{tr}}$, our ICRM *zooms-in* and behaves like the appropriate environment risk minimizer, as shown in Table 1. We assume that $P(Y = 1 \mid X = x, E = e)$ is parametrized by a function $h^\star(x, \theta_x^e)$, where $\theta_x^e$ describes features of the environment relevant to the query $x$, for all $e \in \mathcal{E}$. We also assume there exists an ideal *amortization function* $b$ that takes as input the query $X$ and context $C_t$ preceding it—both sampled from environment $E$—and approximates $\theta_X^E$. Formally, the sequence of random variables $b(X, C_t)$ indexed by $t$ converges almost surely to the random variable $\theta_X^E$.

**Theorem 1** (Full iid zoom-in). *Let $h^\star(x, \theta_x^e)$ describe $P(Y = 1 \mid X = x, E = e)$ for all $e \in \mathcal{E}$. Further, we assume the existence of an* amortization function $b(X, C_t) \overset{a.s.}{\to} \theta_X^E$. *Then,* ICRM *zooms-in on the environment risk minimizer and achieves a cross-entropy loss over the training distribution*

$$\lim_{t \to \infty} H(Y \mid X, C_t) = H(Y \mid X, E).$$

*Further, if $I(Y; E \mid X) > 0$,* ICRM *has better performance than the global risk minimizer.*

Theorem 1 states that ICRM converges to empirical risk minimizer of the environment under infinitely long contexts. Next, we show that ICRM can partially zoom-in on the appropriate environment risk minimizer even with contexts of length of one.

**Theorem 2** (Partial iid zoom-in). *Suppose the joint distribution $((X_1, \cdots X_t), (Y_1, \ldots, Y_t), E)$ is Markov with respect to a Bayesian network. The query $X$ and the environment $E$ are statistically dependent and form the Markov blanket of $Y$. Then* ICRM *partially zooms-in on the environment risk minimizer, improving over the performance of the global empirical risk minimizer in terms of the cross-entropy loss. Further, the improvement is strictly monotonic in context length $t$.*

Next, we move to the out-of-distribution setting where the test environments can be different from the training environments. To provide theory for a domain generalization result, we must place some assumptions on the data generation process. In particular, and for all $e \in \mathcal{E}$, let $z \mid y, e \sim \mathcal{N}(\mu_e^y, \Sigma_e^y)$, and $x \leftarrow g(z)$ where the latent variables $z$ are sampled conditional on the label $y$ and environment $e$ from a Gaussian distribution with mean and covariance depending on $(y, e)$, and are then mixed by a map $g$ to generate the observations $x$. We summarize the environment as a parameter vector $\gamma_e = \left[ (p_e^y, \mu_e^y, \Sigma_e^y)_{y \in \{0,1\}} \right]$, where $p_e^y$ is the probability of label $y$ in environment $e$. Our next result shows that ICL algorithms that learn $h(x; c)$ exhibit robust behavior under distribution shifts. In contrast, such guarantees are not known for algorithms that generate predictors of the form $h(x)$.

Define $\delta_e$ to be a permutation of $\gamma_e$ that swaps its two components. We construct the Voronoi cells corresponding to the points in the union of sets $\{\gamma_e\}_{e \in \mathcal{E}_{tr}}$ and $\{\delta_e\}_{e \in \mathcal{E}_{tr}}$. The set of points in the Voronoi cells corresponding to the set of points $\{\gamma_e\}_{e \in \mathcal{E}_{tr}}$ define the *Voronoi cells of the training environments*. Next, we show that there exists an ICL algorithm, which takes the data from multiple environments as input and outputs a predictor that takes current query and context as input, whose output predictors perform well in novel test environments even those that are sufficiently far away from the training environments, so long as they are in the Voronoi cells of training environments.

**Theorem 3** (Full OOD zoom-in). *Consider data triplets $(x, y, e)$ generated from $z \sim \mathcal{N}(\mu_e^y, \Sigma_e^y)$ and $x \leftarrow g(z)$, $\forall e \in \mathcal{E}$, where $g$ is the identity map (see Appendix A for extension to general diffeomorphism $g$). There exists an ICL algorithm that in the limit of infinitely long contexts produces Bayes optimal predictions for all the test environments in the Voronoi cells of the training environments.*

## 5 ICRM UNDER THE LENS OF INVARIANCE

Common advice in domain generalization recommends following the *invariance principle* to learn robust predictors (Peters et al., 2016; Arjovsky et al., 2019). One simple version of the invariance principle is to "select those inputs leading to stable predictors across training environments." At first sight, one could argue that the proposed ICRM does not adhere to such a principle, as it is adapting to environment-specific information provided in the form of context. As we shall now illustrate, ICRM's implementation of ERM on the extended input-context feature space reveals invariant predictors that a vanilla implementation of ERM on the standard feature space fails to find. To see this, consider a linear least-squares regression problem mapping two dimensional inputs $x = (x^1, x^2)$ into a target $y$ under environments $e \in \mathcal{E}$ as $y = \alpha \cdot x^1 + \beta \cdot \mu_e^2 + \varepsilon$, where $\mu_e^i = \mathbb{E}[X^i \mid E = e]$, the pair $(\alpha, \beta)$

are invariant regression coefficients, and $\varepsilon$ is an independent noise term. We make one simplifying assumption for pedagogic purposes. During training, we provide ICRM training directly with the relevant extended feature space $(x^1, x^2, \mu_e^1, \mu_e^2)$, instead of requiring the algorithm to learn such representation from general-form sequential context.

In this setup, ICRM learns to predict using $\alpha \cdot x^1 + 0 \cdot x^2 + 0 \cdot \mu_e^1 + \beta \cdot \mu_e^2$. In contrast, ERM trains a linear model on $(x^1, x^2)$ and predicts using $\tilde{\alpha} \cdot x^1 + \tilde{\beta} \cdot x^2$. The main point is: if $\beta \neq 0$ and $\mathrm{cov}(X^1, X^2) \neq 0$, then $\tilde{\alpha} \neq \alpha$, and the error of ERM in a new environment grows with the variance of $x^1$. On the other hand, ICRM estimates the true invariant coefficient $\alpha$. and the resulting error is independent of variance of $x^1$, even in the absence of context during test time. As a result, ICRM exhibits better out-of-distribution performance than ERM without any contextual information at test time. For a derivation and generalization of these claims, see Appendix A.

We believe that ICRM, and more generally ICL, provide a novel view on invariance. On the one hand, prior DG algorithms advocated to remove features as a guide to reveal invariance. On the other hand, in-context learners suggest that extending features with context affords invariance otherwise unnoticed. This needs further clarification: while the process of zooming-in to an environment risk minimizer does not provide us with an invariant predictor over the original feature space, the *process of zooming-in* is often an invariant mechanism over the extended feature space. These points are reminiscent of the concept of "fragility" in the philosophy of causation (Menzies and Beebee, 2020). Does smoking cause cancer? Not invariably across all contexts or environments. Yet, smoking does cause cancer invariably—across all contexts or environments—when extending the feature space as to include additional causes such as diet, genetic predispositions, and the number of smoked cigarettes. The ever-growing collection of causes approaches what Mill (1856) called the *total cause*, a large context sharpening invariance at the expense of constraining the diameter of the environment. In the extreme, when constraining the environment to contain only one smoker, the outcome of lung cancer disease invariably follows.

## 6 EXPERIMENTS

To evaluate the efficacy of ICRM, our experiments address the following questions:

1. How does ICRM fare against competitive DG algorithms, for different context sizes?
2. What is the impact of model architecture on ICRM's gains?
3. Can ICRM search for query relevant signals in the context?

In our experiments, we compare ICRM against marginal transfer methods such as Adaptive Risk Minimization (Zhang et al., 2020, ARM), and test-time adaptation proposals such as TENT (Wang et al., 2020). As a strong baseline, we also include ERM in our experimental protocol. To ensure a fair comparison across different algorithms for each dataset, we use a standardized neural network backbone (ConvNet or ResNet-50 depending on the dataset) as described in Appendix C.4. For ICRM, the same backbone is used to featurize the input, which is then processed by the decoder-only GPT-2 (Radford et al., 2019). During both training and inference for ICRM, data in a sequence is sampled from the same environment. For fair comparisons, we adhere to DomainBed's protocols for training, hyper-parameter tuning, and testing (Gulrajani and Lopez-Paz, 2020), details in Appendix C.4. We assess these methods across six image classification benchmarks, featuring diversity shift – FEMNIST (Cohen et al., 2017) contains MNIST digits and handwritten letters from individual writers as environments. Rotated MNIST concerns varied rotational angles as environments. Tiny ImageNet-C and CIFAR10-C (Hendrycks and Dietterich, 2019) introduce diverse image corruptions to create multiple environments. WILDS Camelyon17 (Koh et al., 2021) studies tumor detection with data from multiple hospitals as distinct environments and Imagenet-R (Hendrycks et al., 2021) contains various renditions (e.g., paintings, embroidery, etc.) of ImageNet object classes as domains. More details are provided in Appendix C.3. We consider addressing correlation shifts (Ye et al., 2022) as a future work of our paper, as it involves scenarios where the test domain contains naturally occurring subpopulations or time-based shifts without clear domain separation

### 6.1 ADAPTATION TO DISTRIBUTION SHIFT

To study the adaptation of other approaches to distribution shifts, we report the average performance on four datasets across three independent runs of the entire sweep for test context lengths of 0, 25, 50,

Table 2: Average/worst OOD test accuracy for different context lengths, for Adaptive Risk Minimization (ARM), Empirical Risk Minimization (ERM), Test Entropy Minimization (TENT) and our ICRM on FEMNIST, Rotated MNIST, WILDS Camelyon17 and Tiny-ImageNet-C.

| Data / method | Average test accuracy | | | | | Worst case test accuracy | | | | |
|---|---|---|---|---|---|---|---|---|---|---|
| **FEMNIST** | 0 | 25 | 50 | 75 | 100 | 0 | 25 | 50 | 75 | 100 |
| ARM | 49.5 | 83.9 | 84.4 | 84.7 | 84.6 | 23.6 | 59.5 | 60.7 | 57.0 | 58.8 |
| TENT | 78.1 | 77.9 | 81.2 | 82.5 | 83.3 | 55.2 | 57.2 | 63.3 | 65.9 | 67.2 |
| ERM | **79.3** | 79.3 | 79.3 | 79.3 | 79.3 | 59.0 | 59.0 | 59.0 | 59.0 | 59.0 |
| ICRM | 78.7 | **87.2** | **87.4** | **87.5** | **87.8** | **59.8** | **69.3** | **70.6** | **70.6** | **70.6** |
| **Rotated MNIST** | 0 | 25 | 50 | 75 | 100 | 0 | 25 | 50 | 75 | 100 |
| ARM | 36.5 | 94.2 | 95.1 | 95.3 | 95.5 | 28.2 | 85.3 | 87.2 | 87.9 | 87.9 |
| TENT | 94.1 | 88.0 | 91.9 | 93.8 | 94.3 | 80.2 | 88.5 | 88.5 | 80.2 | 81.3 |
| ERM | **94.2** | 94.2 | 94.2 | 94.2 | 94.2 | 80.8 | 80.8 | 80.8 | 80.8 | 80.8 |
| ICRM | 93.6 | **96.1** | **96.2** | **96.2** | **96.2** | **82.5** | **88.5** | **88.5** | **88.8** | **88.8** |
| **WILDS Camelyon17** | 0 | 25 | 50 | 75 | 100 | 0 | 25 | 50 | 75 | 100 |
| ARM | 61.2 | 59.5 | 59.7 | 59.7 | 59.7 | | | | | |
| TENT | 67.9 | 81.8 | 87.2 | 89.4 | 89.4 | | same as average accuracy | | | |
| ERM | 68.6 | 68.6 | 68.6 | 68.6 | 68.6 | | | | | |
| ICRM | **92.0** | **90.7** | **90.8** | **90.8** | **90.8** | | | | | |
| **Tiny ImageNet-C** | 0 | 25 | 50 | 75 | 100 | 0 | 25 | 50 | 75 | 100 |
| ARM | 30.8 | 31.0 | 31.0 | 31.0 | 31.0 | 8.2 | 8.3 | 8.2 | 8.3 | 8.2 |
| TENT | 31.7 | 1.6 | 1.7 | 2.0 | 2.1 | 9.4 | 1.2 | 1.4 | 1.6 | 1.6 |
| ERM | 31.8 | 31.8 | 31.8 | 31.8 | 31.8 | 9.5 | 9.5 | 9.5 | 9.5 | 9.5 |
| ICRM | **38.3** | **39.2** | **39.2** | **39.2** | **39.2** | **18.8** | **19.2** | **19.5** | **19.5** | **19.4** |

75, and 100 samples. The results for other datasets and more DG algorithms are reported in Table 7 and Table 8. As Table 2 shows, ICRM outperforms all methods across context lengths, except at null context length on MNIST datasets, where ERM exceeds by 1%. Further, our gains persist over both the worst group and average accuracy across testing environments. Figure 5 zooms into the model's performance between no-context and 25 context samples, highlighting the consistent superiority of ICRM even with small contexts. Additionally, ICRM demonstrates gains in performance even in the absence of test context. Specifically for both WILDS Camelyon17 and Tiny ImageNet-C, ICRM outperforms baselines despite not leveraging any context from the test environment. We hypothesize that ICRM training still benefits from contexts as to find contextual features that ERM ignores.

## 6.2 UNDERSTANDING THE IMPACT OF ARCHITECTURE

To dissect the performance gains potentially arising from ICRM's transformer architecture, we explore two additional competitors. First, we train an ERM baseline, $ERM^+$ using an identical architecture to ICRM, but without context. Second, we train an ARM baseline, $ARM^+$, where the input and context coarse summary are sent to a GPT-2 such that the model now attends to the summary through attention layers.

Table 3: Worst group OOD test accuracies for $ARM^+$ and $ERM^+$ in contrast to their base algorithms, ARM and ERM across FEMNIST, Rotated MNIST, WILDS, Camelyon17, and Tiny-ImageNet-C.

| Dataset | **ARM** | | $\mathbf{ARM^+}$ | | **ERM** | | $\mathbf{ERM^+}$ | |
|---|---|---|---|---|---|---|---|---|
| # Context Samples | 0 | 100 | 0 | 100 | 0 | 100 | 0 | 100 |
| FEMNIST | 23.6 | 58.8 | 51.7 | 62.0 | 59.0 | 59.0 | 53.3 | 53.3 |
| Rotated MNIST | 28.2 | 87.9 | 71.4 | 81.1 | 80.8 | 80.8 | 81.9 | 81.9 |
| WILDS Camelyon17 | 61.2 | 59.7 | 55.8 | 55.0 | 68.6 | 68.6 | 50.1 | 50.1 |
| Tiny ImageNet-C | 8.2 | 8.2 | 1.9 | 1.9 | 9.5 | 9.5 | 8.3 | 8.3 |

Table 3 presents the performance of both $ERM^+$ and $ARM^+$ relative to ERM and ARM, across the four datasets. $ARM^+$ demonstrates superior zero-shot performance over ARM on both FEMNIST and Rotated MNIST. However, ARM maintains a performance advantage over $ARM^+$ across varying counts of in-context samples on WILDS Camelyon17 and Tiny ImageNet-C, with a pronounced

difference on the latter. Similarly, ERM either matches or outperforms ERM$^+$ on all four datasets. Therefore, even with similar architectures, prior protocols fall short of the proposed ICRM.

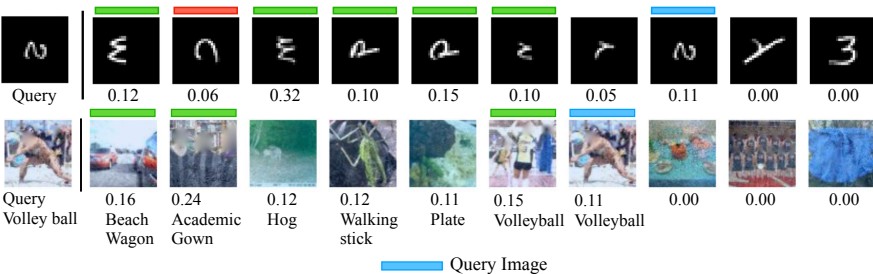

Figure 2: Attention scores for random test sequences, for ICRM on FEMNIST (top) and Tiny ImageNet-C (bottom), with the class label shown below. Samples following the query have an attention score of 0.0 because of the causal attention mechanism.

### 6.3    INVESTIGATING ATTENTION IN ICRM

As discussed in Section 2, ICRM can learn an amortization function by paying attention to the input query and context. To understand this better, we construct a random data sequence from the test environment and analyze attention scores between each example in this context and a novel input query. Figure 2 illustrates attention scores from a single head for a query image (marked in blue) for FEMNIST and Tiny ImageNet-C. Note that samples following the query have an attention score of 0.0 because of the causal attention mechanism. The top row reveals that the model selectively attends to images featuring at least two curved arcs (marked in green) while paying little attention to a partial circle (highlighted in red). Similarly, in the bottom row, the model effectively discerns *individuals* across samples within the sequence and also indicates a semantic understanding of similarity.

## 7    DISCUSSION

We introduced In-Context Risk Minimization (ICRM), a framework to address domain generalization as context-based prediction. ICRM learns in-context about environmental features by paying attention to unlabeled instances as they arrive. In such a away, ICRM dynamically zooms-in on the test environment risk minimizer, enabling competitive out-of-distribution generalization.

ICRM provides a new perspective on invariance. While prior work on DG focused on information removal as a guide to generalization, ICRM suggests that extending the feature space with the relevant environment information affords further invariance. By addressing the general problem of context-based prediction *in*-distribution, we amortize the performance over a multitude of specific *out-of*-distribution tasks. More generally, by framing DG as next-token prediction, our approach can be adapted to fully exploit data in *natural order* (such as in video or text, ordered by time and position), more closely mimicking the human learning experience—as Léon Bottou once said, *Nature does not shuffle data*. That said, we view extending ICRM to scenarios where the environment contains naturally occurring subpopulations or time-based shifts without clear separation as an exciting future direction. As a word of caution, we must conduct research to guarantee that in-context learners do not "zoom-in" on toxic spurious correlations with high predictive power in certain environments. We close with a quote from Andersen et al. (2022), for whom zooming-in

> refers to a cognitive agent's ability to intelligently ignore irrelevant information and zero in on those aspects of the world that are relevant to their goals. The relevance realization framework suggests that the brain achieves this feat by attempting to balance the competing goals of remaining efficient in the current environment while also being resilient in the face of environmental perturbations.

Paralleling the examples from Andersen et al. (2022), we would like to further understand how next-token prediction and in-context learning serves as a powerful mechanism to amortize and dynamically navigate trade-offs such as such as efficiency-resiliency, exploration-exploitation, specialization-generalization, and focusing-diversifying.

## ACKNOWLEDGEMENTS

SG and SJ acknowledge funding from the Office of Naval Research grant N00014-20-1-2023 (MURI ML-SCOPE) and NSF award CCF-2112665 (TILOS AI Institute). We are thankful to Martin Arjovsky, Léon Bottou, Elvis Dohmatob, Badr Youbi Idrissi, Maxime Oquab, and Ahmed Touati for their valuable feedback and help.

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

APPENDIX

## A   THEOREMS AND PROOFS

### A.1   PROOF OF PROPOSITION 1

**Lemma 1. ICRM** *is Bayes optimal at all context lengths. Suppose $\ell$ is the binary cross-entropy loss and the labels $Y$ are binary. The optimal in-context learner $\tilde{h}$ (equation 5) satisfies the following condition, i.e., for each $k \in [t]$*

$$\tilde{h}(x_k; c_k) = \mathbb{E}[Y | X_k = x_k, C_k = c_k], \tag{6}$$

*for almost all $(c_k, x_k)$ in the support of training distribution except over a set of a measure zero, and where the expectation is over $Y$ conditional on $[c_k, x_k]$. In other words, the in-context learner is Bayes optimal at each context length.*

*Proof.* In this result, we consider the problem of binary classification. Suppose $h(x_k; c_k)$ is the predicted probability of class $Y = 1$ conditional on $x_k$ and $c_k$. Define $\bar{h}(x_k; c_k) = [h(x_k; c_k), 1 - h(x_k; c_k)]$ describing the probability of both the classes.

From equation 5, recall that the objective of ICRM is to minimize

$$\sum_{j=1}^{t} \mathbb{E}_{(X_j, C_j, Y_j)}[\ell(h(X_j; C_j), Y_j)]. \tag{7}$$

Consider one of the terms in the sum above - $\mathbb{E}\big[\ell(h(X_k; C_k), Y_k)\big]$. Substituting $\ell$ as the cross-entropy in this term, we obtain

$$\mathbb{E}\big[\ell(h(X_k; C_k), Y_k)\big] = H(Y_k|X_k, C_k) + \mathbb{E}\big[\mathsf{KL}\big(P(Y_k|X_k, C_k)\big\|\bar{h}(X_k; C_k)\big)\big].$$

If $\bar{h}(X_k; C_k) = P(Y_k|X_k, C_k)$, then the second term in the above is zero and $\mathbb{E}\big[\ell(h(X_k; C_k), Y_k)\big]$ equals $H(Y_k|X_k; C_k)$. Since KL divergence is always non-negative, $H(Y_k|X_k, C_k)$ corresponds to the lowest value that can be achieved by $\mathbb{E}\big[\ell(h(X_k; C_k), Y_k)\big]$. If $\bar{h}(X_k; C_k) = P(Y_k|X_k, C_k)$ for all $k \in [t]$, then each of the terms in the sum in equation 7 are minimized. As a result, $\bar{h}(X_k; C_k) = P(Y_k|X_k, C_k)$ for all $k \in [t]$ is a solution to equation 5.

Consider another minimizer $h^{'}$ of equation 5 and define the corresponding distribution $\bar{h}^{'}$. For each $k \in [t]$, the second term $\mathbb{E}\big[\mathsf{KL}(P(Y_k|X_k, C_k)\|\bar{h}^{'}(X_k; C_k)\big]$ has to be zero for $\bar{h}^{'}$ to be a minimizer.

If $\mathbb{E}\big[\mathsf{KL}(P(Y_k|X_k, C_k)\|\bar{h}^{'}(X_k; C_k)\big] = 0$, then we claim that $\bar{h}^{'}(x_k; c_k) = P(Y_k|X_k = x_k, C_k = c_k)$ for almost all $(x_k, c_k)$ in the support of training distribution except over a set of measure zero. If the probability measure associated with $X_k, C_k$ is absolutely continuous w.r.t Lebesgue measure, then this claim follows from Theorem 1.6.6 (Ash and Doléans-Dade, 2000). If the probability measure associated with $X_k, C_k$ is absolutely continuous w.r.t counting measure, then this claim trivially follows. $\qquad \square$

We proved the above result for classification and cross-entropy loss for measures over $X, C$ that are either absolutely continuous w.r.t Lebesgue measure or the counting measure. It is easy to extend the above result for regressions and least square loss; see Lemma 1 in Ahuja and Lopez-Paz (2023).

**Proposition 1** (Zoom-out). *In the absence of context,* ICRM *behaves as the global empirical risk minimizer across the support of the training environments, i.e., $\tilde{h}(\cdot \ ; \ \emptyset) = h^{\dagger}(\cdot)$.*

*Proof.* From Lemma 1, it follows that $\tilde{h}(x_k; c_k) = \mathbb{E}[Y|X_k = x_k, C_k = c_k]$. The solution to empirical risk minimization is $h^{\dagger}(x) = \mathbb{E}[Y|X_1 = x]$, where the expectation is computed over the training distribution of $Y$ conditional on $x$. When the context is empty, then we have $\tilde{h}(x; \emptyset) = \mathbb{E}[Y|X_1 = x] = h^{\dagger}(x)$ for almost all $x$ in the support of training distribution except over a set of measure zero. $\qquad \square$

## A.2 PROOF OF THEOREM 1

Before stating the proof of Theorem 1, we provide an example of an ideal amortization map $b(\cdot)$.

**Example of ideal amortization map.** Consider the example from equation **??**, where $y = \alpha \cdot x^1 + \beta \cdot \mu_e^2 + \varepsilon$. $P(Y = y|X = x, E = e) = p_\varepsilon(y - \alpha x^1 - \beta \mu_e^2)$, where $p_\varepsilon$ is the probability density of noise. Observe that $P(Y = y|X = x, E = e)$ is parametrized in terms of $\mu_e^2$ and the sequence of random variables $b(X, C_t) = \frac{1}{t-1} \sum_{j=1}^{t-1} X_j^2$ converge almost surely to $\mu_e^2$, where $X_j^2$ is the second component of $X_j$ and $C_t = (X_1, \cdots, X_{t-1})$.

For ease of exposition, we start with the case when all the concerned random variables $X, Y, C_t, E, b(X, C_t)$, where $X$ is the current query and $Y$ is its label and $C_t$ is the context preceeding it sampled from environment $E$, and $b(\cdot)$ is the ideal amortization map, are discrete-valued with a finite support. Subsequently, we study more general settings.

**Theorem 1** (Full iid zoom-in). *Let $h^\star(x, \theta_x^e)$ describe $P(Y = 1 \mid X = x, E = e)$ for all $e \in \mathcal{E}$. Further, we assume the existence of an amortization function $b(X, C_t) \overset{a.s.}{\to} \theta_X^E$. Then, ICRM zooms-in on the environment risk minimizer and achieves a cross-entropy loss over the training distribution*

$$\lim_{t \to \infty} H(Y \mid X, C_t) = H(Y \mid X, E).$$

*Further, if $I(Y; E \mid X) > 0$, ICRM has better performance than the global risk minimizer.*

*Proof.* As stated above, in this proof, we work with discrete-valued $X, Y, C_t, E, b(X, C_t)$ that also have finite support, where $X$ is the current query and $Y$ is its label and $C_t$ is the context preceeding it sampled from environment $E$, and $b(\cdot)$ is the ideal amortization map. Subsequently, we study more general settings.

Since each $(X_j, Y_j)$ is sampled independently given a training environment $E$, we can conclude $I(Y; C_t | X, E) = 0$. Therefore,

$$I(Y; C_t | X, E) = 0 \implies H(Y | X, E) = H(Y | X, E, C_t).$$

Observe that for all $t \in \mathbb{Z}_+$

$$H(Y | X, E) = H(Y | X, E, C_t) \leq H(Y | X, C_t) \leq H(Y | X, b(X, C_t)), \tag{8}$$

where $\mathbb{Z}_+$ is the set of all positive integers. The first inequality in the above follows from the fact that conditioning reduces entropy. For the second inequality, we use the following property. Consider $U, V$ as two random variables and define $W = a(V)$. Observe that $I(U; W | V) = 0 \implies H(U | V) = H(U | V, W) \leq H(U | W)$.

Since the inequality above equation 8 holds for all $t$, we obtain

$$H(Y | X, E) \leq \lim_{t \to \infty} H(Y | X, C_t) \leq \lim_{t \to \infty} H(Y | X, b(X, C_t)). \tag{9}$$

In the above, we use the following property. If $a_n \leq b_n, \forall n \in \mathbb{Z}_+$ and $\lim_{n \to \infty} a_n$ and $\lim_{n \to \infty} b_n$ exist, then $\lim_{n \to \infty} a_n \leq \lim_{n \to \infty} b_n$. In what follows, we will show that both the limits $\lim_{t \to \infty} H(Y | X, C_t)$ and $\lim_{t \to \infty} H(Y | X, b(X, C_t))$ exist. First observe that $H(Y | X, C_{t+1}) \leq H(Y | X, C_t)$ for all $t$ as a result the sequence is decreasing bounded below by $0$ and thus from monotone convergence theorem (Rudin, 1953) $\lim_{t \to \infty} H(Y | X, C_t)$ exists. Next, we will show that $\lim_{t \to \infty} H(Y | X, b(X, C_t)) = H(Y | X, E)$. We will then combine it equation 9 to obtain what we intend to prove, i.e., $\lim_{t \to \infty} H(Y | X, C_t) = H(Y | X, E)$.

For each $X = x$ and $E = e$ in the support of training distribution, we argue that $b(X, C_t) \overset{a.s.}{\to} \theta_x^e$. Suppose this was not true. This implies that the probability that $P(\lim_{t \to \infty} b(X, C_t) \neq \theta_x^e | X = x, E = e) = \beta > 0$. Since $X = x, E = e$ occurs with a finite probability (as $X$ and $E$ are discrete-valued and $x, e$ is in the support) say $\alpha$, then $\alpha\beta$ fraction of sequences of $b(X, C_t)$ do not converge to $\theta_x^e$, which contradicts the assumption that $b(X, C_t) \overset{a.s.}{\to} \theta_X^E$.

Consider a $(x, \theta)$ from the support of $(X, \theta_X^E)$, where $X$ is the current query and $E$ is the environment from which $X$ and context preceeding it is sampled. Let us consider the distribution $P(Y | X, b(X, C_t))$.

$$P(Y = y | X = x, b(X, C_t) = \theta) = \frac{P(Y = y, X = x, b(X, C_t) = \theta)}{P(X = x, b(X, C_t) = \theta)} \tag{10}$$

We simplify $\lim_{t \to \infty} P(Y | X, b(X, C_t))$ below.

$$\lim_{t \to \infty} P(Y = y | X = x, b(X, C_t) = \theta) = \frac{\lim_{t \to \infty} P(Y = y, X = x, b(X, C_t) = \theta)}{\lim_{t \to \infty} P(X = x, b(X, C_t) = \theta)} \tag{11}$$

We show that the limits of the numerator and denominator exist (and non-zero for the denominator) and we simplify these separately below.

$$\lim_{t\to\infty} P(Y=y, X=x, b(X,C_t)=\theta) = \lim_{t\to\infty} \sum_e P(Y=y, X=x, E=e, b(X,C_t)=\theta)$$

$$= \sum_e P(Y=y|X=x, E=e) \lim_{t\to\infty} P(X=x, E=e, b(X,C_t)=\theta) \tag{12}$$

$$= \sum_e P(Y=y|X=x, E=e) P(X=x, E=e) \lim_{t\to\infty} P(b(X,C_t)=\theta|X=x, E=e)$$

In the simplification above, we firstly used the fact that we can interchange sum and limits, this is true because $e$ only takes finitely many values. In the simplification above, we also use the fact $Y \perp C_t | X, E$. Since $b(X,C_t)$ converges to $\theta_x^e$ almost surely, the distribution $\lim_{t\to\infty} P(b(X,C_t) = \theta|X=x, E=e)$ takes a value one if $\theta = \theta_x^e$ and zero otherwise. As a result, the above expression becomes

$$\lim_{t\to\infty} P(Y=y, X=x, b(X,C_t)=\theta) = \sum_{e\in\mathcal{E}_{x,\theta}} P(Y=y|X=x, E=e) P(X=x, E=e). \tag{13}$$

where $\mathcal{E}_{x,\theta}$ is the set of all the environments observed conditional on $X=x$ with $\theta_x^e = \theta$. Observe that all the environments in $\mathcal{E}_{x,\theta}$ have the same $P(Y=1|X=x, E=e)$ given by $h^\star(x,\theta)$. We can write

$$\lim_{t\to\infty} P(Y=1, X=x, b(X,C_t)=\theta) = h^\star(x,\theta) \sum_{e\in\mathcal{E}_{x,\theta}} P(X=x, E=e). \tag{14}$$

We simplify $\lim_{t\to\infty} P(X=x, b(X,C_t)=\theta)$ in a similar manner to obtain

$$\lim_{t\to\infty} P(X=x, b(X,C_t)=\theta) = \sum_{e\in\mathcal{E}_{x,\theta}} P(X=x, E=e). \tag{15}$$

Observe that the denominator is positive and not zero because $x, \theta$ is in support of $X, \theta_X^E$. We use equation 14 and equation 15 to obtain

$$\begin{aligned} \lim_{t\to\infty} P(Y=1|X=x, b(X,C_t)=\theta) &= \frac{\lim_{t\to\infty} P(Y=1, X=x, b(X,C_t)=\theta)}{\lim_{t\to\infty} P(X=x, b(X,C_t)=\theta)} \\ &= \frac{h^\star(x,\theta) \sum_{e\in\mathcal{E}_{x,\theta}} P(X=x, E=e)}{\sum_{e\in\mathcal{E}_{x,\theta}} P(X=x, E=e)} = h^\star(x,\theta). \end{aligned} \tag{16}$$

Therefore,

$$\lim_{t\to\infty} P(Y=1|X=x, b(X,C_t)=\theta) = P(Y=1|X=x, E=e). \tag{17}$$

where $e$ is any environment in $\mathcal{E}_{x,\theta}$, i.e., it is in the support of data sampled with $X=x$ and that also satisfies $\theta_x^e = \theta$.

$$\lim_{t\to\infty} H(Y|X, b(X,C_t)) = \sum_{x,\theta} \lim_{t\to\infty} P(X=x, b(X,C_t)=\theta) \lim_{t\to\infty} H(Y|X=x, b(X,C_t)=\theta)$$

$$\sum_{x,\theta} \Big( \sum_{\tilde{e}\in\mathcal{E}_{x,\theta}} P(X=x, E=\tilde{e}) \Big) \lim_{t\to\infty} H(Y|X=x, b(X,C_t)=\theta)$$

$$\tag{18}$$

In the above simplification, we again swap limits and sum because the summation is over a finite set of values. From equation 17, it follows that $\lim_{t\to\infty} H(Y|X = x, b(X, C_t) = \theta) = H(Y|X = x, E = e)$, where $e$ is any environment in $\mathcal{E}_{x,\theta}$. We use this in the above to get

$$\lim_{t\to\infty} H(Y|X, b(X, C_t)) = \sum_{x,\theta} \Big( \sum_{\tilde{e}\in\mathcal{E}_{x,\theta}} P(X = x, E = \tilde{e}) \Big) H(Y|X = x, E = e)$$

$$= \sum_{x,\theta} \Big( \sum_{\tilde{e}\in\mathcal{E}_{x,\theta}} P(X = x, E = \tilde{e}) \Big) H(Y|X = x, E = \tilde{e}) \tag{19}$$

$$= \sum_{x,\tilde{e}} P(X = x, E = \tilde{e}) H(Y|X = x, E = \tilde{e}) = H(Y|X, E).$$

We combine the above with equation 9 to obtain $\lim_{t\to\infty} H(Y|X, C_t) = H(Y|X, E)$. Finally, observe that if $I(Y; E|X) > 0 \implies H(Y|X, E) < H(Y|X)$. Since $\lim_{t\to\infty} H(Y|X, C_t) = H(Y|X, E)$, ICRM impoves over ERM that attains a cross-entropy loss of $H(Y|X)$.

This completes the proof.

$\square$

We now extend the argument to setting beyond discrete random variables. In particular, we consider settings where $X, E, b(X, C_t)$ can be either discrete or continuous random variables. In the notation to follow, we use $dP$ to denote the Radon-Nikodym derivatives. For discrete random variable, the Radon-Nikodym derivatives correspond to the standard probability mass function and for continuous random variables it would correspond to standard probability density functions. We operate under some regularity assumptions. We assume that the support of $E$ has a finite volume and the support of $(X, b(X, C_t))$ has a finite volume for all $t$. Further, we assume that the Radon-Nikodym derivative of the joint $dP(X = x, E = e, b(X, C_t) = \theta)$ is bounded above. While much of the proof that follows is same as the previous proof, we repeat the arguments for completeness.

**Theorem 4.** *Let $h^\star(x, \theta_x^e)$ describe $dP(Y = 1 \mid X = x, E = e)$ for all $e \in \mathcal{E}$. Further, we assume the existence of an* amortization function $b(X, C_t) \overset{a.s.}{\to} \theta_X^E$. *Then,* ICRM *zooms-in on the environment risk minimizer and achieves a cross-entropy loss over the training distribution*

$$\lim_{t\to\infty} H(Y \mid X, C_t) = H(Y \mid X, E).$$

*Further, if $I(Y; E \mid X) > 0$,* ICRM *has better performance than the global risk minimizer.*

*Proof.* Since each $(X_j, Y_j)$ is sampled independently given a training environment $E$, we can conclude $I(Y; C_t|X, E) = 0$. Therefore,

$$I(Y; C_t|X, E) = 0 \implies H(Y|X, E) = H(Y|X, E, C_t).$$

Observe that for all $t \in \mathbb{Z}_+$

$$H(Y|X, E) = H(Y|X, E, C_t) \le H(Y|X, C_t) \le H(Y|X, b(X, C_t)), \tag{20}$$

where $\mathbb{Z}_+$ is the set of all positive integers. The first inequality in the above follows from the fact that conditioning reduces entropy. For the second inequality, we use the following property. Consider $U, V$ as two random variables and define $W = a(V)$. Observe that $I(U; W|V) = 0 \implies H(U|V) = H(U|V, W) \le H(U|W)$.

Since the inequality above equation 20 holds for all $t$, we obtain

$$H(Y|X, E) \le \lim_{t\to\infty} H(Y|X, C_t) \le \lim_{t\to\infty} H(Y|X, b(X, C_t)). \tag{21}$$

In the above, we use the following property. If $a_n \le b_n, \forall n \in \mathbb{Z}_+$ and $\lim_{n\to\infty} a_n$ and $\lim_{n\to\infty} b_n$ exist, then $\lim_{n\to\infty} a_n \le \lim_{n\to\infty} b_n$. In what follows, we will show that both the limits $\lim_{t\to\infty} H(Y|X, C_t)$ and $\lim_{t\to\infty} H(Y|X, b(X, C_t))$ exist. First observe that $H(Y|X, C_{t+1}) \le H(Y|X, C_t)$ for all $t$ as a result the sequence is decreasing bounded below by $0$ and thus from Monotone convergence theorem $\lim_{t\to\infty} H(Y|X, C_t)$ exists. Next, we will show that

$\lim_{t\to\infty} H(Y|X, b(X, C_t)) = H(Y|X, E)$. We will then combine it with equation 21 to obtain what we intend to prove, i.e., $\lim_{t\to\infty} H(Y|X, C_t) = H(Y|X, E)$.

For each $X = x$ and $E = e$ in the support except over a set of probability measure zero, we argue that $b(X, C_t) \overset{a.s.}{\to} \theta_x^e$. Suppose this was not true. Define $\Gamma$ to be the set of values of $x, e$ for which $b(X, C_t) \overset{a.s.}{\not\to} \theta_x^e$. Let $P((X, E) \in \Gamma) > 0$ and the probability that $P(\lim_{t\to\infty} b(X, C_t) \neq \theta_X^E | (X, E) \in \Gamma) > 0$. If this is true then $P(\lim_{t\to\infty} b(X, C_t) \neq \theta_x^e) > 0$ contradicts the fact that $b(X, C_t) \overset{a.s.}{\to} \theta_X^E$. Therefore, $P((X, E) \in \Gamma) = 0$.

Consider a $(x, \theta)$ from the support of $(X, \theta_X^E)$ except from $\Gamma$, where $X$ is the current query and $E$ is the environment from which $X$ and context preceeding it is sampled. Let us consider the distribution $dP(Y|X, b(X, C_t))$.

$$dP(Y = y | X = x, b(X, C_t) = \theta) = \frac{dP(Y = y, X = x, b(X, C_t) = \theta)}{dP(X = x, b(X, C_t) = \theta)} \qquad (22)$$

We simplify $\lim_{t\to\infty} dP(Y = y | X = x, b(X, C_t) = \theta)$ below.

$$\lim_{t\to\infty} dP(Y = y | X = x, b(X, C_t) = \theta) = \frac{\lim_{t\to\infty} dP(Y = y, X = x, b(X, C_t) = \theta)}{\lim_{t\to\infty} dP(X = x, b(X, C_t) = \theta)} \qquad (23)$$

We simplify the numerator and the denominator of the above separately.

$$\lim_{t\to\infty} dP(Y = y, X = x, b(X, C_t) = \theta) = \lim_{t\to\infty} \int_e dP(Y = y, X = x, E = e, b(X, C_t) = \theta)$$
$$= \int_e dP(Y = y | X = x, E = e) \lim_{t\to\infty} dP(X = x, E = e, b(X, C_t) = \theta)$$
$$= \int_e dP(Y = y | X = x, E = e) dP(X = x, E = e) \lim_{t\to\infty} dP(b(X, C_t) = \theta | X = x, E = e) \qquad (24)$$

In the above, we use dominated convergence theorem (Ash and Doléans-Dade, 2000) to swap limit and the integrals (to use dominated convergence theorem, we use the fact that the $dP(X = x, E = e, b(X, C_t) = \theta)$ is bounded and support of $E$ has a finite volume). In the simplification above, we also use the fact $Y \perp C_t | X, E$. Since $b(X, C_t)$ converges to $\theta_x^e$ almost surely, the distribution $\lim_{t\to\infty} dP(b(X, C_t) = \theta | X = x, E = e)$ evaluates to probability one when $\theta = \theta_x^e$ and is zero otherwise. As a result, the above expressions become

$$\lim_{t\to\infty} dP(Y = y, X = x, b(X, C_t) = \theta) = \int_{e \in \mathcal{E}_{x,\theta}} dP(Y = y | X = x, E = e) dP(X = x, E = e). \qquad (25)$$

where $\mathcal{E}_{x,\theta}$ is the set of all the environments observed conditional on $X = x$ with $\theta_x^e = \theta$. Observe that all the environments in $\mathcal{E}_{x,\theta}$ have the same $dP(Y = 1 | X = x, E = e)$ given by $h^\star(x, \theta)$. Similarly,

$$\lim_{t\to\infty} dP(X = x, b(X, C_t) = \theta) = \int_{e \in \mathcal{E}_{x,\theta}} dP(X = x, E = e). \qquad (26)$$

As a result, we can write

$$\lim_{t\to\infty} dP(Y = 1, X = x, b(X, C_t) = \theta) = h^\star(x, \theta) \int_{e \in \mathcal{E}_{x,\theta}} dP(X = x, E = e).$$

We use this to obtain

$$
\begin{aligned}
\lim_{t\to\infty} dP(Y=1|X=x, b(X,C_t)=\theta) &= \frac{\lim_{t\to\infty} dP(Y=1, X=x, b(X,C_t)=\theta)}{\lim_{t\to\infty} dP(X=x, b(X,C_t)=\theta)} \\
&= \frac{h^\star(x,\theta) \int_{e\in\mathcal{E}_{x,\theta}} dP(X=x, E=e)}{\int_{e\in\mathcal{E}_{x,\theta}} dP(X=x, E=e)} = h^\star(x,\theta).
\end{aligned}
\tag{27}
$$

Therefore,

$$
\lim_{t\to\infty} dP(Y=y|X=x, b(X,C_t)=\theta) = dP(Y=y|X=x, E=e).
\tag{28}
$$

where $e$ is any environment that is in the support of data sampled with $X=x$ and that also satisfies $\theta_x^e = \theta$.

$$
\begin{aligned}
\lim_{t\to\infty} H(Y|X, b(X,C_t)) &= \int_{x,\theta} \lim_{t\to\infty} dP(X=x, b(X,C_t)=\theta) \lim_{t\to\infty} H(Y|X=x, b(X,C_t)=\theta) \\
&= \int_{x,\theta} \Big( \int_{\tilde{e}\in\mathcal{E}_{x,\theta}} dP(X=x, E=\tilde{e}) \Big) \lim_{t\to\infty} H(Y|X=x, b(X,C_t)=\theta)
\end{aligned}
\tag{29}
$$

In the above, we use dominated convergence theorem to swap the limits and integrals (Recall that $dP(X=x, E=e, b(X,C_t)=\theta)$ is bounded say by say $\varsigma$ and the volume of the support of $E$ is bounded say by $\varphi$. As a result, $dP(X=x, b(X,C_t)=\theta)H(Y|X=x, b(X,C_t)=\theta) \leq \varsigma\varphi\log(2)$.). From equation 28, it follows that $\lim_{t\to\infty} H(Y|X=x, b(X,C_t)=\theta) = H(Y|X=x, E=e)$, where $e$ is any environment in $\mathcal{E}_{x,\theta}$. We use this in the above to get

$$
\begin{aligned}
\lim_{t\to\infty} H(Y|X, b(X,C_t)) &= \int_{x,\theta} \Big( \int_{\tilde{e}\in\mathcal{E}_{x,\theta}} dP(X=x, E=\tilde{e}) \Big) H(Y|X=x, E=e) \\
&= \int_{x,\theta} \Big( \int_{\tilde{e}\in\mathcal{E}_{x,\theta}} dP(X=x, E=\tilde{e}) \Big) H(Y|X=x, E=\tilde{e}) \\
&= \int_{x,\tilde{e}} dP(X=x, E=\tilde{e}) H(Y|X=x, E=\tilde{e}) = H(Y|X, E).
\end{aligned}
\tag{30}
$$

We combine the above with equation 21 to obtain $\lim_{t\to\infty} H(Y|X, C_t) = H(Y|X, E)$. Finally, observe that if $I(Y;E|X) > 0 \implies H(Y|X,E) < H(Y|X)$. Since $\lim_{t\to\infty} H(Y|X, C_t) = H(Y|X, E)$, ICRM impoves over ERM that attains a cross-entropy loss of $H(Y|X)$.

This completes the proof.

$\square$

### A.3 PROOF OF THEOREM 2

**Theorem 2** (Partial iid zoom-in). *Suppose the joint distribution $((X_1, \cdots X_t), (Y_1, \ldots, Y_t), E)$ is Markov with respect to a Bayesian network. The query $X$ and the environment $E$ are statistically dependent and form the Markov blanket of $Y$. Then* ICRM *partially zooms-in on the environment risk minimizer, improving over the performance of the global empirical risk minimizer in terms of the cross-entropy loss. Further, the improvement is strictly monotonic in context length $t$.*

*Proof.* Let us consider the setting where the context is of length one. We denote the current query as $X$ with corresponding label $Y$ and environment $E$. The example in the context is $\tilde{X}$ which has corresponding label $\tilde{Y}$ and it shares the same environment $E$. Recall that as part of the context, the learner only sees $\tilde{X}$ and not $\tilde{Y}$. Both $Y$ and $E$ are real-valued scalars and $X$ is a $d$ dimensional vector.

Following the assumption in the theorem, the distribution of $(\tilde{X}, \tilde{Y}, X, Y, E)$ is Markov with respect to a Bayesian network. We first establish that $E$ cannot be a child of any variable in the directed acyclic graph (DAG). The assumption $(X, Y) \perp (\tilde{X}, \tilde{Y})|E$ implies $X \perp \tilde{X}|E$ and $Y \perp \tilde{Y}|E$. Suppose $E$ is a child variable of $Y$. Due to the symmetry, $(X, Y, E)$ and $(\tilde{X}, \tilde{Y}, E)$ follow the same distribution. As a result, $E$ is also a child variable of $\tilde{Y}$, which implies $Y \not\perp \tilde{Y}|E$ (since $E$ is a collider on the path from $Y$ to $\tilde{Y}$). This contradicts $Y \perp \tilde{Y}|E$. Suppose $E$ is a child variable of some component of $X$ say $X^i$. Due to symmetry, $E$ is also a child variable of $\tilde{X}^i$, which implies $X^i \not\perp \tilde{X}^i|E$. This contradicts $X \perp \tilde{X}|E$. Therefore, $E$ cannot be a child of any of the variables in the DAG.

Since both $X$ and $E$ form the Markov blanket of $Y$, there are two possible cases. Either $E$ is directly connected to $Y$ or $E$ is connected to $Y$ through some element of $X$.

In the first case, $E$ can only have an arrow into $Y$ and not the other way around as $E$ is not a child of any other node. Let us consider the setting when $E$ is one of the parents of $Y$ and denote it as $E \rightarrow Y$. Since $X$ ($\tilde{X}$) is on the Markov Blanket of $Y$ ($\tilde{Y}$), we claim that each component of $X$ is either a parent of $Y$ or a child of $Y$. Suppose this was not the case. This implies that there exists a component of $X$ say $X^i$, which is on the Markov Blanket as a parent of $E$. But that would make $E$ a child of $Y$. However, $E$ cannot be a child variable as shown above. As a result, each component of $X$ is either a parent or a child of $Y$. We now consider two subcases.

Let us consider the setting when there exists a child $X^i$ of $Y$. Observe that $\tilde{X}^i$ is a child of $\tilde{Y}$ and it has a path to $E$ and as a result it has a path to $Y$. This path from elements of $\tilde{X}^i$ to $\tilde{Y}$ passes through $E$. This path has no colliders and does not contain any element of $X$ on it (We show this case in Figure 3a). As a result, $Y \not\perp \tilde{X}^i|X$. Thus $I(Y; \tilde{X}|X) > 0$ (use chain rule of mutual information).

Let us consider the other setting when each $X^i$ is a parent of $Y$ (shown in Figure 3b). In this case, $E$ has to have a path to some element of $X$, say $X^j$ as otherwise $E \perp X$, which contradicts the assumption that $E \not\perp X$. Consider the path $\tilde{X}^j$ to $E$ to $Y$. Observe that this path is not blocked. As a result, $I(Y; \tilde{X}|X) > 0$.

Let us consider the other possibility when $Y$ is connected to $E$ through $X$. Here the only way this is possible is if some element of $X$ say $X^i$ is a child of $Y$ and $E$ is a parent of that element (as shown in Figure 3c). Therefore, we know that $\tilde{X}^i$ is connected to $Y$ through $E$ and $X^i$.

Observe that this path from $\tilde{X}^i$ to $Y$ is not blocked as $X^i$ is a collider. Therefore, $I(Y; \tilde{X}|X) > 0$. We showed the result so far assuming that the context length was one. Suppose that the context has $k - 1$ examples denoted as $C_k = [X_1 \cdots, X_{k-1}]$. The chain rule of mutual information tells us $I(Y; C_k|X) = I(Y; X_{k-1}|X) + I(Y; C_{k-1}|X, X_{k-1})$. The proof above already demonstrates that the first term $I(Y; X_{k-1}|X)$ is strictly positive. Since mutual information is non-negative, we can conclude that $I(Y; C_k|X) > 0$.

Next, we want to argue that entropy strictly reduces as context length increases. In other words,

$$H(Y|X, C_k) < H(Y|X, C_{k-1}) \iff I(Y; X_k|X, C_{k-1}) > 0.$$

We want to show $Y \not\perp X_k|(X, C_{k-1})$. In the proof above, we had three cases shown in Figure 3. In each of these cases, we argued that the path from $X_k$ to $Y$ is not blocked. Even if we condition on contexts $C_{k-1}$ this continues to be the case. In the first two cases, the path from $X_k$ to $Y$ is direct and does not contain any element from the conditioning set. In the third case, the direct path involves a collider $X$ from the conditioning set and thus is also not blocked. As a result, $Y \not\perp X_k|(X, C_{k-1})$. This completes the proof.

$\square$

**Remark on the Theorem 2**   It is possible to extend Theorem 2 to the case when only a subset of $X$ and $E$ form the Markov blanket. Observe that the analysis of Case a) and Case c) in Figure 3a, Figure 3c does not change. The analysis of Case b) is more nuanced now. In Case b), we used the fact that $E$ is connected to $X$ that is on the Markov blanket. This need not be the case if only a subset of $X$ is on the Markov blanket. Suppose $X_{\mathrm{MB}}$ denote the set of $X$ that are on the Markov Blanket. If $E$ is connected to any member of $X_{\mathrm{MB}}$, the same analysis as Case b) continues to hold. Consider the

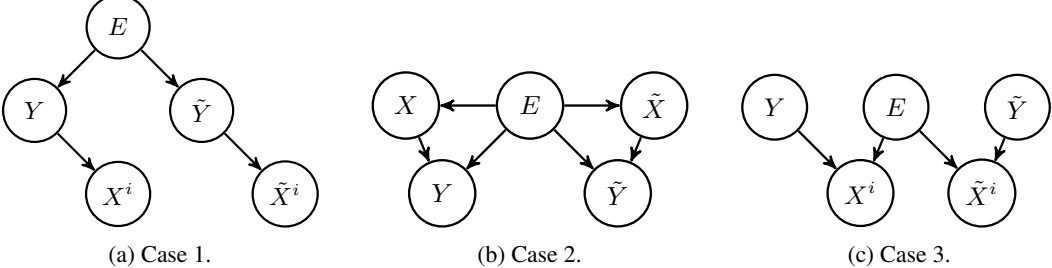

(a) Case 1.  (b) Case 2.  (c) Case 3.

Figure 3: Illustrating the different key cases for Theorem 2.

case when $E$ is connected to some other member of $X$ that is not in $X_{\mathrm{MB}}$. Denote this member as $X^i$. Observe that the same element $\tilde{X}^i$ from $\tilde{X}$ will have a direct path into $Y$ through $E$ that is not blocked. As a result, even in this case conditioning on $\tilde{X}$ helps.

### A.4 PROOF OF THEOREM 3

**Theorem 3** (Full ood zoom-in) *Consider data triplets $(x, y, e)$ generated from $z \sim \mathcal{N}(\mu_e^y, \Sigma_e^y)$ and $x \leftarrow g(z)$, for all environments $e \in \mathcal{E}$, where $g$ is the identity map. There exists an ICL algorithm that in the limit of infinitely long contexts produces Bayes optimal predictions for all the test environments that fall in the Voronoi cells of the training environments.*

*Proof.* The learning algorithm works as follows. For each $e, y$ pair in the training data, define the set of $x's$ as $\mathcal{D}_x^{e,y}$. Maximize the likelihood of $\mathcal{D}_x^{e,y}$ assuming that the underlying distribution is Gaussian. This can be stated as

$$\hat{\mu}_e^y, \hat{\Sigma}_e^y = \underset{\mu_e^y, \Sigma_e^y}{\arg\min} \Big( \sum_{x \in \mathcal{D}_x^{e,y}} \Big[ \|x - \mu_e^y\|_{(\Sigma_e^y)^{-1}}^2 \Big] - \log(\det(\Sigma_e^y)) \Big).$$

The solution to the above are standard sample mean based estimators of means and covariance. Also, use a sample mean based estimator to estimate the probability of each class in environment $e$ and denote it as $\hat{p}_e^y$. Define $\hat{\gamma}_e = [(\hat{p}_e^0, \hat{\mu}_e^0, \hat{\Sigma}_e^0), (\hat{p}_e^1, \hat{\mu}_e^1, \hat{\Sigma}_e^1)]$. The model at test time works as follows.

- We are given samples $\mathcal{D}_x^{e'}$ at test time from some environment $e' \in \mathcal{E}_{te}$. Estimate the parameters of Gaussian mixture model with two mixture components to maximize the likelihood of observing $\mathcal{D}_x^{e'}$. We denote the estimated parameters as $\theta_{e'} = [p_{e'}, \mu_{e'}, \Sigma_{e'}, \tilde{p}_{e'}, \tilde{\mu}_{e'}, \tilde{\Sigma}_{e'}]$. Define a permutation of $\theta_{e'}$ as $\beta_{e'} = [\tilde{p}_{e'}, \tilde{\mu}_{e'}, \tilde{\Sigma}_{e'}, p_{e'}, \mu_{e'}, \Sigma_{e'}]$.

- Find the closest environment to the estimated parameters in the training set.

$$\min_{e \in \mathcal{E}_{tr}} \Big( \min\{\|\theta_{e'} - \hat{\gamma}_e\|, \|\beta_{e'} - \hat{\gamma}_e\|\} \Big) \tag{31}$$

Suppose $\tilde{e}$ is the closest training environment that solves the above. If $\theta_{e'}$ is closer to $\hat{\gamma}_{\tilde{e}}$ than $\beta_{e'}$, then $p_{e'}, \mu_{e'}, \Sigma_{e'}$ correspond to the label 0 and $\tilde{p}_{e'}, \tilde{\mu}_{e'}, \tilde{\Sigma}_{e'}$ correspond to the label 1. For the query $x$, the probability assigned to label 0 is

$$c(x) = \frac{p_{e'} e^{-\|x - \mu_{e'}\|_{(\Sigma_{e'})^{-1}}^2}}{p_{e'} e^{-\|x - \mu_{e'}\|_{(\Sigma_{e'})^{-1}}^2} + \tilde{p}_{e'} e^{-\|x - \tilde{\mu}_{e'}\|_{(\tilde{\Sigma}_{e'})^{-1}}^2}}.$$

If $\beta_{e'}$ is closest to this environment, then $p_{e'}, \mu_{e'}, \Sigma_{e'}$ correspond to the label 1 and $\tilde{p}_{e'}, \tilde{\mu}_{e'}, \tilde{\Sigma}_{e'}$ is the label 0. For the query $x$, the probability assigned to label 0 is $1 - c(x)$.

For the training environments, in the limit of infinitely long contexts the estimated parameters take exact values, i.e., $\hat{\gamma}_e = \gamma_e$, for all $e \in \mathcal{E}_{tr}$.

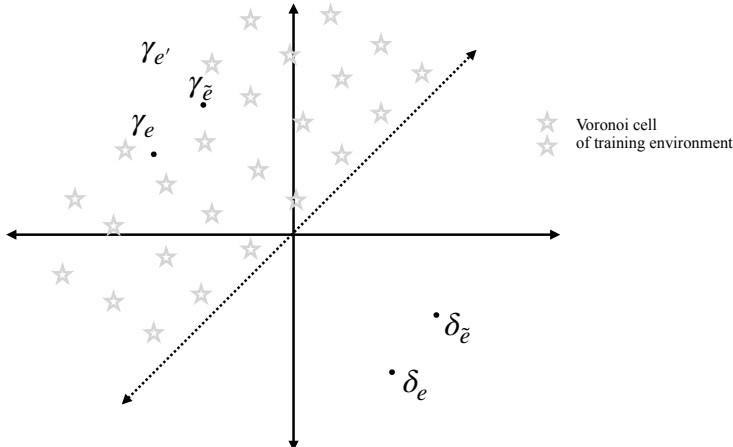

Figure 4: Illustration of Voronoi cells of training environment.

For the test environment, the true set of parameters that generate the data are $\gamma_{e'}$, where $\gamma_{e'} = \left[(p_{e'}^0, \mu_{e'}^0, \Sigma_{e'}^0), (p_{e'}^1, \mu_{e'}^1, \Sigma_{e'}^1)\right]$. Define the permutation of $\gamma_{e'}$ as $\delta_{e'} = \left[(p_{e'}^1, \mu_{e'}^1, \Sigma_{e'}^1), (p_{e'}^0, \mu_{e'}^0, \Sigma_{e'}^0)\right]$.

There can be two types of test environments. One in which the mean and covariance for both classes are identical. The method above assigns a probability of $\frac{1}{2}$ to both the classes, which is the Bayes optimal prediction. Let us consider the latter environments, where the class conditional parameters for $x$ are not the same. In the limit of infinitely long contexts at test time, there are two possible values $\theta_{e'}$ can take, either $\theta_{e'} = \gamma_{e'}$ or $\theta_{e'} = \delta_{e'}$. This follows from identifiability of Gaussian mixtures, Yakowitz and Spragins (1968).

Consider the first case, $\theta_{e'} = \gamma_{e'}$. In this case, the equation 31 becomes

$$\min_{e \in \mathcal{E}_{tr}} \left( \min\{\|\gamma_{e'} - \gamma_e\|, \|\delta_{e'} - \gamma_e\|\} \right).$$

Suppose some environment $\tilde{e}$ solves the above optimization. Following the assumption in we know that $\gamma_{e'}$ falls in the Voronoi region of some $\gamma_{\tilde{e}}$ and thus $\gamma_{e'}$ is closer to $\gamma_{\tilde{e}}$ than $\delta_{\tilde{e}}$ (see Figure 4). As a result, $p_{e'}^0, \mu_{e'}^0, \Sigma_{e'}^0$ is associated with class 0, which is actually correct and thus the final predictor would match the Bayes optimal predictor for the test environment. In the second case, $\theta_{e'} = \delta_{e'}$. Therefore, $\beta_{e'} = \gamma_{e'}$ and $p_{e'}^1, \mu_{e'}^1, \Sigma_{e'}^1$ would be correctly associated with class one thus leading to Bayes optimal predictions. This completes the argument we set out to prove.

We now briefly explain how the method fails if test parameter is outside the Voronoi cell of training parameters. Suppose $\theta_{e'} = \gamma_{e'}$ but $\gamma_{e'}$ is in Voronoi region of some $\delta_e$. In this case, $\beta_{e'}$ would be closest to $\gamma_e$ and $p_{e'}^0, \mu_{e'}^0, \Sigma_{e'}^0$ would be incorrectly associated with class 1. This shows that beyond the Voronoi region the proposed algorithm fails.

$\square$

## A.5 EXTENSION OF THEOREM 3

In the previous theorem, we assumed that $g$ is identity. We now describe how the result can be extended to general non-linear mixing maps $g$. For this result, we leverage the theoretical results from identifiable variational autoencoders (i-VAE) (Khemakhem et al., 2020).

**A short review of identifiable variational autoencoders (Khemakhem et al., 2020)** We are provided with observations $x$'s that are generated from a latent variable $z$ using an injective map $g$,

where $x \leftarrow g(z)$. The theory of i-VAE provides with a method and the conditions under which the underlying true latent variables $z$ can be identified up to permutation and scaling. In i-VAEs, it is assumed that along with each sample $x$, we are provided with auxiliary information, which they term as $u$. For our results, auxiliary information is available to us in the form of the environment index and the label of the data point. In the theory of i-VAE, the distribution of the latent variables are assumed to follow a conditionally factorial exponential distribution stated as follows.

$$p_{T,\lambda}(z|u) = \prod_i \frac{Q_i(z_i)}{M_i(u)} \exp\left[ \sum_{j=1}^{k} T_{i,j}(z_i) \lambda_{i,j}(u) \right] \tag{32}$$

where $T_i = (T_{i,1}, \cdots, T_{i,k})$ are the sufficient statistics, $\lambda_i(u) = (\lambda_{i,1}(u), \cdots, \lambda_{i,k}(u))$ are the parameters of the distribution that vary with $u$, $Q_i$ is a base measure and $M_i$ is a normalizing constant. We concatenate $T_i$'s and $\lambda_i's$ across $d$ latent dimensions to make construct $dk$ dimensional vectors denoted as $\lambda(u)$ and $T(z)$. Thus the data generation process is summarized as

$$\begin{aligned} z &\sim p_{T,\lambda}(\cdot|u), \\ x &\leftarrow g(z), \end{aligned} \tag{33}$$

where $g, T, \lambda$ are the parameters. We now revisit the data generation process that we consider and explain how it falls under the umbrella of the data generation processes considered in i-VAE. For all $e \in \mathcal{E}$,

$$\begin{aligned} z|y, e &\sim \mathcal{N}(\mu_e^y, \Sigma_e^y), \\ x &\leftarrow g(z), \end{aligned} \tag{34}$$

where the latent variables $z$ are sampled conditional on the label $y$ and environment $e$ from a Normal distribution whose mean and covariance depend on both $y, e$. Define $\mathcal{X}$ as the image of $g$, i.e., $\mathcal{X} = g(\mathbb{R}^d)$. We further assume that the covariance matrix has a diagonal structure as stated below.

**Assumption 1.** *Each $\Sigma_e^y$ is a diagonal matrix.*

Since $\Sigma_e^y$ is a diagonal matrix, we denote the $i^{th}$ diagonal element as $(\sigma_e^y(i))^2$. Similarly, the $i^{th}$ component of $\mu_e^y$ is denoted as $\mu_e^y(i)$. Observe that the distribution of $z$ conditional on $y, e$ belongs to the family conditionally factorial exponential distributions studied in i-VAE (Khemakhem et al., 2020). If we substitute $Q_i(z_i) = \frac{1}{\sqrt{2\pi}}$, $M_i(y,e) = e^{\left((\mu_e^y(i))^2/(\sigma_e^y(i))^2\right)}$, $\lambda_{i,1}(y,e) = \frac{2\mu_e^y(i)}{(\sigma_e^y(i))^2}$, $\lambda_{i,2}(y,e) = -\frac{1}{(\sigma_e^y(i))^2}$, $T_{i,1}(z) = z$ and $T_{i,2}(z) = z^2$, then we obtain the distribution of $z$ described by equation 34.

**Definition 1.** *We define an equivalence relation between sets of parameters of the model as follows.*

$$(g, T, \lambda) \sim (\tilde{g}, \tilde{T}, \tilde{\lambda}) \iff \exists A, c \mid T(g^{-1}(x)) = A\tilde{T}(\tilde{g}^{-1}(x)) + c, \forall x \in \mathcal{X}. \tag{35}$$

*If $A$ is invertible, then we denote the relation by $\sim_A$. If $A$ is a block permutation matrix, then we denote it by $\sim_P$.*

We now state some key results from (Khemakhem et al., 2020).

**Theorem 5.** *Assume that the data is sampled from the data generation in equation 33 according to with parameters $(g, T, \lambda)$. Assume the following holds*

- *The mixing function $g$ is injective*

- *The sufficient statistics $T_{i,j}$ are differentiable almost everywhere, and $(T_{i,j})_{1 \leq j \leq k}$ are linearly independent on any subset of $\mathcal{X}$ of measure greater than zero.*

- *There exists $dk + 1$ distinct points $u_0, \cdots, u_{dk}$ such that the matrix*

$$L = (\lambda(u_1) - \lambda(u_0), \cdots, \lambda(u_{dk}) - \lambda(u_0))$$

*of size $dk \times dk$ is invertible.*

*then the parameters $(g, T, \lambda)$ are $\sim_A$ identifiable.*

**Theorem 6.** *Assume the hypotheses of the Theorem 5 holds, and $k \geq 2$. Further assume:*

- *The sufficient statistics $T_{i,j}$ are twice differentiable.*

- *The mixing function $g$ is $C^2$-diffeomorphism.*

*then the parameters $(g, T, \lambda)$ are $\sim_P$ identifiable.*

We can leverage the above two theorems (Theorem 5, Theorem 6 and Theorem 4 from Lachapelle et al. (2022)) and arrive at the following corollary for the Gaussian data generation process from equation 34.

**Theorem 7.** *If the data generation process follows equation 34, where $g$ is a $C^2$-diffeomorphism. Suppose there exist $2d + 1$ points $u^0 = (y_0, e_0), \cdots, u^{2d} = (y_{2d}, e_{2d})$ in the support of $(y, e)$ observed in training distribution such that*

$$(\lambda(u_1) - \lambda(u_0), \cdots, \lambda(u_{2d}) - \lambda(u_0))$$

*is invertible. If $p_{g,T,\lambda}(\cdot|y, e) = p_{\tilde{g}, \tilde{T}, \tilde{\lambda}}(\cdot|y, e)$ for all $y, e$ in the support of $(y, e)$ in the training distribution, then $\tilde{z} = \Lambda\Pi z + r$, where $\tilde{z} = \tilde{g}^{-1}(x)$ and $z = g^{-1}(x)$.*

*Proof.* We equate the probability of observations $x$ under two models $g, T, \lambda$ and $\tilde{g}, \tilde{T}, \tilde{\lambda}$ for each $y, e$. Consider a $z \sim p_{T,\lambda}(\cdot|y, e)$ and the corresponding $x = g(z)$. These $x$'s follow $p_{\tilde{g}, \tilde{T}, \tilde{\lambda}}(\cdot|y, e)$ since $p_{g,T,\lambda}(\cdot|y, e) = p_{\tilde{g}, \tilde{T}, \tilde{\lambda}}(\cdot|y, e)$. Define $\tilde{z} = \tilde{g}^{-1}(x)$ and these $\tilde{z}$ follow $p_{\tilde{T}, \tilde{\lambda}}(\cdot|y, e)$. We can write $\tilde{z} = a(z)$, where $a = \tilde{g}^{-1} \circ g$.

Observe $p_z(z|y, e) = p_{\tilde{z}}(a(z)|y, e)\det(Da(z))$ and

$$\log p_z\big(z|y_k, e_k\big) = \log\big(p_{\tilde{z}}(a(z)|y_k, e_k)\big) + \log\det(Da(z)),$$
$$\log p_z\big(z|y_0, e_0\big) = \log\big(p_{\tilde{z}}(a(z)|y_0, e_0)\big) + \log\det(Da(z)), \quad (36)$$
$$\log p_z\big(z|y_k, e_k\big) - \log\big(p_z(z|y_0, e_0)\big) = \log\big(p_{\tilde{z}}(a(z)|y_k, e_k)\big) - \log\big(p_{\hat{z}}(a(z)|y_0, e_0)\big).$$

Substituting the exponential form we obtain that
$$T(z)^\top[\lambda(y_k, e_k) - \lambda(y_0, e_0))] = T(\tilde{z})^\top[\tilde{\lambda}(y_k, e_k) - \tilde{\lambda}(y_0, e_0))]$$

If we use sufficient variability conditions, we obtain $T(z) = AT(\tilde{z}) + c$. We now use the fact that sufficient statistics $T(z) = (z, z^2)$ are minimal to conclude that
$$T(z) = AT(\tilde{z}) + c$$

where $A$ is invertible. In the above, we use the line of reasoning used in in the proof of Theorem 4 in (Lachapelle et al., 2022).

After this point, we leverage Theorem 6 to conclude that

$$T_i(z_i) = AT_j(\tilde{z}_j) + c.$$

We can expand the above to write

$$\begin{bmatrix} \tilde{z}_j \\ \tilde{z}_j^2 \end{bmatrix} = D \begin{bmatrix} z_i \\ z_i^2 \end{bmatrix} + e.$$

Note that the above relationship holds for all $z \in \mathcal{Z}$. If $\tilde{z}_j$ depends on $z_i^2$, then $\tilde{z}_j^2$ would be a degree four polynomial in $z_i$ and it would be equated to a degree 2 polynomial $z_i$ stated in the RHS. This cannot be true for all $z_i$ in the support. As a result, $\tilde{z}_j$ is a scalar multiple of $z_i$. Since for every $i$ there is such a $j$, it follows that $\tilde{z} = \Lambda\Pi z + r$.

$\square$

**Theorem 8.** *(**Zoom-in [ood]**) Consider the data generation process in equation 34. We make a few additional assumptions on the data generation stated below.*

- *Each $\Sigma_e^y$ is a diagonal matrix*

- *There exist $2d + 1$ points $u^0 = (y_0, e_0), \cdots, u^{2d} = (y_{2d}, e_{2d})$ in the support of $(y, e)$ observed in training distribution such that*

$$(\lambda(u_1) - \lambda(u_0), \cdots, \lambda(u_{2d}) - \lambda(u_0))$$

  *is invertible.*

- *$g$ is a $C^2$-diffeomorphism.*

*Under the above assumptions, we can guarantee that there exists an in-context learning algorithm that in the limit of infinitely long contexts generates Bayes optimal predictions for all the test environments that fall in Voronoi cells of training parameters weighted by a certain vector.*

*Proof.* The training proceeds as follows. Train an autoencoder on training data under the constraint that the output of the encoder follow a Gaussian distribution with independent components conditional on each $y, e$. This is stated as the following minimization.

$$\hat{g}, \hat{f}, \hat{\mu}_e^y, \hat{\Sigma}_e^y = \arg \min_{\tilde{g}, \tilde{f}, \{\mu_e^y, \Sigma_e^y\}} \mathbb{E}[\|(\tilde{g} \circ \tilde{f}(x) - x)\|^2] + \alpha \sum_{y,e} \mathsf{KL}\left(p_{\tilde{z}}(\cdot | y, e) \, \| \, \mathcal{N}(\mu_e^y, \Sigma_e^y)\right) \quad (37)$$

where $\tilde{z} = \tilde{f}(x)$, $p_{\tilde{z}}(\cdot | y, e)$ is the distribution of $\tilde{z}$. The first term is standard reconstruction loss and the second term is the KL divergence between distribution of $\tilde{z}$ and a Normal distribution with independent components. Also, estimate the class probabilities for each environment and denote them as $\hat{p}_e^y$. Similar to the proof of Theorem 3 define $\hat{\gamma}_e = [(\hat{p}_e^y, \hat{\mu}_e^y, \hat{\Sigma}_e^y)_{y \in \{0,1\}}]$

The model at test time works as follows. We first use the trained encoder $\hat{f}$ and generate $\tilde{z}$ for test time inputs. After this the model operates in exactly the same way on $\tilde{z}'s$ as in the proof of Theorem 3. Basically the output of encoder takes place of raw $x$'s in the procedure described in proof of Theorem 3.

The assumptions in this theorem along with following i) $\tilde{z}$ follows a Gaussian distribution with independent components, ii) $g(\tilde{z})$ follows distribution of $x$ conditional on $y, e$ for each $y, e$, implies we can use the previous result in Theorem 7 to conclude that $\tilde{z} = \Lambda \Pi z + r$. Observe that $\tilde{z}$ also follows a Gaussian distribution with independent components conditional on each $y, e$. In the limit of infinitely long contexts, $\hat{\gamma}_e$ is equal to scaled means of original training environments and covariances also scaled componentwise according to the transform $\Lambda \Pi$. We can now apply the previous Theorem 3 on $\tilde{z}'s$ as follows. If the parameters of the test environment are in the Voronoi cell of the train distribution of $\tilde{z}'s$, then the procedure described above continues to generate Bayes optimal predictions in those environments.

□

## A.6 COMPARING ICRM AND ERM UNDER THE LENS OF INVARIANCE

The label $y$ is related to $x^1$ and mean of $x^2$ in environment $e$ as follows.

$$y \leftarrow \alpha x^1 + \beta \mu_e^2 + \varepsilon \quad (38)$$

ERM learns a linear model on two dimensional feature vector $x = (x^1, x^2)$. The closed form solution for linear regression is $\Lambda^{-1}\rho$, where $\Lambda = \mathbb{E}[XX^\top]$, which is assumed to be invertible, and $\rho = \mathbb{E}[XY]$. The covariance matrix of $X$ is defined as $\Sigma = \begin{bmatrix} \sigma_1^2 & \sigma_{12} \\ \sigma_{12} & \sigma_2^2 \end{bmatrix}$.

**Proposition 2.** *Let $\mathbb{E}[X^1 | E = e] = 0$ for all $e \in \mathcal{E}$. If $\Sigma$ is invertible, $\beta \neq 0$, $\sigma_{12} \neq 0$, $\mu_e^2 \neq 0$ for some $e \in \mathcal{E}_{tr}$, then the coefficient estimated by ERM for $x^1$ is not the same as the invariant coefficient $\alpha$.*

*Proof.* We compute $\rho$ first.

$$\rho = \mathbb{E}[XY] = \begin{bmatrix} \alpha\mathbb{E}[(X^1)^2] + \beta\mathbb{E}[\mu_E^2 X^1] \\ \alpha\mathbb{E}[X^1 X^2] + \beta[\mu_E^2 X^2] \end{bmatrix}$$
$$= \alpha \begin{bmatrix} \sigma_1^2 \\ \sigma_{12} + \frac{\beta}{\alpha}\delta \end{bmatrix}, \tag{39}$$

where $\delta = \mathbb{E}[(\mu_E^2)^2]$.

Next, we compute $\Lambda$.

$$\Lambda = \mathbb{E}[XX^\top] = \begin{bmatrix} \sigma_1^2 & \sigma_{12} \\ \sigma_{12} & \sigma_2^2 + \delta \end{bmatrix}. \tag{40}$$

The solution to ERM is

$$\begin{bmatrix} \alpha' \\ \beta' \end{bmatrix} = \frac{\alpha}{(\sigma_2^2 + \delta)\sigma_1^2 - \sigma_{12}^2} \begin{bmatrix} \sigma_2^2 + \delta & -\sigma_{12} \\ -\sigma_{12} & \sigma_1^2 \end{bmatrix} \begin{bmatrix} \sigma_1^2 \\ \sigma_{12} + \frac{\beta}{\alpha}\delta \end{bmatrix}. \tag{41}$$

Simplifying the above, we obtain the coefficient for $x_1$ to be

$$\alpha' = \alpha - \frac{\sigma_{12}\beta\mathbb{E}[(\mu_E^2)^2]}{\sigma_1^2(\sigma_2^2 + \mathbb{E}[(\mu_E^2)^2]) - \sigma_{12}^2}. \tag{42}$$

Owing to the assumptions, $\beta \neq 0, \sigma_{12} \neq 0$ and $\mu_e^2$ for some $e$ we obtain that the second term in the above is not zero. As a result, the estimate computed by ERM for $\alpha$ is biased. $\qquad\square$

**Proposition 3.** *Let* $\mathbb{E}[X_1|E = e] = 0$ *for all* $e \in \mathcal{E}$. *If* $\Sigma$ *is invertible,* $\beta \neq 0$, $\sigma_{12} \neq 0$, $\mu_e^2 \neq 0$. *The error of ERM in test environment increases in* $\sigma_1^2$

*Proof.* The error of ERM is given as

$$\mathbb{E}[(\alpha X^1 + \beta\mu_e^2 - \alpha' X^1 - \beta' X^2)^2] + \sigma_\varepsilon^2$$
$$= (\alpha - \alpha')^2\sigma_1^2 + \beta^2\mathbb{E}[(\mu_E^2)^2] + (\beta')^2\mathbb{E}[(X^2)^2] - 2\beta\beta'\mathbb{E}[(\mu_E^2)^2] - 2(\alpha - \alpha')\beta\sigma_{12} + \sigma_\varepsilon^2, \tag{43}$$

where $\sigma_\varepsilon^2$ is the variance of the noise variable $\varepsilon$. If we take the derivative of the above error w.r.t $\sigma_1^2$, we obtain $(\alpha - \alpha')^2$, which is positive. This completes the proof.

$$\square$$

ICRM learns a linear model on $(x^1, x^2, \mu_e^1, \mu_e^2)$. We study two settings to analyze the error of ICRM at test time. If at test time, the model has seen sufficiently long contexts, then it knows the means corresponding to $x^1$ and $x^2$ and the model achieves the test error of $\sigma_\varepsilon^2$. On the other hand, if the context is empty, then also note that the expected error of the model is $\beta^2\|\mu_{e'}^2\|^2$ (assuming the model uses a default value of zero for the mean in the absence of any context), where $\mu_{e'}^2$ is the mean of $x^2$ in environment $e'$. Since the error of ICRM in the absence of any context is independent of variance of $x^1$, the error of ERM can be much worse than that of ICRM in this setting as well.

**Extending the above example beyond linear settings.** Let us consider a more general setting.

$$y = u(x^1, \mu_e^2) + \varepsilon,$$
$$x^2 = v(\mu_e^2, \vartheta), \tag{44}$$

where $u(\cdot)$ and $v(\cdot)$ are maps (potentially non-linear), $\varepsilon$ and $\vartheta$ are independent zero mean noise variables. Following the same line of thought as the above example. ICRM learns a non-linear model on $(x_1, x_2, \mu_1^e, \mu_2^e)$ and learns $\mathbb{E}[Y|x^1, x^2, \mu_e^1, \mu_e^2]$. From equation 44, it follows that

$$Y \perp (X^2, \mu_E^1)|(X^1, \mu_E^2) \implies \mathbb{E}[Y|x^1, x^2, \mu_e^1, \mu_e^2] = \mathbb{E}[Y|x^1, \mu_e^2] = u(x^1, \mu_e^2).$$

From the above it follows that ICRM learns $u(x^1, \mu_e^2)$. In comparison, consider standard ERM learns a non-linear model on $(x^1, x^2)$. Consider the DAG corresponding to setting equation 44. We assume that the joint distribution described in equation 44 is Markov w.r.t to the following DAG $X^1 \to Y \leftarrow \mu_E^2 \to X^2$. As a result, $Y \not\perp X^2 | X^1$. This follows from the fact there is a path $Y$ to $X^2$ through $\mu_E^2$ and is not blocked by $X^1$. From $Y \not\perp X^2 | X^1$ it follows that ERM learns a predictor that relies on both $x^1$ and $x^2$. Therefore, ICRM learns the right invariant model and does not rely on $x^2$ and ERM relies on spurious feature $x^2$.

## A.7 ILLUSTRATION OF FAILURE OF EXISTING MTL METHODS

In this section, we provide a simple example to show the failure mode of marginal transfer learning (MTL) methods that are based on averaging $\frac{1}{|c|} \sum_{x_i \in c} \Phi(\cdot)$ to summarize information about the environment. These methods can be summarized to take the following form:

$$f\left(\frac{1}{|c|} \sum_{x_i \in c} \Phi(x_i), x\right). \tag{45}$$

We are only going to consider maps $\Phi$ that are differentiable.

**Example.**    Suppose we want to learn the following function

$$w(x, c) = \frac{1}{|c|} \sum_{x_i \in C} I(x < x_i), \tag{46}$$

where $x_i$ is the $i^{th}$ input in the context and $x$ is the current query and $I(\cdot)$ is indicator function that takes a value of one if the argument inside is true and zero otherwise. We claim that if $f\left(\frac{1}{|c|} \sum_{x_i \in c} \Phi(x_i), x\right) = w(x, c)$ for all $x \in \mathbb{R}, c \in \mathbb{R}^{|c|}$, then the output dimension of $\Phi$ grows in context length $|c|$. Suppose this was not the case. If $\Phi's$ output dimension is smaller than $|c|$, then $\Phi$ cannot be a differentiable bijection. As a result, there exists two contexts $c$ and $c'$ of same length for which $\sum_{x_i \in c} \Phi(x_i) = \sum_{x_i \in c'} \Phi(x_i)$. We argue that there exists an $x$ such that $w(x, c) \neq w(x, c')$. This would lead to a contradiction as $f\left(\frac{1}{|c|} \sum_{x_i \in c} \Phi(x_i), x\right) = w(x, c)$ for all $x, c$. Without loss of generality, suppose that the smallest value of context $c$ is smaller than that in context $c'$. If $x$ is larger than smallest value of $c$ but lesser than smallest value of $c'$, then $w(x, c') = 1$ on the other hand $w(x, c) \leq 1 - \frac{1}{|c|}$.

We can translate the insight from the above example into more general settings. Consider any map $w(x, c)$, that satisfies the following property. For no two distinct contexts $c$ and $c'$, $w(x, c) = w(x, c')$ for all $x \in \mathbb{R}$. Maps of the form $f\left(\frac{1}{|c|} \sum_{x_i \in c} \Phi(x_i), x\right)$ can only approximate such $w's$ provided dimension of $\Phi$ grows in length of $c$.

We explain how the above example can be described by attention-based architectures with much fewer parameters. First take the current query $x$ and transform it through a linear map $\tilde{x} = \begin{bmatrix} x \\ 1 \end{bmatrix}$ and transform the past context values through a linear map as well to obtain a transform for $x_i$ to $\tilde{x}_i = \begin{bmatrix} 1 \\ x_i \end{bmatrix}$. We set the Query $Q$ and Key $K$ matrices such that $Q^\top K = \begin{bmatrix} -1 & 0 \\ 0 & 1 \end{bmatrix}$ and thus $\tilde{x}^\top Q^\top K \tilde{x}_i = (-x + x_i)$. Instead of softmax, if we pass the output through a sigmoid, we obtain $\sigma(\tau \tilde{x}^\top Q^\top K \tilde{x}_i) = \frac{1}{1 + e^{-\tau(x_i - x)}}$. If $\tau$ is sufficiently large, then this approximates $I(x < x_i)$. Therefore, one layer linear attention with sigmoid and sufficiently high $\tau$ achieves the target, i.e., $\sum_{x_i \in c} \sigma(\tau \tilde{x}^\top Q^\top K \tilde{x}_i) \approx \sum_{x_i \in C} I(x < x_i)$.

## B   RELATED WORK

**A brief tour of domain generalization.**   Muandet et al. (2013) developed kernel methods to learn transformations such that the distance between the feature distributions across domains is minimized and the information between the features and the target labels is preserved. The pioneering work of Ganin et al. (2016) proposes a method inspired from generative adversarial networks to learn feature representations that are similar across domains. Sun and Saenko (2016) developed a method based on a natural strategy to match the means and covariances of feature representations across domains. Li et al. (2018) went a step further to enforce invariance on the distribution of representations conditional on the labels. In a parallel line of work, led by Peters et al. (2016); Rojas-Carulla et al. (2018); Arjovsky et al. (2019), the proposals sought to learn representations such that the distribution of labels conditional on the representation are invariant across domains. These works were followed by several interesting proposals to enforce invariance – (Teney et al., 2020; Krueger et al., 2020; Ahuja et al., 2020; Jin et al., 2020; Chang et al., 2020; Mahajan et al., 2020; Koyama and Yamaguchi, 2020; Müller et al., 2020; Parascandolo et al., 2021; Robey et al., 2021; Wald et al., 2021; Chen et al., 2022; Wang et al., 2022; Zhang et al., 2023; Eastwood et al., 2022; Rame et al., 2022; Veitch et al., 2021; Makar et al., 2022; Wald et al., 2022; Salaudeen and Koyejo, 2022; Eastwood et al., 2023) – which is an incomplete representative list. See Shen et al. (2021) for a more comprehensive survey of these works. Most of the above works have focused on learning features that enable better generalization. Recently there been an intriguing line of work from Kirichenko et al. (2022); Izmailov et al. (2022) that shifts the focus from feature learning to last layer retraining. These works show that under certain conditions (e.g., avaiability of some data that does not carry spurious correlations) one can carry out last layer retraining and achieve significant out-of-distribution performance improvements.

In the main body of the paper, we already discussed the other prominent line of work in domain generalization on marginal transfer learning, where the focus is to leverage the distributional features and learn environment specific relationships. This line of work was started by the notable work of Blanchard et al. (2011) and has been followed up by several important proposals such as Zhang et al. (2020); Bao and Karaletsos (2023).

**Context-supported prediction frameworks.**   Existing works have exploited contextual information to develop a variety of prediction frameworks. The works on neural processes and conditional neural processes (Garnelo et al., 2018b;a) combined the uncertainty estimation capabilities of Gaussian processes with function aprpoximation capabilities of neural networks and showed promising results on meta-learning. These works were later improved through transformer based archictectures in attentive neural processes (Kim et al., 2019; Nguyen and Grover, 2022). Context-based architectures have also been used to study in-context learning in (Garg et al., 2022; Akyürek et al., 2022; Von Oswald et al., 2023). These works use labeled data in the context to enable adaptation. In contrast, our work adheres to the constraints of domain generalization and only leverages unlabeled data.

## C   SUPPLEMENTARY EXPERIMENTAL DETAILS AND ASSETS DISCLOSURE

### C.1   ASSETS

We do not introduce new data in the course of this work. Instead, we use publicly available widely used image datasets for the purposes of benchmarking and comparison.

### C.2   HARDWARE AND SETUP

Each experiment was performed on 8 NVIDIA Tesla V100 GPUs with 32GB accelerator RAM for a single training run. The CPUs used were Intel Xeon E5-2698 v4 processors with 20 cores and 384GB RAM. All experiments use the PyTorch deep-learning framework.

### C.3   DATASETS

#### C.3.1   FEDERATED EXTENDED MNIST (FEMNIST)

Building on the Extended MNIST (EMNIST) dataset, which includes images of handwritten uppercase and lowercase alphabets along with digits, FEMNIST (Zhang et al., 2020) enriches

this data by attributing each data point to its originating writer. This extension associates each 28×28-sized image in the dataset to one of the 62 classes. In our setup, each writer serves as a distinct environment. We evaluate the performance of each method based on both worst-case and average accuracy across a set of 35 test users, who are distinct from the 262 training users and 50 validation users. Unlabelled data from an environment in this dataset could provide cues about the writing style of the user and disambiguate data points.

### C.3.2 ROTATED MNIST

We employ a customized version of the MNIST dataset as in Zhang et al. (2020). The dataset contains images rotated in increments of 10 degrees, ranging from 0 to 130 degrees. Each degree of rotation constitutes a separate environment, effectively acting as a distinct value. The training set for the two most extreme rotations, 120 and 130 degrees, contains only 108 data points each. For rotations between 90 and 110 degrees, each environment includes 324 data points. The total training set comprises 32,292 points. For evaluation, test images are generated from the MNIST test set, and are duplicated for each environment. Performance metrics include both worst-case and average accuracy across these testing domains. Analogous to FEMNIST, unlabeled samples from an environment within this dataset can assist in distinguishing images that may seem similar due to their rotated orientations.

### C.3.3 WILDS CAMELYON17

We use the Camelyon17 dataset, part of the WILDS benchmark (Koh et al., 2021), which features image patches derived from whole-slide lymph node sections of patients with potential metastatic breast cancer. Each patch is labeled to indicate the presence or absence of a tumor. In our experimental design, each participating hospital is treated as a distinct environment. The dataset is partitioned in alignment with the official WILDS configuration: three hospitals contribute to the training set, a fourth is designated for validation, and the remaining hospital's data is used for testing.

### C.3.4 CIFAR10-C AND TINY IMAGENET-C

Adapting the methodology from Hendrycks and Dietterich (2019), we introduce 56 distinct distortions to the training set, treating each as a separate environment. For evaluation, we use a non-overlapping set of 22 test distortions, largely differing in nature from those used in training. For Tiny ImageNet-C, each 64×64-sized distorted image is associated with one of the 200 classes in the dataset. The same set of corruptions are employed to augment the CIFAR10 dataset, resulting in $32 \times 32$-sized images for the CIFAR10-C dataset.This setup permits an investigation into whether exposure to distortions during training equips the model to better manage novel distortions during testing. We assess performance through both worst-case and average accuracies across these test distortions.

### C.3.5 IMAGENET R

ImageNet-R comprises a diverse range of artistic and creative content, encompassing art, cartoons, deviant art, graffiti, embroidery, graphics, origami, paintings, patterns, plastic objects, plush objects, sculptures, sketches, tattoos, toys, and video game interpretations of ImageNet classes. This dataset consists of renditions for 200 ImageNet classes, totaling 30,000 images. For our experiments, we utilize images from categories of cartoons, paintings, stickers, graphics, sculptures, sketches, tattoos, toys, and video games for training. Validation is conducted using images from embroidery, miscellaneous, and graffiti categories, while the test environments incorporate images from art, deviant art, and origami categories. This segregation introduces extreme real-world distribution shifts, where the data features during testing differ significantly from those observed during training.

## C.4 EXPERIMENTAL PROTOCOLS

To ensure a fair comparison across different algorithms for each dataset, we use a standardized neural network backbone. The details for these architectures are provided in Table 4 and Table 5. We use the ConvNet architecture as outlined in Zhang et al. (2020). '

For ICRM, the same backbone is used to featurize the input, which is then processed by the decoder-only Transformer (Vaswani et al., 2017) architecture from the GPT-2 Transformer family (Radford et al., 2019). Our model is standardized to have 12 layers, 4 attention heads, and a 128-dimensional embedding space across all datasets. Linear layers are employed to map both the input sequence to the transformer's latent embedding and the model's predicted output vector to the output label. For training ICRM on larger datasets like ImageNet R, CIFAR10-C, WILDS Camelyon17 and Tiny ImageNet-C, we start with a ResNet50 model pre-trained on ImageNet (as shown in Table 4) and freeze all batch normalization layers before fine-tuning.

We adopt the same Context Network as used in ARM, specifically retaining their choice of output channels – one for smaller datasets like FEMNIST and Rotated MNIST, and three for the others.

For TENT, all reported metrics are based on its episodic version, where the model is reset to its trained state after processing each batch. This ensures a fair comparison with other methods. Additionally, during testing, the model's parameters are updated for 10 steps using stochastic gradient descent by minimization test entropy across all datasets.

Table 4: Network architectures for each dataset.

| Dataset | Architecture | |
| --- | --- | --- |
| | ICRM | Others |
| FEMNIST Rotated MNIST | ConvNet + GPT2 Transformer | ConvNet |
| CIFAR10-C Camelyon17 Tiny ImageNet-C ImageNet R | ResNet-50 + GPT2 Transformer | ResNet-50 |

Table 5: ConvNet architecture for (Zhang et al., 2020). We use $2\times2$ kernels and "same" padding.

| # | Layer |
| --- | --- |
| 1 | Conv2D (in=$d$, out=128) |
| 2 | BatchNorm2d (dim=129) |
| 3 | ReLU |
| 4 | Max Pooling (2) |
| 5 | Conv2D (in=128, out=128) |
| 6 | BatchNorm2d (dim=128) |
| 7 | ReLU |
| 8 | Max Pooling (2) |
| 9 | Global average-pooling |

We list all hyperparameters, their default settings, and search boundaries for random sweeps in Table 6. The maximum context length, or support, is fixed at 100 for all algorithms. All models are optimized using the Adam optimizer (Kingma and Ba, 2014). To ensure a fair comparison, we perform a random search of 5 trials across the hyperparameter range (refer to Table 6) for each algorithm. The model with the highest validation set accuracy is selected for each run. We then report the average of this number across three independent runs of the entire sweep, and its corresponding standard error.

Table 6: Hyperparameters, their default values and distributions for random search.

| Condition | Parameter | Default value | Random distribution |
| --- | --- | --- | --- |
| ResNet | learning rate | 0.0001 | $10^{\text{Uniform}(-5,-3.5)}$ |
| | weight decay | 0 | $10^{\text{Uniform}(-6,-2)}$ |
| not ResNet | learning rate | 0.0001 | $10^{\text{Uniform}(-4.5,-2.5)}$ |
| | weight decay | 0 | $10^{\text{Uniform}(-6,-2)}$ |

# D ADDITIONAL EXPERIMENTS

## D.1 ADAPTATION CURVES OF VARIOUS ALGORITHMS

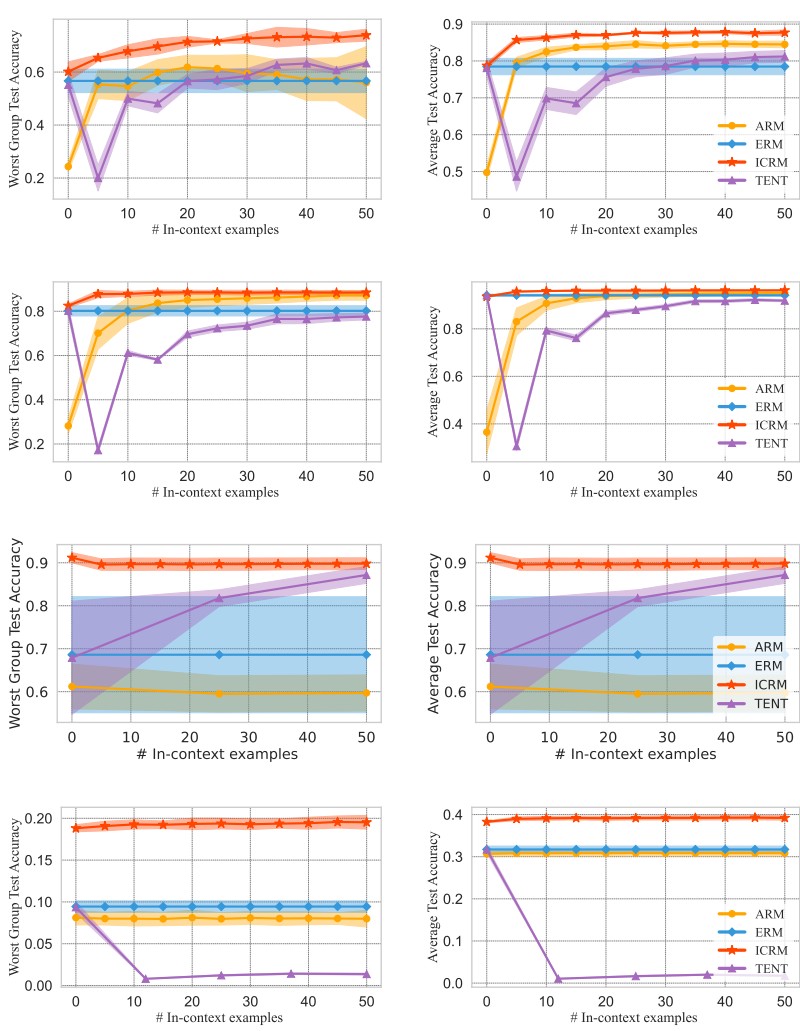

Figure 5: Accuracy adaptation curves for worst accuracy (left) and average accuracy (right) across the test environment as a function of increasing count of context samples. Showing results in order for FEMNIST(top), RotatedMNIST, WILDS Camelyon17 and Tiny ImageNet-C(bottom). The average and worst-case accuracy plots for WILDS Camelyon17 are identical since the dataset contains only a single test environment.

## D.2 DOMAIN GENERALIZATION ACCURACIES PER ALGORITHM AND DATASET

### D.2.1 ADAPTATION TO DISTRIBUTION SHIFT

In our experiments, we compare ICRM against marginal transfer methods such as Adaptive Risk Minimization (Zhang et al., 2020, ARM), test-time adaptation proposals such as TENT (Wang et al., 2020) and Empirical Risk Minimization (Vapnik, 1998, ERM). We also include comparisons with six additional baselines, including approaches that follow the invariance-based paradigm like Fish (Shi et al., 2021) and IB-IRM (Ahuja et al., 2021), alongside those that use contextual information differently, such as BN Adapt (Schneider et al., 2020; Li et al., 2016b) and Bayesian BN Adapt (Schneider et al., 2020). BN Adapt replaces the global batch normalization statistics learned during training with test batch statistics at inference. On the other hand, Bayesian BN Adapt assumes the global statistics of the training data as a prior and linearly interpolates between these statistics and the test batch statistics during inference. Additionally, we evaluate methods that employ classic regularization techniques such as Mixup (Yan et al., 2020) and IB-ERM (Ahuja et al., 2021). Table 2 shows the average performance attained by these methods across four benchmark datasets. Further, Table 7 and Table 8 demonstrate the average and worst group out-of-distribution performance, respectively, accompanied by the corresponding standard errors. These statistics are computed across three independent runs of the entire sweep, wherein the model selected for evaluation is the one with hyper-parameters yielding the highest validation accuracy.

Table 7: Average out-of-distribution test accuracies along with their corresponding standard errors for various counts of context samples. The methods compared include Adaptive Risk Minimization (ARM), Empirical Risk Minimization (ERM), Test Entropy Minimization (TENT), Batch Norm Adaptation (BN Adapt), Bayesian Batch Norm Adaptation (Bayesian BN Adapt), Fish, IB-ERM, IB-IRM, Mixup and our method ICRM on FEMNIST, Rotated MNIST, WILDS Camelyon17, Tiny-ImageNet-C, ImageNet R and CIFAR10-C.

| Dataset / algorithm | Average test accuracy (by # in-context examples) | | | | |
|---|---|---|---|---|---|
| **FEMNIST** | 0 | 25 | 50 | 75 | 100 |
| ARM | $49.5 \pm 1.0$ | $83.9 \pm 0.5$ | $84.4 \pm 0.5$ | $84.7 \pm 0.6$ | $84.6 \pm 0.3$ |
| TENT | $78.1 \pm 1.2$ | $77.9 \pm 1.2$ | $81.2 \pm 0.9$ | $82.5 \pm 0.9$ | $83.3 \pm 0.8$ |
| BN Adapt | $78.3 \pm 0.6$ | $76.9 \pm 1.4$ | $80.3 \pm 0.9$ | $81.5 \pm 0.7$ | $82.4 \pm 0.6$ |
| Bayesian BN Adapt | $78.3 \pm 1.8$ | $79.6 \pm 1.0$ | $81.3 \pm 0.6$ | $82.2 \pm 0.7$ | $82.9 \pm 0.8$ |
| Fish | $77.2 \pm 0.6$ | $77.2 \pm 0.6$ | $77.2 \pm 0.6$ | $77.2 \pm 0.6$ | $77.2 \pm 0.6$ |
| IB-ERM | $79.0 \pm 1.5$ | $79.0 \pm 1.5$ | $79.0 \pm 1.5$ | $79.0 \pm 1.5$ | $79.0 \pm 1.5$ |
| IB-IRM | $79.0 \pm 0.4$ | $79.0 \pm 0.4$ | $79.0 \pm 0.4$ | $79.0 \pm 0.4$ | $79.0 \pm 0.4$ |
| Mixup | $78.6 \pm 0.9$ | $78.6 \pm 0.9$ | $78.6 \pm 0.9$ | $78.6 \pm 0.9$ | $78.6 \pm 0.9$ |
| ERM | $\mathbf{79.3 \pm 0.4}$ | $79.3 \pm 0.4$ | $79.3 \pm 0.4$ | $79.3 \pm 0.4$ | $79.3 \pm 0.4$ |
| ICRM | $78.7 \pm 0.5$ | $\mathbf{87.2 \pm 0.4}$ | $\mathbf{87.4 \pm 0.5}$ | $\mathbf{87.5 \pm 0.2}$ | $\mathbf{87.8 \pm 0.2}$ |
| **Rotated MNIST** | 0 | 25 | 50 | 75 | 100 |
| ARM | $36.5 \pm 5.2$ | $94.2 \pm 0.7$ | $95.1 \pm 0.4$ | $95.3 \pm 0.4$ | $95.5 \pm 0.3$ |
| TENT | $94.1 \pm 0.3$ | $88.0 \pm 0.4$ | $91.9 \pm 0.3$ | $93.8 \pm 0.2$ | $94.3 \pm 0.2$ |
| BN Adapt | $94.6 \pm 0.8$ | $87.0 \pm 2.3$ | $91.5 \pm 1.5$ | $93.7 \pm 1.2$ | $94.3 \pm 1.0$ |
| Bayesian BN Adapt | $94.6 \pm 1.0$ | $91.2 \pm 1.6$ | $93.4 \pm 1.2$ | $94.3 \pm 1.0$ | $94.7 \pm 1.0$ |
| Fish | $94.8 \pm 0.4$ | $94.8 \pm 0.4$ | $94.8 \pm 0.4$ | $94.8 \pm 0.4$ | $94.8 \pm 0.4$ |
| IB-ERM | $92.2 \pm 0.5$ | $92.2 \pm 0.5$ | $92.2 \pm 0.5$ | $92.2 \pm 0.5$ | $92.2 \pm 0.5$ |
| IB-IRM | $91.0 \pm 1.1$ | $91.0 \pm 1.1$ | $91.0 \pm 1.1$ | $91.0 \pm 1.1$ | $91.0 \pm 1.1$ |
| Mixup | $93.6 \pm 0.0$ | $93.6 \pm 0.0$ | $93.6 \pm 0.0$ | $93.6 \pm 0.0$ | $93.6 \pm 0.0$ |
| ERM | $\mathbf{94.2 \pm 0.3}$ | $94.2 \pm 0.3$ | $94.2 \pm 0.3$ | $94.2 \pm 0.3$ | $94.2 \pm 0.3$ |
| ICRM | $93.6 \pm 0.2$ | $\mathbf{96.1 \pm 0.1}$ | $\mathbf{96.2 \pm 0.1}$ | $\mathbf{96.2 \pm 0.1}$ | $\mathbf{96.2 \pm 0.1}$ |
| **WILDS Camelyon17** | 0 | 25 | 50 | 75 | 100 |
| ARM | $61.2 \pm 5.2$ | $59.5 \pm 4.2$ | $59.7 \pm 4.2$ | $59.7 \pm 4.3$ | $59.7 \pm 4.2$ |
| TENT | $67.9 \pm 7.6$ | $81.8 \pm 1.1$ | $87.2 \pm 1.1$ | $89.4 \pm 1.1$ | $89.4 \pm 1.0$ |
| BN Adapt | $67.5 \pm 5.9$ | $82.0 \pm 0.3$ | $87.4 \pm 0.3$ | $89.7 \pm 0.3$ | $89.9 \pm 0.3$ |
| Bayesian BN Adapt | $67.5 \pm 6.1$ | $82.0 \pm 0.3$ | $87.3 \pm 0.3$ | $89.6 \pm 0.2$ | $89.7 \pm 0.3$ |
| Fish | $53.9 \pm 2.8$ | $53.9 \pm 2.8$ | $53.9 \pm 2.8$ | $53.9 \pm 2.8$ | $53.9 \pm 2.8$ |
| IB-ERM | $51.8 \pm 1.0$ | $51.8 \pm 1.0$ | $51.8 \pm 1.0$ | $51.8 \pm 1.0$ | $51.8 \pm 1.0$ |
| IB-IRM | $53.9 \pm 1.3$ | $53.9 \pm 1.3$ | $53.9 \pm 1.3$ | $53.9 \pm 1.3$ | $53.9 \pm 1.3$ |
| Mixup | $62.8 \pm 5.7$ | $62.8 \pm 5.7$ | $62.8 \pm 5.7$ | $62.8 \pm 5.7$ | $62.8 \pm 5.7$ |
| ERM | $68.6 \pm 7.8$ | $68.6 \pm 7.8$ | $68.6 \pm 7.8$ | $68.6 \pm 7.8$ | $68.6 \pm 7.8$ |
| ICRM | $\mathbf{92.0 \pm 0.6}$ | $\mathbf{90.7 \pm 0.8}$ | $\mathbf{90.8 \pm 0.8}$ | $\mathbf{90.8 \pm 0.8}$ | $\mathbf{90.8 \pm 0.8}$ |
| **Tiny ImageNet-C** | 0 | 25 | 50 | 75 | 100 |
| ARM | $30.8 \pm 0.2$ | $31.0 \pm 0.2$ | $31.0 \pm 0.2$ | $31.0 \pm 0.2$ | $31.0 \pm 0.2$ |
| TENT | $31.7 \pm 0.5$ | $1.6 \pm 0.1$ | $1.7 \pm 0.1$ | $2.0 \pm 0.1$ | $2.1 \pm 0.1$ |
| BN Adapt | $31.7 \pm 0.7$ | $1.7 \pm 0.1$ | $1.7 \pm 0.1$ | $1.9 \pm 0.1$ | $2.1 \pm 0.1$ |
| Bayesian BN Adapt | $31.7 \pm 0.8$ | $2.2 \pm 0.1$ | $2.1 \pm 0.1$ | $2.3 \pm 0.1$ | $2.4 \pm 0.1$ |
| Fish | $33.7 \pm 0.8$ | $33.7 \pm 0.8$ | $33.7 \pm 0.8$ | $33.7 \pm 0.8$ | $33.7 \pm 0.8$ |
| IB-ERM | $35.5 \pm 0.4$ | $35.5 \pm 0.4$ | $35.5 \pm 0.4$ | $35.5 \pm 0.4$ | $35.5 \pm 0.4$ |
| IB-IRM | $35.4 \pm 0.3$ | $35.4 \pm 0.3$ | $35.4 \pm 0.3$ | $35.4 \pm 0.3$ | $35.4 \pm 0.3$ |
| Mixup | $35.5 \pm 0.3$ | $35.5 \pm 0.3$ | $35.5 \pm 0.3$ | $35.5 \pm 0.3$ | $35.5 \pm 0.3$ |
| ERM | $31.8 \pm 0.6$ | $31.8 \pm 0.6$ | $31.8 \pm 0.6$ | $31.8 \pm 0.6$ | $31.8 \pm 0.6$ |
| ICRM | $\mathbf{38.3 \pm 0.1}$ | $\mathbf{39.2 \pm 0.3}$ | $\mathbf{39.2 \pm 0.3}$ | $\mathbf{39.2 \pm 0.3}$ | $\mathbf{39.2 \pm 0.3}$ |
| **ImageNet R** | 0 | 25 | 50 | 75 | 100 |
| ARM | $56.3 \pm 0.8$ | $58.1 \pm 0.3$ | $58.8 \pm 0.8$ | $\mathbf{59.8 \pm 0.8}$ | $59.0 \pm 0.3$ |
| TENT | $\mathbf{58.9 \pm 0.5}$ | $10.1 \pm 0.2$ | $10.7 \pm 0.1$ | $12.1 \pm 0.2$ | $13.0 \pm 0.1$ |
| BN Adapt | $\mathbf{58.9 \pm 0.5}$ | $9.9 \pm 0.1$ | $10.9 \pm 0.1$ | $12.2 \pm 0.2$ | $13.1 \pm 0.1$ |
| Bayesian BN Adapt | $\mathbf{58.9 \pm 0.5}$ | $11.9 \pm 0.1$ | $12.3 \pm 0.1$ | $13.9 \pm 0.3$ | $14.6 \pm 0.1$ |
| Fish | $58.6 \pm 1.0$ | $58.6 \pm 1.0$ | $58.6 \pm 1.0$ | $58.6 \pm 1.0$ | $58.6 \pm 1.0$ |
| IB-ERM | $58.5 \pm 0.5$ | $58.5 \pm 0.5$ | $58.5 \pm 0.5$ | $58.5 \pm 0.5$ | $58.5 \pm 0.5$ |
| IB-IRM | $57.8 \pm 0.0$ | $57.8 \pm 0.0$ | $57.8 \pm 0.0$ | $57.8 \pm 0.0$ | $57.8 \pm 0.0$ |
| Mixup | $58.8 \pm 0.8$ | $58.8 \pm 0.8$ | $58.8 \pm 0.8$ | $58.8 \pm 0.8$ | $58.8 \pm 0.8$ |
| ERM | $\mathbf{58.9 \pm 0.5}$ | $58.9 \pm 0.5$ | $58.9 \pm 0.5$ | $58.9 \pm 0.5$ | $58.9 \pm 0.5$ |
| ICRM | $57.4 \pm 0.4$ | $\mathbf{59.7 \pm 0.4}$ | $\mathbf{59.6 \pm 0.6}$ | $59.4 \pm 0.4$ | $\mathbf{60.5 \pm 0.3}$ |
| **CIFAR10-C** | 0 | 25 | 50 | 75 | 100 |
| ARM | $65.9 \pm 1.3$ | $66.0 \pm 1.3$ | $66.0 \pm 1.3$ | $66.0 \pm 1.3$ | $66.0 \pm 1.3$ |
| TENT | $66.1 \pm 1.6$ | $63.9 \pm 2.1$ | $68.4 \pm 2.1$ | $69.9 \pm 2.0$ | $70.5 \pm 2.0$ |
| BN Adapt | $66.1 \pm 1.6$ | $63.9 \pm 2.1$ | $68.4 \pm 2.1$ | $69.9 \pm 2.0$ | $70.1 \pm 2.0$ |
| Bayesian BN Adapt | $66.1 \pm 1.6$ | $65.1 \pm 2.1$ | $68.9 \pm 2.0$ | $69.8 \pm 2.0$ | $70.0 \pm 2.0$ |
| Fish | $72.3 \pm 1.0$ | $72.3 \pm 1.0$ | $72.3 \pm 1.0$ | $72.3 \pm 1.0$ | $72.3 \pm 1.0$ |
| IB-ERM | $65.2 \pm 2.9$ | $65.2 \pm 2.9$ | $65.2 \pm 2.9$ | $65.2 \pm 2.9$ | $65.2 \pm 2.9$ |
| IB-IRM | $64.3 \pm 2.6$ | $64.3 \pm 2.6$ | $64.3 \pm 2.6$ | $64.3 \pm 2.6$ | $64.3 \pm 2.6$ |
| Mixup | $\mathbf{72.8 \pm 0.4}$ | $\mathbf{72.8 \pm 0.4}$ | $\mathbf{72.8 \pm 0.4}$ | $\mathbf{72.8 \pm 0.4}$ | $\mathbf{72.8 \pm 0.4}$ |
| ERM | $66.1 \pm 1.6$ | $66.1 \pm 1.6$ | $66.1 \pm 1.6$ | $66.1 \pm 1.6$ | $66.1 \pm 1.6$ |
| ICRM | $70.6 \pm 0.2$ | $71.0 \pm 0.2$ | $71.0 \pm 0.2$ | $71.0 \pm 0.2$ | $71.0 \pm 0.3$ |

Table 8: Worst environment out-of-distribution test accuracies along with their corresponding standard errors for various counts of context samples. The methods compared include Adaptive Risk Minimization (ARM), Empirical Risk Minimization (ERM), Test Entropy Minimization (TENT), Batch Norm Adaptation (BN Adapt), Bayesian Batch Norm Adaptation (Bayesian BN Adapt), Fish, IB-ERM, IB-IRM, Mixup and our method ICRM on FEMNIST, Rotated MNIST, WILDS Camelyon17, Tiny-ImageNet-C, ImageNet R and CIFAR10-C.

| Dataset / algorithm | Worst case test accuracy (by # in-context examples) | | | | |
| --- | --- | --- | --- | --- | --- |
| | 0 | 25 | 50 | 75 | 100 |
| **FEMNIST** | | | | | |
| ARM | $23.6 \pm 1.7$ | $59.5 \pm 3.5$ | $60.7 \pm 3.8$ | $57.0 \pm 7.3$ | $58.8 \pm 4.0$ |
| TENT | $55.2 \pm 2.5$ | $57.2 \pm 2.2$ | $63.3 \pm 0.4$ | $65.9 \pm 0.6$ | $67.2 \pm 1.0$ |
| BN Adapt | $52.7 \pm 6.2$ | $56.2 \pm 2.5$ | $61.9 \pm 0.1$ | $64.7 \pm 2.5$ | $65.3 \pm 0.9$ |
| Bayesian BN Adapt | $54.3 \pm 2.6$ | $60.4 \pm 1.2$ | $64.7 \pm 0.9$ | $65.5 \pm 2.2$ | $66.3 \pm 1.2$ |
| Fish | $52.8 \pm 1.2$ | $52.8 \pm 1.2$ | $52.8 \pm 1.2$ | $52.8 \pm 1.2$ | $52.8 \pm 1.2$ |
| IB-ERM | $58.6 \pm 3.4$ | $58.6 \pm 3.4$ | $58.6 \pm 3.4$ | $58.6 \pm 3.4$ | $58.6 \pm 3.4$ |
| IB-IRM | $57.3 \pm 2.6$ | $57.3 \pm 2.6$ | $57.3 \pm 2.6$ | $57.3 \pm 2.6$ | $57.3 \pm 2.6$ |
| Mixup | $57.0 \pm 1.9$ | $57.0 \pm 1.9$ | $57.0 \pm 1.9$ | $57.0 \pm 1.9$ | $57.0 \pm 1.9$ |
| ERM | $59.0 \pm 0.2$ | $59.0 \pm 0.2$ | $59.0 \pm 0.2$ | $59.0 \pm 0.2$ | $59.0 \pm 0.2$ |
| ICRM | $\mathbf{59.8 \pm 0.7}$ | $\mathbf{69.3 \pm 0.0}$ | $\mathbf{70.6 \pm 2.3}$ | $\mathbf{70.6 \pm 1.5}$ | $\mathbf{70.6 \pm 0.7}$ |
| **Rotated MNIST** | 0 | 25 | 50 | 75 | 100 |
| ARM | $28.2 \pm 2.1$ | $85.3 \pm 1.6$ | $87.2 \pm 1.0$ | $87.9 \pm 1.0$ | $87.9 \pm 0.9$ |
| TENT | $80.2 \pm 1.3$ | $\mathbf{88.5 \pm 0.8}$ | $88.5 \pm 0.9$ | $80.2 \pm 1.0$ | $81.3 \pm 1.0$ |
| BN Adapt | $80.5 \pm 2.4$ | $70.9 \pm 2.8$ | $76.9 \pm 2.5$ | $79.8 \pm 2.7$ | $80.9 \pm 2.3$ |
| Bayesian BN Adapt | $80.5 \pm 2.9$ | $75.4 \pm 3.1$ | $79.2 \pm 2.6$ | $80.7 \pm 2.8$ | $81.3 \pm 2.5$ |
| Fish | $83.2 \pm 1.9$ | $83.2 \pm 1.9$ | $83.2 \pm 1.9$ | $83.2 \pm 1.9$ | $83.2 \pm 1.9$ |
| IB-ERM | $72.0 \pm 0.9$ | $72.0 \pm 0.9$ | $72.0 \pm 0.9$ | $72.0 \pm 0.9$ | $72.0 \pm 0.9$ |
| IB-IRM | $69.9 \pm 3.4$ | $69.9 \pm 3.4$ | $69.9 \pm 3.4$ | $69.9 \pm 3.4$ | $69.9 \pm 3.4$ |
| Mixup | $81.2 \pm 0.7$ | $81.2 \pm 0.7$ | $81.2 \pm 0.7$ | $81.2 \pm 0.7$ | $81.2 \pm 0.7$ |
| ERM | $80.8 \pm 1.1$ | $80.8 \pm 1.1$ | $80.8 \pm 1.1$ | $80.8 \pm 1.1$ | $80.8 \pm 1.1$ |
| ICRM | $\mathbf{82.5 \pm 0.5}$ | $\mathbf{88.5 \pm 0.5}$ | $\mathbf{88.5 \pm 0.5}$ | $\mathbf{88.8 \pm 0.5}$ | $\mathbf{88.8 \pm 0.4}$ |
| **WILDS Camelyon17** | 0 | 25 | 50 | 75 | 100 |
| ARM | $61.2 \pm 5.2$ | $59.5 \pm 4.2$ | $59.7 \pm 4.2$ | $59.7 \pm 4.3$ | $59.7 \pm 4.2$ |
| TENT | $67.9 \pm 7.6$ | $81.8 \pm 1.1$ | $87.2 \pm 1.1$ | $89.4 \pm 1.1$ | $89.4 \pm 1.0$ |
| BN Adapt | $67.5 \pm 5.9$ | $82.0 \pm 0.3$ | $87.4 \pm 0.3$ | $89.7 \pm 0.3$ | $89.9 \pm 0.3$ |
| Bayesian BN Adapt | $67.5 \pm 6.1$ | $82.0 \pm 0.3$ | $87.3 \pm 0.3$ | $89.6 \pm 0.2$ | $89.7 \pm 0.3$ |
| Fish | $53.9 \pm 2.8$ | $53.9 \pm 2.8$ | $53.9 \pm 2.8$ | $53.9 \pm 2.8$ | $53.9 \pm 2.8$ |
| IB-ERM | $51.8 \pm 1.0$ | $51.8 \pm 1.0$ | $51.8 \pm 1.0$ | $51.8 \pm 1.0$ | $51.8 \pm 1.0$ |
| IB-IRM | $53.9 \pm 1.3$ | $53.9 \pm 1.3$ | $53.9 \pm 1.3$ | $53.9 \pm 1.3$ | $53.9 \pm 1.3$ |
| Mixup | $62.8 \pm 5.7$ | $62.8 \pm 5.7$ | $62.8 \pm 5.7$ | $62.8 \pm 5.7$ | $62.8 \pm 5.7$ |
| ERM | $68.6 \pm 7.8$ | $68.6 \pm 7.8$ | $68.6 \pm 7.8$ | $68.6 \pm 7.8$ | $68.6 \pm 7.8$ |
| ICRM | $\mathbf{92.0 \pm 0.6}$ | $\mathbf{90.7 \pm 0.8}$ | $\mathbf{90.8 \pm 0.8}$ | $\mathbf{90.8 \pm 0.8}$ | $\mathbf{90.8 \pm 0.8}$ |
| **Tiny ImageNet-C** | 0 | 25 | 50 | 75 | 100 |
| ARM | $8.2 \pm 0.3$ | $8.3 \pm 0.3$ | $8.2 \pm 0.3$ | $8.3 \pm 0.3$ | $8.2 \pm 0.3$ |
| TENT | $1.2 \pm 0.4$ | $1.4 \pm 0.0$ | $1.6 \pm 0.1$ | $1.6 \pm 0.0$ | $1.6 \pm 0.0$ |
| BN Adapt | $9.4 \pm 0.7$ | $1.3 \pm 0.0$ | $1.4 \pm 0.0$ | $1.6 \pm 0.0$ | $1.7 \pm 0.0$ |
| Bayesian BN Adapt | $9.4 \pm 0.7$ | $1.6 \pm 0.2$ | $1.6 \pm 0.1$ | $1.8 \pm 0.0$ | $1.8 \pm 0.0$ |
| Fish | $11.1 \pm 0.1$ | $11.1 \pm 0.1$ | $11.1 \pm 0.1$ | $11.1 \pm 0.1$ | $11.1 \pm 0.1$ |
| IB-ERM | $15.8 \pm 0.6$ | $15.8 \pm 0.6$ | $15.8 \pm 0.6$ | $15.8 \pm 0.6$ | $15.8 \pm 0.6$ |
| IB-IRM | $15.9 \pm 0.4$ | $15.9 \pm 0.4$ | $15.9 \pm 0.4$ | $15.9 \pm 0.4$ | $15.9 \pm 0.4$ |
| Mixup | $11.3 \pm 0.5$ | $11.3 \pm 0.5$ | $11.3 \pm 0.5$ | $11.3 \pm 0.5$ | $11.3 \pm 0.5$ |
| ERM | $9.5 \pm 0.4$ | $9.5 \pm 0.4$ | $9.5 \pm 0.4$ | $9.5 \pm 0.4$ | $9.5 \pm 0.4$ |
| ICRM | $\mathbf{18.8 \pm 0.2}$ | $\mathbf{19.2 \pm 0.1}$ | $\mathbf{19.5 \pm 0.2}$ | $\mathbf{19.5 \pm 0.1}$ | $\mathbf{19.4 \pm 0.2}$ |
| **ImageNet R** | 0 | 25 | 50 | 75 | 100 |
| ARM | $47.4 \pm 1.1$ | $45.3 \pm 0.4$ | $47.2 \pm 1.9$ | $\mathbf{49.8 \pm 1.2}$ | $47.4 \pm 1.0$ |
| TENT | $\mathbf{48.0 \pm 1.0}$ | $8.6 \pm 0.1$ | $8.4 \pm 0.1$ | $8.9 \pm 0.1$ | $9.1 \pm 0.1$ |
| BN Adapt | $\mathbf{48.0 \pm 1.0}$ | $8.5 \pm 0.1$ | $8.5 \pm 0.1$ | $8.9 \pm 0.1$ | $9.1 \pm 0.0$ |
| Bayesian BN Adapt | $\mathbf{48.0 \pm 1.0}$ | $10.5 \pm 0.2$ | $10.3 \pm 0.2$ | $10.7 \pm 0.2$ | $10.9 \pm 0.2$ |
| Fish | $46.0 \pm 2.1$ | $46.0 \pm 2.1$ | $46.0 \pm 2.1$ | $46.0 \pm 2.1$ | $46.0 \pm 2.1$ |
| IB-ERM | $47.2 \pm 1.3$ | $47.2 \pm 1.3$ | $47.2 \pm 1.3$ | $47.2 \pm 1.3$ | $47.2 \pm 1.3$ |
| IB-IRM | $47.2 \pm 0.4$ | $47.2 \pm 0.4$ | $47.2 \pm 0.4$ | $47.2 \pm 0.4$ | $47.2 \pm 0.4$ |
| Mixup | $47.9 \pm 2.1$ | $47.9 \pm 2.1$ | $47.9 \pm 2.1$ | $47.9 \pm 2.1$ | $47.9 \pm 2.1$ |
| ERM | $\mathbf{48.0 \pm 1.0}$ | $\mathbf{48.0 \pm 1.0}$ | $\mathbf{48.0 \pm 1.0}$ | $48.0 \pm 1.0$ | $48.0 \pm 1.0$ |
| ICRM | $45.4 \pm 0.7$ | $\mathbf{48.0 \pm 0.2}$ | $47.2 \pm 0.8$ | $46.9 \pm 0.4$ | $\mathbf{50.6 \pm 1.3}$ |
| **CIFAR10-C** | 0 | 25 | 50 | 75 | 100 |
| ARM | $39.3 \pm 1.7$ | $39.3 \pm 1.7$ | $39.4 \pm 1.7$ | $39.3 \pm 1.7$ | $39.4 \pm 1.7$ |
| TENT | $39.8 \pm 2.5$ | $45.4 \pm 2.1$ | $48.9 \pm 2.1$ | $49.7 \pm 2.0$ | $52.6 \pm 2.0$ |
| BN Adapt | $39.8 \pm 2.5$ | $43.8 \pm 2.1$ | $45.1 \pm 2.0$ | $47.8 \pm 2.0$ | $48.6 \pm 2.0$ |
| Bayesian BN Adapt | $39.8 \pm 2.5$ | $44.5 \pm 2.0$ | $46.8 \pm 2.0$ | $49.6 \pm 2.0$ | $51.0 \pm 2.1$ |
| Fish | $49.9 \pm 1.5$ | $49.9 \pm 1.5$ | $49.9 \pm 1.5$ | $49.9 \pm 1.5$ | $49.9 \pm 1.5$ |
| IB-ERM | $44.9 \pm 3.4$ | $44.9 \pm 3.4$ | $44.9 \pm 3.4$ | $44.9 \pm 3.4$ | $44.9 \pm 3.4$ |
| IB-IRM | $43.3 \pm 2.3$ | $43.3 \pm 2.3$ | $43.3 \pm 2.3$ | $43.3 \pm 2.3$ | $43.3 \pm 2.3$ |
| Mixup | $53.9 \pm 2.4$ | $53.9 \pm 2.4$ | $53.9 \pm 2.4$ | $53.9 \pm 2.4$ | $53.9 \pm 2.4$ |
| ERM | $39.8 \pm 2.5$ | $39.8 \pm 2.5$ | $39.8 \pm 2.5$ | $39.8 \pm 2.5$ | $39.8 \pm 2.5$ |
| ICRM | $\mathbf{54.6 \pm 0.4}$ | $\mathbf{56.0 \pm 0.5}$ | $\mathbf{55.8 \pm 0.5}$ | $\mathbf{55.8 \pm 0.5}$ | $\mathbf{55.9 \pm 0.5}$ |

### D.2.2  Robustness of ICRM in the absence of environment labels

As outlined in Section 4, the training regimen of ICRM assumes a dataset $\mathcal{D} = \{(x_i, y_i, e_i)\}_{i=1}^n$ collected under multiple training environments $e_i \in \mathcal{E}_{tr}$. However, in scenarios lacking such domain separation during training, does ICRM continue to show an edge over ERM baselines? To study this question, we modify the sampling strategy: rather than constructing context vectors containing examples from one environment, we construct context vectors containing iid samples from all of the environments pooled together. To continue to test for out-of-distribution generalization, however, we evaluate the performance on examples from a novel test environment. We term this modified approach ICRM-Mix.

Table 9 and Table 10 contrasts the performance of ICRM with ICRM-Mix. ICRM consistently outperforms ICRM-Mix across varying counts of in-context samples on both FEMNIST and Rotated MNIST. Surprisingly, ICRM-Mix and ICRM perform similarly on WILDS Camelyon17 and Tiny ImageNet-C. Consider a setting where the model benefits the most attending to examples from the same class or related classes. If classes are distributed uniformly across domains, then ICRM and ICRM-mix are bound to perform similarly. Consider another setting where the model benefits the most by attending to environment-specific examples such as characters drawn by the same user. In such a case, ICRM and ICRM-mix have very different performances.

Table 9: Average out-of-distribution test accuracies along with their corresponding standard errors for ICRM and ICRM-Mix across FEMNIST, Rotated MNIST, WILDS Camelyon17 and Tiny-ImageNet-C. ICRM-Mix trains on sequences with samples drawn i.i.d. from the unified dataset comprising various environments.

| Dataset / algorithm | Average test accuracy (by # in-context examples) | | | | |
|---|---|---|---|---|---|
| **FEMNIST** | 0 | 25 | 50 | 75 | 100 |
| ICRM | $78.7 \pm 0.5$ | $87.2 \pm 0.4$ | $87.4 \pm 0.5$ | $87.5 \pm 0.2$ | $87.8 \pm 0.2$ |
| ICRM-Mix | $77.6 \pm 0.8$ | $81.1 \pm 0.2$ | $81.1 \pm 0.2$ | $80.9 \pm 0.3$ | $80.9 \pm 0.1$ |
| **Rotated MNIST** | 0 | 25 | 50 | 75 | 100 |
| ICRM | $93.6 \pm 0.2$ | $96.1 \pm 0.1$ | $96.2 \pm 0.1$ | $96.2 \pm 0.1$ | $96.2 \pm 0.1$ |
| ICRM-Mix | $88.9 \pm 1.4$ | $92.6 \pm 0.3$ | $92.7 \pm 0.2$ | $92.6 \pm 0.3$ | $92.7 \pm 0.2$ |
| **WILDS Camelyon17** | 0 | 25 | 50 | 75 | 100 |
| ICRM | $92.0 \pm 0.6$ | $90.7 \pm 0.8$ | $90.8 \pm 0.8$ | $90.8 \pm 0.8$ | $90.8 \pm 0.8$ |
| ICRM-Mix | $92.9 \pm 0.3$ | $90.7 \pm 0.6$ | $90.8 \pm 0.5$ | $90.7 \pm 0.5$ | $90.7 \pm 0.5$ |
| **Tiny ImageNet-C** | 0 | 25 | 50 | 75 | 100 |
| ICRM | $38.3 \pm 0.1$ | $39.2 \pm 0.3$ | $39.2 \pm 0.3$ | $39.2 \pm 0.3$ | $39.2 \pm 0.3$ |
| ICRM-Mix | $38.4 \pm 0.2$ | $39.3 \pm 0.2$ | $39.3 \pm 0.2$ | $39.3 \pm 0.2$ | $39.3 \pm 0.2$ |
| **Imagenet R** | 0 | 25 | 50 | 75 | 100 |
| ICRM | $57.4 \pm 0.4$ | $59.7 \pm 0.4$ | $59.6 \pm 0.6$ | $59.4 \pm 0.4$ | $60.5 \pm 0.3$ |
| ICRM-Mix | $54.9 \pm 1.0$ | $54.9 \pm 1.0$ | $57.8 \pm 1.0$ | $57.8 \pm 1.0$ | $57.8 \pm 1.0$ |
| **CIFAR10-C** | 0 | 25 | 50 | 75 | 100 |
| ICRM | $70.6 \pm 0.2$ | $71.0 \pm 0.2$ | $71.0 \pm 0.2$ | $71.0 \pm 0.2$ | $71.0 \pm 0.3$ |
| ICRM-Mix | $69.2 \pm 0.2$ | $69.4 \pm 0.3$ | $69.4 \pm 0.3$ | $69.4 \pm 0.3$ | $69.4 \pm 0.3$ |

### D.2.3  Understanding the impact of architecture

Table 3 presents the average performance of both $\text{ERM}^+$ and $\text{ARM}^+$ relative to ERM and ARM, across the four datasets. Further, Table 11 and Table 12 demonstrate the average and worst group out-of-distribution performance of these approaches, respectively, along with the corresponding standard errors. These statistics are computed across three independent runs of the entire sweep, wherein the model selected for evaluation is the one with hyper-parameters yielding the highest validation accuracy.

Table 10: Worst environment out-of-distribution test accuracies along with their corresponding standard errors for ICRM and ICRM-Mix across FEMNIST, Rotated MNIST, WILDS Camelyon17 and Tiny-ImageNet-C. ICRM-Mix trains on sequences with samples drawn i.i.d. from the unified dataset comprising various environments.

| Dataset / algorithm | Worst case test accuracy (by # in-context examples) | | | | |
|---|---|---|---|---|---|
| FEMNIST | 0 | 25 | 50 | 75 | 100 |
| ICRM | $59.8 \pm 0.7$ | $69.3 \pm 0.0$ | $70.6 \pm 2.3$ | $70.6 \pm 1.5$ | $70.6 \pm 0.7$ |
| ICRM-Mix | $57.5 \pm 1.4$ | $62.7 \pm 1.1$ | $65.0 \pm 0.3$ | $64.1 \pm 1.5$ | $62.9 \pm 2.3$ |
| Rotated MNIST | 0 | 25 | 50 | 75 | 100 |
| ICRM | $82.5 \pm 0.5$ | $88.5 \pm 0.5$ | $88.5 \pm 0.5$ | $88.8 \pm 0.5$ | $88.8 \pm 0.4$ |
| ICRM-Mix | $68.8 \pm 3.8$ | $77.1 \pm 0.7$ | $76.8 \pm 0.9$ | $76.4 \pm 0.9$ | $76.6 \pm 0.9$ |
| WILDS Camelyon17 | 0 | 25 | 50 | 75 | 100 |
| ICRM | $92.0 \pm 0.6$ | $90.7 \pm 0.8$ | $90.8 \pm 0.8$ | $90.8 \pm 0.8$ | $90.8 \pm 0.8$ |
| ICRM-Mix | $92.9 \pm 0.3$ | $90.7 \pm 0.6$ | $90.8 \pm 0.5$ | $90.7 \pm 0.5$ | $90.7 \pm 0.5$ |
| Tiny ImageNet-C | 0 | 25 | 50 | 75 | 100 |
| ICRM | $18.8 \pm 0.2$ | $19.2 \pm 0.1$ | $19.5 \pm 0.2$ | $19.5 \pm 0.1$ | $19.4 \pm 0.2$ |
| ICRM-Mix | $18.7 \pm 0.2$ | $19.2 \pm 0.2$ | $19.4 \pm 0.1$ | $19.5 \pm 0.1$ | $19.4 \pm 0.1$ |
| Imagenet R | 0 | 25 | 50 | 75 | 100 |
| ICRM | $45.4 \pm 0.7$ | $48.0 \pm 0.2$ | $47.2 \pm 0.8$ | $46.9 \pm 0.4$ | $50.6 \pm 1.3$ |
| ICRM-Mix | $44.4 \pm 1.7$ | $46.9 \pm 0.4$ | $48.1 \pm 1.6$ | $46.6 \pm 0.6$ | $48.7 \pm 1.0$ |
| CIFAR10-C | 0 | 25 | 50 | 75 | 100 |
| ICRM | $54.6 \pm 0.4$ | $56.0 \pm 0.5$ | $55.8 \pm 0.5$ | $55.8 \pm 0.5$ | $55.9 \pm 0.5$ |
| ICRM-Mix | $53.3 \pm 0.0$ | $54.2 \pm 1.1$ | $54.2 \pm 1.1$ | $54.3 \pm 1.1$ | $54.2 \pm 1.1$ |

Table 11: Average out-of-distribution test accuracies along with their corresponding standard errors for ARM$^+$ and ERM$^+$ in contrast to their base algorithms, ARM and ERM across FEMNIST, Rotated MNIST, WILDS Camelyon17 and Tiny-ImageNet-C.

| Dataset / algorithm | Average test accuracy (by # in-context examples) | | | | |
|---|---|---|---|---|---|
| FEMNIST | 0 | 25 | 50 | 75 | 100 |
| ARM | $49.5 \pm 1.0$ | $83.9 \pm 0.5$ | $84.4 \pm 0.5$ | $84.7 \pm 0.6$ | $84.6 \pm 0.3$ |
| ARM$^+$ | $71.4 \pm 1.2$ | $83.4 \pm 0.2$ | $84.0 \pm 0.2$ | $83.8 \pm 0.2$ | $83.5 \pm 0.1$ |
| ERM | $79.3 \pm 0.4$ | $79.3 \pm 0.4$ | $79.3 \pm 0.4$ | $79.3 \pm 0.4$ | $79.3 \pm 0.4$ |
| ERM$^+$ | $77.4 \pm 1.3$ | $77.4 \pm 1.3$ | $77.4 \pm 1.3$ | $77.4 \pm 1.3$ | $77.4 \pm 1.3$ |
| Rotated MNIST | 0 | 25 | 50 | 75 | 100 |
| ARM | $36.5 \pm 5.2$ | $94.2 \pm 0.7$ | $95.1 \pm 0.4$ | $95.3 \pm 0.4$ | $95.5 \pm 0.3$ |
| ARM$^+$ | $86.9 \pm 2.0$ | $92.6 \pm 0.7$ | $92.7 \pm 0.6$ | $92.8 \pm 0.6$ | $92.8 \pm 0.6$ |
| ERM | $94.2 \pm 0.3$ | $94.2 \pm 0.3$ | $94.2 \pm 0.3$ | $94.2 \pm 0.3$ | $94.2 \pm 0.3$ |
| ERM$^+$ | $94.3 \pm 0.4$ | $94.3 \pm 0.4$ | $94.3 \pm 0.4$ | $94.3 \pm 0.4$ | $94.3 \pm 0.4$ |
| WILDS Camelyon17 | 0 | 25 | 50 | 75 | 100 |
| ARM | $61.2 \pm 5.2$ | $59.5 \pm 4.2$ | $59.7 \pm 4.2$ | $59.7 \pm 4.3$ | $59.7 \pm 4.2$ |
| ARM$^+$ | $55.8 \pm 0.8$ | $55.1 \pm 1.7$ | $55.0 \pm 1.7$ | $55.0 \pm 1.8$ | $55.0 \pm 1.8$ |
| ERM | $68.6 \pm 7.8$ | $68.6 \pm 7.8$ | $68.6 \pm 7.8$ | $68.6 \pm 7.8$ | $68.6 \pm 7.8$ |
| ERM$^+$ | $50.1 \pm 0.1$ | $50.1 \pm 0.1$ | $50.1 \pm 0.1$ | $50.1 \pm 0.1$ | $50.1 \pm 0.1$ |
| Tiny ImageNet-C | 0 | 25 | 50 | 75 | 100 |
| ARM | $30.8 \pm 0.2$ | $31.0 \pm 0.2$ | $31.0 \pm 0.2$ | $31.0 \pm 0.2$ | $31.0 \pm 0.2$ |
| ARM$^+$ | $5.5 \pm 0.2$ | $5.7 \pm 0.2$ | $5.7 \pm 0.2$ | $5.7 \pm 0.2$ | $5.7 \pm 0.2$ |
| ERM | $31.8 \pm 0.6$ | $31.8 \pm 0.6$ | $31.8 \pm 0.6$ | $31.8 \pm 0.6$ | $31.8 \pm 0.6$ |
| ERM$^+$ | $29.7 \pm 0.3$ | $29.7 \pm 0.3$ | $29.7 \pm 0.3$ | $29.7 \pm 0.3$ | $29.7 \pm 0.3$ |
| Imagenet R | 0 | 25 | 50 | 75 | 100 |
| ARM | $56.3 \pm 0.8$ | $58.1 \pm 0.3$ | $58.8 \pm 0.8$ | $59.8 \pm 0.8$ | $59.0 \pm 0.3$ |
| ARM$^+$ | $1.8 \pm 0.2$ | $1.5 \pm 0.2$ | $1.5 \pm 0.2$ | $1.5 \pm 0.2$ | $1.5 \pm 0.1$ |
| ERM | $58.9 \pm 0.5$ | $58.9 \pm 0.5$ | $58.9 \pm 0.5$ | $58.9 \pm 0.5$ | $58.9 \pm 0.5$ |
| ERM$^+$ | $57.0 \pm 0.4$ | $57.0 \pm 0.4$ | $57.0 \pm 0.4$ | $57.0 \pm 0.4$ | $57.0 \pm 0.4$ |
| CIFAR10-C | 0 | 25 | 50 | 75 | 100 |
| ARM | $65.9 \pm 1.3$ | $66.0 \pm 1.3$ | $66.0 \pm 1.3$ | $66.0 \pm 1.3$ | $66.0 \pm 1.3$ |
| ARM$^+$ | $42.7 \pm 0.1$ | $44.2 \pm 0.1$ | $44.3 \pm 0.1$ | $44.3 \pm 0.1$ | $44.3 \pm 0.1$ |
| ERM | $66.1 \pm 1.6$ | $66.1 \pm 1.6$ | $66.1 \pm 1.6$ | $66.1 \pm 1.6$ | $66.1 \pm 1.6$ |
| ERM$^+$ | $66.5 \pm 1.2$ | $66.5 \pm 1.2$ | $66.5 \pm 1.2$ | $66.5 \pm 1.2$ | $66.5 \pm 1.2$ |

Table 12: Worst environment out-of-distribution test accuracies along with their corresponding standard errors for ARM$^+$ and ERM$^+$ in contrast to their base algorithms, ARM and ERM across FEMNIST, Rotated MNIST, WILDS Camelyon17 and Tiny-ImageNet-C.

| Dataset / algorithm | Worst case test accuracy (by # in-context examples) | | | | |
|---|---|---|---|---|---|
| FEMNIST | 0 | 25 | 50 | 75 | 100 |
| ARM | $23.6 \pm 1.7$ | $59.5 \pm 3.5$ | $60.7 \pm 3.8$ | $57.0 \pm 7.3$ | $58.8 \pm 4.0$ |
| ARM$^+$ | $51.7 \pm 2.2$ | $63.0 \pm 2.1$ | $64.0 \pm 0.8$ | $60.7 \pm 1.6$ | $62.0 \pm 0.8$ |
| ERM | $59.0 \pm 0.2$ | $59.0 \pm 0.2$ | $59.0 \pm 0.2$ | $59.0 \pm 0.2$ | $59.0 \pm 0.2$ |
| ERM$^+$ | $53.3 \pm 2.7$ | $53.3 \pm 2.7$ | $53.3 \pm 2.7$ | $53.3 \pm 2.7$ | $53.3 \pm 2.7$ |
| Rotated MNIST | 0 | 25 | 50 | 75 | 100 |
| ARM | $28.2 \pm 2.1$ | $85.3 \pm 1.6$ | $87.2 \pm 1.0$ | $87.9 \pm 1.0$ | $87.9 \pm 0.9$ |
| ARM$^+$ | $71.4 \pm 2.6$ | $80.9 \pm 1.8$ | $81.0 \pm 1.8$ | $81.2 \pm 1.9$ | $81.1 \pm 1.8$ |
| ERM | $80.8 \pm 1.1$ | $80.8 \pm 1.1$ | $80.8 \pm 1.1$ | $80.8 \pm 1.1$ | $80.8 \pm 1.1$ |
| ERM$^+$ | $81.9 \pm 0.7$ | $81.9 \pm 0.7$ | $81.9 \pm 0.7$ | $81.9 \pm 0.7$ | $81.9 \pm 0.7$ |
| WILDS Camelyon17 | 0 | 25 | 50 | 75 | 100 |
| ARM | $61.2 \pm 5.2$ | $59.5 \pm 4.2$ | $59.7 \pm 4.2$ | $59.7 \pm 4.3$ | $59.7 \pm 4.2$ |
| ARM$^+$ | $55.8 \pm 0.8$ | $55.1 \pm 1.7$ | $55.0 \pm 1.7$ | $55.0 \pm 1.8$ | $55.0 \pm 1.8$ |
| ERM | $68.6 \pm 7.8$ | $68.6 \pm 7.8$ | $68.6 \pm 7.8$ | $68.6 \pm 7.8$ | $68.6 \pm 7.8$ |
| ERM$^+$ | $50.1 \pm 0.1$ | $50.1 \pm 0.1$ | $50.1 \pm 0.1$ | $50.1 \pm 0.1$ | $50.1 \pm 0.1$ |
| Tiny ImageNet-C | 0 | 25 | 50 | 75 | 100 |
| ARM | $8.2 \pm 0.3$ | $8.3 \pm 0.3$ | $8.2 \pm 0.3$ | $8.3 \pm 0.3$ | $8.2 \pm 0.3$ |
| ARM$^+$ | $1.9 \pm 0.1$ | $1.9 \pm 0.1$ | $1.9 \pm 0.1$ | $1.9 \pm 0.1$ | $1.9 \pm 0.1$ |
| ERM | $9.5 \pm 0.4$ | $9.5 \pm 0.4$ | $9.5 \pm 0.4$ | $9.5 \pm 0.4$ | $9.5 \pm 0.4$ |
| ERM$^+$ | $8.3 \pm 0.3$ | $8.3 \pm 0.3$ | $8.3 \pm 0.3$ | $8.3 \pm 0.3$ | $8.3 \pm 0.3$ |
| Imagenet R | 0 | 25 | 50 | 75 | 100 |
| ARM | $47.4 \pm 1.1$ | $45.3 \pm 0.4$ | $47.2 \pm 1.9$ | $49.8 \pm 1.2$ | $47.4 \pm 1.0$ |
| ARM$^+$ | $1.4 \pm 0.2$ | $0.9 \pm 0.3$ | $1.0 \pm 0.4$ | $0.8 \pm 0.3$ | $1.0 \pm 0.3$ |
| ERM | $48.0 \pm 1.0$ | $48.0 \pm 1.0$ | $48.0 \pm 1.0$ | $48.0 \pm 1.0$ | $48.0 \pm 1.0$ |
| ERM$^+$ | $45.3 \pm 1.0$ | $45.3 \pm 1.0$ | $45.3 \pm 1.0$ | $45.3 \pm 1.0$ | $45.3 \pm 1.0$ |
| CIFAR10-C | 0 | 25 | 50 | 75 | 100 |
| ARM | $39.3 \pm 1.7$ | $39.3 \pm 1.7$ | $39.4 \pm 1.7$ | $39.3 \pm 1.7$ | $39.4 \pm 1.7$ |
| ARM$^+$ | $24.5 \pm 0.7$ | $24.7 \pm 0.6$ | $24.8 \pm 0.5$ | $24.8 \pm 0.5$ | $24.8 \pm 0.6$ |
| ERM | $39.8 \pm 2.5$ | $39.8 \pm 2.5$ | $39.8 \pm 2.5$ | $39.8 \pm 2.5$ | $39.8 \pm 2.5$ |
| ERM$^+$ | $40.4 \pm 1.8$ | $40.4 \pm 1.8$ | $40.4 \pm 1.8$ | $40.4 \pm 1.8$ | $40.4 \pm 1.8$ |

### D.3 COMPARISON OF ICRM WITH IN-CONTEXT LEARNING

The first and most popular conception of In-context learning (Brown et al., 2020) involves providing the model with contextual information, typically a few sample $(x, y)$ pairs that represent a specific "task". In contrast, our approach ICRM introduces an alternative perspective on in-context learning, where unlabeled inputs $x$ act as the contextual backdrop for a task, also known as an environment. Note that, in order to benefit from in-context learning in domain generalization, the context itself must be a sequence of unlabeled inputs since at test-time the learner only has access to the unlabeled $x$'s from the test environment and not the labels $y$'s.

Our method, ICRM, can seamlessly adapt to supervised settings, functioning with input sequences containing both $(x, y)$ pairs, instead of just $x$. We refer to this approach as Supervised ICRM or ICL. We evaluate both Supervised ICRM and ICRM on FEMNIST and Rotated MNIST datasets. As anticipated, Supervised ICRM demonstrates superior performance compared to ICRM. However, it is not suitable for domain generalization settings where data labels are unavailable at inference.

Table 13: Average/worst OOD test accuracy for different context lengths, for ICRM and Supervised ICRM on FEMNIST and Rotated MNIST. Supervised ICRM refers to ICRM trained on labeled input sequences containing $(x, y)$ pairs as context.

| Data / method | Average test accuracy | | | | | Worst case test accuracy | | | | |
|---|---|---|---|---|---|---|---|---|---|---|
| FEMNIST | 0 | 25 | 50 | 75 | 100 | 0 | 25 | 50 | 75 | 100 |
| ICRM | 78.7 | 87.2 | 87.4 | 87.5 | 87.8 | 59.8 | 69.3 | 70.6 | 70.6 | 70.6 |
| Supervised ICRM | **79.0** | **87.8** | **87.7** | **88.2** | **87.9** | **61.2** | **72.2** | **73.5** | **74.5** | **74.9** |
| Rotated MNIST | 0 | 25 | 50 | 75 | 100 | 0 | 25 | 50 | 75 | 100 |
| ICRM | **93.6** | 96.1 | 96.2 | 96.2 | 96.2 | **82.5** | 88.5 | 88.5 | 88.8 | 88.8 |
| Supervised ICRM | 93.3 | **96.3** | **96.3** | **96.3** | **96.3** | 82.0 | **89.0** | **89.0** | **89.1** | **89.3** |

### D.4 INVESTIGATING THE FEATURES LEARNED BY ICRM

Figure 2 presents attention maps for two randomly sampled sequences for FEMNIST and Tinu ImageNet-C datasets. Figure 6 shows similar visualization for two other random sequences. Particularly in the second row, attention is predominantly allocated to lines of length similar to that of the query (also in green), thereby disregarding shorter lines (shown in red). Similarly, the third row shows that the model, when presented with a query image of a "train", attends not only on other trains but also on a "bus"—indicating a semantic understanding of similarity.

The key takeaway from this visualization is that ICRM effectively learns to attend to a select few samples in an input sequence. Interestingly, these samples either belong to the *same class* or exhibit *similar features*, despite potentially belonging to different classes.

To gain deeper insights into the features learned by ICRM, we examine its capability to transition from broad domain indices to nuanced, compositional contextual descriptions of environments. This analysis is crucial for understanding how ICRM facilitates amortization across similar environments. In particular, we extract embeddings from the penultimate layer of our trained model, for data from every environment within the training set. Subsequently, we train a linear classifier to predict the corresponding environment from each embedding. We repeat the experiment for models trained using both ICRM and ICRM-Mix.

As illustrated in Figure 7, the linear model using embeddings from ICRM attains an accuracy of up to 75% on FEMNIST and 98% on Rotated MNIST. This efficacy suggests that ICRM embeds representations that are linearly separable with respect to their environmental origins. Additionally, ICRM not only exhibits superior accuracy but also a faster rate of convergence in comparison to ICRM-Mix. This advantage is likely due to ICRM's i.i.d data sampling from each environment. In

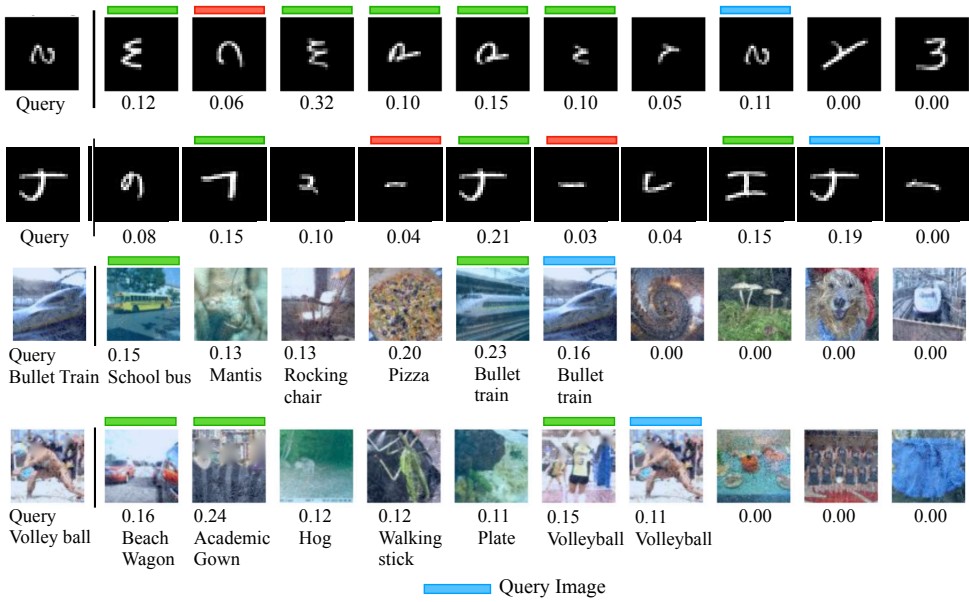

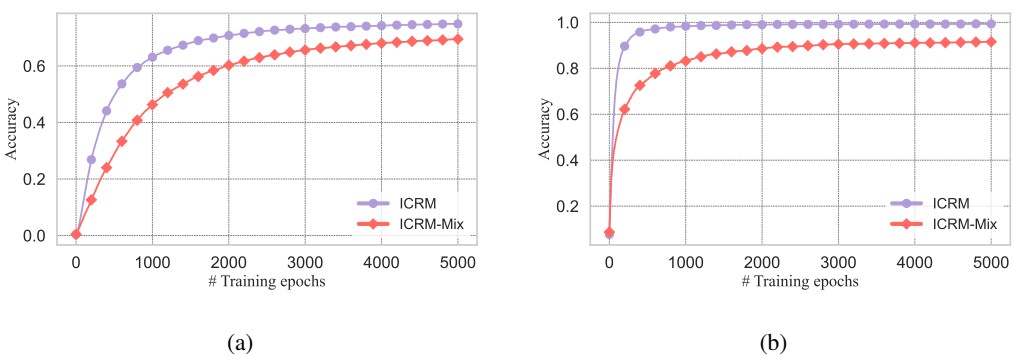

Figure 6: Attention scores for random test sequences, for ICRM on FEMNIST (top two rows) and Tiny ImageNet-C (bottom two rows).

contrast, a linear model trained on embeddings from a ICRM-Mix model also achieves significant accuracy, reaching 70% on FEMNIST and 91% on Rotated MNIST. This further explains ICRM-Mix's robust performance in out-of-distribution generalization even in the absence of explicit domain separation during training, as analyzed in Appendix D.2.2.

Figure 7: Evolution of the classification accuracy of the linear model trained on the retrieved embeddings as a function of training epochs for (a) FEMNIST (b) Rotated MNIST

