# OpenReview forum: "Context is Environment"
_ICLR.cc/2024/Conference — ICLR 2024 poster_

### Official Review · Reviewer_cBZm · 2023-10-27

**Soundness:** 2 fair
**Presentation:** 3 good
**Contribution:** 2 fair
**Rating:** 6
**Confidence:** 4

**Summary:**

The authors propose a domain generalization (DG) method called In-Context Risk Minimization (ICRM).
ICRM uses a transformer architecture that takes multiple datapoints as input and predicts their labels.
In DG, inputs come from different environments for which the distribution of inputs and labels change.
DG assumes that the environment index is known for each datapoint at training and test time, i.e. we can group datapoints by their environment.
In ICRM, each set of 'in-context' inputs belongs to the same environment.
At test time, this allows ICRM to learn _in-context_ to adapt to a particular set of test inputs.
The authors argue this is advantageous to previous work in DG, which either learns models that aggregate test points into a single embedding or learn invariant predictors that ignore dependencies between test observations.

The authors provide theoretical evidence that ICRM can achieve better loss in iid and ood settings than naive context-free empirical risk minimization.
They provide experiments on FEMNIST, Rotated MNIST, WILDS Camelyon17, and Tiny ImageNet-C where they show ICRM outperforms the following methods: ARM, TENT, and ERM.

**Strengths:**

I am convinced that ICRM is advantageous to methods that do not rely on context or aggregate all context points into a single embedding.

The linear regression toy example they provide nicely illustrates the advantages of ICRM over invariant methods.

The paper is largely well-written and the description of the ICRM method is clear.

**Weaknesses:**

Unfortunately,  I cannot recommend acceptance of the submission in its current state.

A) My biggest concern is the strong resemblance the proposed approach bears to neural processes (NPs) (Garnelo et al., 2019), and in particular attentive neural processes NPs (Kim et al., 2019); ANPs), as well as other related work, e.g. from Müller et al. (2022) and Kossen et al. (2021).  This line of related work is not discussed at all in the current draft. Chiefly, these works are related because, excluding the original NP paper, they all propose transformer architectures that process multiple datapoints at the same time to improve predictions. Neural Processes are framed around 'function learning', where 'functions' are analogous to 'environments' for ICRM. For a set of context points $\{(x_i, y_i)\}_{i\in\mathcal{C}}$ drawn from a 'function' they make predictions for the labels $y_t$ of test points $x_t$, $x\in \mathcal{T}$ of that function. In the attentive neural process, attention is used to predict test labels by attending to the context points – strikingly similar to ICRM!

I would strongly encourage the authors to highlight ANPs (and related work) as important prior work, and dial down claims of novelty when appropriate.
I do believe that the application of ANP-like architectures to domain generalization is interesting and sufficiently novel, however, the current omission of relevant prior work is not acceptable.

Note that there are some differences between prior work and ICRM.
For ANPs, the $x_i$ are often single features, e.g. the pixels of an image or (multi-dimensional) observations drawn from a Gaussian Process.
However,  ICRM itself does not process high-dimensional inputs itself, and instead relies on ConvNet/ResNet-50 embeddings: these could also be used as pre-processing with ANPs.
(Note that Neural Processes have been scaled to image classification before by Requeima et al. (2019))

B) Perhaps the biggest difference between prior work and ICRM is, that ICRM does not use labels of examples as input.  This could be a restriction of ICRM: it can only model p(X|C) of the context. If there are any shifts in p(Y|X, C) between environments, ICRM cannot model these. For ANPs, this is not a problem, as inputs are (x, y) pairs. I would argue, ANPs are a generalisation of the proposed ICRM architecture.
Does domain generalization always preclude access to (even a few) of the test environment labels? Relatedly, does DG always assume that p(Y|X, E) does not shift? Even if both of these are true,  ANPs can still be used without test-context labels as these can be iteratively predicted. Conversely, if either is not true, then it seems like ANP-like architectures are clearly advantageous.


* Kim, Hyunjik, et al. "Attentive neural processes."  ICLR (2019). https://openreview.net/forum?id=SkE6PjC9KX.
* Garnelo, Marta, et al. "Neural processes." ICML 2018 workshop on Theoretical Foundations and Applications of Deep Generative Models. https://arxiv.org/abs/1807.01622
* Müller, Samuel, et al. "Transformers can do bayesian inference." ICLR (2022). https://openreview.net/forum?id=KSugKcbNf9.
* Kossen, Jannik, et al. "Self-attention between datapoints: Going beyond individual input-output pairs in deep learning." NeurIPS (2021). https://openreview.net/forum?id=wRXzOa2z5T.
* Requeima, James, et al. "Fast and flexible multi-task classification using conditional neural adaptive processes." NeurIPS (2019).


C) My second biggest concern is with the results of the experimental evaluation. Figure 5 and Table 2 show that ICRM does not consistently improve performance when increasing the number of in-context demonstrations. In particular, for WILDS Camelyon17 and Tiny ImageNet-C the results are concerning. For Tiny ImageNet-C performance only improves very slightly when including additional in-context observations: crucially, the zero-shot ICRM performance outperforms all competing methods at _any_ number of in-context observations. Most concerning is WILDS Camelyon17: here, performance _decreases_ when including in-context examples – performance is maximal when the context is empty!

It seems to me as though the main selling point of ICRM, improving performance by including test-time examples in the context, only comes true for FEMNIST. The benefits of including datapoints in the context with ICRM are not as large as the authors proclaim. I did not see any discussion of this in the paper, and would ask the authors to provide one.

D) Thanks for providing the ERM+ baseline, which uses "an identical architecture to ICRM, but without context". Comparing the numbers of Tables 3 and 2, I was surprised to see that ICRM at context size 0 significantly outperforms ERM+ across tasks. Do you have any insights as to why this would be the case? What differences are left between ICRM and ERM+?

E) "[...] even in the absence of test context. Specifically for both WILDS Camelyon17 and Tiny ImageNet-C, ICRM outperforms baselines despite not leveraging any context from the test environment. This is because ICRM training still benefits from contexts as to find contextual features that ERM ignores" --> This is unclear to me – how can ICRM benefit from context if there is no context? Why does ICRM outperform the other baselines in the zero-shot setting?

F) I think your definition of 'in-context learning' is misleading throughout the draft and a significant departure from the meaning of in-context learning (ICL) as introduced by Brown et al. (2020). This is a recurrent problem throughout the submission. For one, ICRM is about learning from _unlabelled_ examples. ICL is about learning novel tasks from a few-shot set of _labelled_ examples. For example, the 'poem' example the authors give on page 4 is inappropriate: neither does it provide multiple examples, nor does it provide labels. A more appropriate example for ICL for poem writing would be a set of input-output pairs, e.g. a list of target audiences and matching poems for them.

G) "The key to our answer resides in a recently discovered emergent ability of next-token predictors, namely, in-context learning."  I agree with you that an interesting aspect of ICL in LLMs is that it seems to 'emerge', at least the model is not knowingly trained to learn ICL. However, for ICRM, you _train_ the model to rely on the context, i.e. very similar to the setup of Neural Processes that predates ICL. Thus, I find it highly misleading to suggest ICRM relies on any emergent abilities.  Further, ICRM does _not_ preform auto-regressive next-token prediction: your inputs are images and your outputs are labels, opposed to input tokens at time t and output tokens at time t+1. One could even argue that what ICRM does, does not meet common expectations for in-context learning, as there are no labelled in-context examples.

H) You frequently write that ICRM can  "fully exploit data in natural order". Are you using causal attention or positional embeddings? If not,  your Transformer architecture should be equivariant to the order of the inputs. Further, the prediction on the test input should then be _invariant_ to the order of the context examples. I therefore find your statements about 'nature does not shuffle data' misleading. In fact, I believe that, often, the order of the data should not affect predictions (cf. exchangeability). On the other hand, if you _are_ using causal attention or positional embeddings (which are part of the GPT-2 default architecture), I believe these might negatively affect your performance, and would like to see an ablation with them removed.

I) "The maximum context length, or support, is fixed at 100 for all algorithms" → Do you sample uniformly between [0, 100] examples at training time, or do you train different ICRM models, one for each context length? Relatedly, you write that you train your model to minimize the 'auto-regressive loss'. Does this mean that, for each training input, you minimize the predictive loss for each of the training inputs, i.e. you perform 'teacher forcing', just like in LLM training?

J) Your highlighting of low and high attention scores in Figure 2 seems arbitrary. The caption suggests you highlight all low/high attention scores. In addition to not explaining how you decide what is a low/high attention score, you also clearly miss some low/high scores, e.g. 0.00, 0.00, and 0.05 in row 1 should be marked as low if 0.11 is, and 0.20 in row3 should be high if 0.15 is. It seems like you highlight only those scores that fit your narrative, e.g. the pizza in row three is not highlighted, but the train and bus matching the query 'train' are. I would suggest you highlight scores consistently. Alternatively, make clear you highlight only some examples: those which you discuss in the main text.

K) Why is the performance on WILDS Camelyon17 the same for worst and average case? Does this mean the performance is exactly equal on all environments? Why is this the case?

## Nits

L) "We attain significant out- of-distribution abilities in a multitude of specific tasks —such as writing poems" --> Why are you sure that writing poems out of distribution for large language models – even if conditioned on some context?

M) Figure 1b): It seems like it should be $x_i^e$ instead of $x_{i,1}^e$?

N) The sentence before Eq. 5 does not make sense to me. Perhaps move the 'estimates the conditional expectation P (Y | X, C)' to after the definition in equation 5?

**Questions:**

O) "Define δe to be a permutation of γe that swaps its two components" → What are the two components (it seems to me there are three?) and what does it mean to swap them?

P) How do the computational costs of ICRM and the baselines compare?

Q) "minimizing the worst risk across" → Later you write that you sample environments at random during training. Does that not mean that you are actually minimizing the average risk across environments?

R) "to minimize the auto-regressive loss" --> Why is this loss auto-regressive? It does not depend on the model's past predictions.

---

> ### Author Response · Authors · 2023-11-15
> **Response to the Reviewer cBZm**
>
> We thank the reviewer for their detailed, thoughtful, and critical review.  We are glad you appreciated the novelty of our framework and its benefits, and we believe this could form the basis for a positive review. We believe that there are a few confusions that have resulted in the low rating. We have tried our best to clarify them below and in the paper. The relevant changes are highlighted in magenta in the main body and the Appendix.
>
> ---
>
> > On Neural processes and attentive neural processes
>
> There are a few important points worth emphasizing:
> - The key challenge of domain generalization is to build learners that succeed at test time in new situations without any access to the labeled data. If we have access to labeled data, it falls in the realm of few-shot learning, an important, related, and complementary area of research. We hope that the reviewer appreciates the challenge of domain generalization.
>
> - We agree that the existing works on NPs, ANPs, which have now been cited in the paper, and the more recent proposals studying ICL have similar architectures. However, as rightly pointed out in the review, there is a critical distinction – they all rely on labeled data in the context. Our work explores the potential of these context-based transformer architectures in the unlabeled data setting. We believe this is an important departure that embraces the essence of domain generalization. As mentioned in the review, this endeavor is deemed sufficiently novel. That said, our method is not constrained to operate on solely $x$’s. As shown below and in Section D.4, ICRM can be easily adapted to training on labeled sequences. However, this approach (Supervised ICRM) is unsuitable for domain generalization settings where data labels are unavailable at inference.
>
> Dataset |  |  | Avg.* |  |  |  |  | WG.*  |  |  |
> |----------|----------|----------|----------|----------|----------|----------|----------|----------|----------|----------|
> **FEMNIST** | 0 | 25 | 50 | 75 | 100 | 0 | 25 | 50 | 75 | 100 |
> ICRM       |  78.7  | 87.2| 87.4 | 87.5|  87.8|  59.8| 69.3 | 70.6 | 70.6| 70.6
> Supervised ICRM | **79.0** | **87.8** | **87.7** | **88.2** | **87.9** |**61.2** | **72.2** | **73.5** | **74.5** | **74.9** |
> | | | | | | | | | |
> **Rotated MNIST** | 0 | 25 | 50 | 75 | 100 | 0 | 25 | 50 | 75 | 100 |
> ICRM       |  **93.6** |   96.1|  96.2| 96.2 | 96.2|  **82.5**|  88.5|  88.5| 88.8| 88.8
> Supervised ICRM | 93.3 | **96.3** | **96.3** |**96.3** | **96.3** | 82.0 | **89.0** | **89.0** | **89.1** | **89.3** |
>
> *Here “Avg.” and “WG.” respectively denote the average and the worst group accuracy across test environments.
>
> - Our work also lays down the theory that shows when unlabeled data can help the trained models perform better than standard ERM in different settings – short and long contexts and in and out-of-distribution evaluations. The theoretical analysis in this setting poses a different challenge than labeled data. This is another reason we strongly believe our work should not be viewed as a special case of ANPs. The theory from existing works would not translate and explain the success of ICRM.
>
> ---
>
> > On the gains achieved using unlabeled examples
>
> We would like to highlight three key points here. Firstly, in three of the four datasets, we observe notable improvements through adaptation. Please also refer to the worst-group accuracy in Table 2. These datasets are particularly challenging, and even a 1-2 percent gain is significant. Secondly, Theorem 2 characterizes scenarios under which increasing unlabeled context can monotonically improve the performance of the trained model. Thirdly, we present results with comprehensive hyperparameter tuning for all baselines for two additional datasets, Imagenet-R and CIFAR 10 C, demonstrating the advantages of incorporating context.
>
> Dataset |  |  | Avg.* |  |  |  |  |  | WG.*  |  |  |
> |----------|----------|----------|----------|----------|----------|----------|----------|----------|----------|----------|----------|
> CIFAR10-C | 0 | 25 | 50 | 75 | 100 | | 0 | 25 | 50 | 75 | 100 |
> ARM   | 65.9 | 66.0 | 66.0 | 66.0 | 66.0  | | 39.3 | 39.3 | 39.4 | 39.3 | 39.4
> ERM   | 66.1 | 66.1 | 66.1 | 66.1 | 66.1 | | 39.8 | 39.8  | 39.8  | 39.8  | 39.8
> ICRM  | **70.6** | **71.9** |  **71.9** | **71.9** | **71.9** | | **54.6** | **56.0** | **55.8** | **55.8** | **55.9**
> |  |  |  |  |  |  |  |  |  |  |
> ImageNet R | 0 | 25 | 50 | 75 | 100 | | 0 | 25 | 50 | 75 | 100 |
> ARM   | 56.3 | 58.1 | 58.8 | **59.8** | 59.0  | | 47.4 | 45.3 | 47.2 | **49.8** | 47.4
> ERM   | **58.9** | 58.9 | 58.9 | 58.9 | 58.9 | |**48.0** | **48.0**  | **48.0**  | 48.0  | 48.0
> ICRM   | 57.4 | **59.7** |  **59.6** | 59.4 | **60.5** | | 45.4 | **48.0** | 47.2 | 46.9 | **50.6**
>
> *Here, “Avg.” and “WG.” respectively denote the average and the worst group accuracy across test environments.

---

> ### Author Response · Authors · 2023-11-15
> **Response to the Reviewer cBZm (Continued.)**
>
> > Performance decreases with context in WILDS Camelyon17.
>
> In the Camelyon dataset, standard ERM often encounters a failure mode where the model exploits shortcut features such as luminosity. Consider a model that only takes the current query as input but has an effective featurizer that eliminates luminosity. Such a model can perform well on Camelyon-17. We posit that ICRM learns a similar featurizer by observing context during training. Consequently, the ICRM model has to learn to ignore additional queries provided during test time.  In some cases, these additional queries may have acted as distractors, causing a 1.2 percent drop in performance.
>
>
> ---
>
>
> > On the ERM+ baseline
>
> ERM+ only operates on the current query and there is no context input to it. However, ICRM accepts the current query and context as input. At test time during zero shot evaluation, it is true that both ICRM and ERM+ do not have any context. However, during train time ICRM is trained with contexts and that can lead to a model that is better than ERM. The context allows the model to selectively augment the current query and train a better featurizer as also explained in the previous answer.
>
> ---
>
> > On the definition of in-context learning
>
> The first and most popular conception of In-context learning [1]  involves providing the model with contextual information, typically a few sample $(x, y)$ pairs that represent a specific “task”. While it is common to leverage additional information through labeled data, it is also possible to use unlabeled data to provide additional information, as demonstrated by our work. As shown in [2], labels do not play a significant role in several in-context learning tasks.
>
> [1] Brown, Tom, et al. "Language models are few-shot learners." Advances in neural information processing systems 33 (2020): 1877-1901.
>
> [2] Work, What Makes In-Context Learning. "Rethinking the Role of Demonstrations: What Makes In-Context Learning Work?."
>
> ---
>
>
> > On the role of emergence
>
> There appears to be some confusion here. We are referring to the emergent ability of LLMs, and nowhere do we claim that this ability is emergent for our method
>
> ---
>
> > On the role of natural order
>
> We employ both causal attention and positional embeddings. Causal attention is used to allow the trained model to only make predictions by attending to the observed data thus far. It is also a popular choice to study in-context learning [1]. In our current setup, we randomly sample data sequences from a domain. However, there is a partial order imposed by the environments, which we have explained in greater detail in the revised manuscript. Our intention was to convey that if the data had been collected with a natural order, our approach can accommodate it by processing the inputs accordingly.
>
> As the review correctly highlights, using positional encodings for our datasets, where predictions are invariant to the order of data in a sequence, may adversely affect performance. The table below illustrates the performance of ICRM with and without positional embeddings. Clearly, both the model achieves similar performance access benchmarks with an exception of FEMNIST, where the model without positional embeddings exhibits 5\% gains in the worst group accuracy.
>
> Dataset |  |  | Avg.* |  |  |  |  | WG.*  |  |  |
> |----------|----------|----------|----------|----------|----------|----------|----------|----------|----------|----------|
> **FEMNIST** | 0 | 25 | 50 | 75 | 100 | 0 | 25 | 50 | 75 | 100 |
> ICRM (with pos) | 78.4 | 86.6 | **87.3** | 87.0 | 87.3 | **56.0** | **72.3** | 69.3 | 70.8 | 70.0
> ICRM (w/o pos) | **79.1** | **86.9** | 86.6 | **87.4** | **88.0** | 55.0 | 70.0 | **71.0** | **72.0** | **75.0**
> | | | | | | | | | |
> **Rotated MNIST** | 0 | 25 | 50 | 75 | 100 | 0 | 25 | 50 | 75 | 100 |
> ICRM (with pos) | **92.7** | **95.8** | **95.7** | **95.8** | **95.9** | **81.1** | **87.8** | **87.0** | **87.5** | **87.8**
> ICRM (w/o pos) | 92.1 | 95.3 | 95.5 | 95.5 | 95.6 | 80.1 | 86.0 | 86.3 | 86.8 | 86.9
> | | | | | | | | | |
> **Tiny ImageNet-C** | 0 | 25 | 50 | 75 | 100 | 0 | 25 | 50 | 75 | 100 |
> ICRM (with pos) | 38.3 |  39.2 |  39.2 |  39.2 | 39.2 | 18.8  | 19.2  |  19.5 | 19.5 | 19.4
> ICRM (w/o pos) | **38.8** | **39.7** | **39.7** | **39.7** | **39.7** | **19.3** | **20.5** | **20.5** | **20.5** | **20.4**
>
> *Here “Avg.” and “WG.” respectively denote the average and the worst group accuracy across test environments. These reported numbers are not obtained through a rigorous hyper-parameter sweep.
>
> [1] Garg, Shivam, et al. "What can transformers learn in-context? a case study of simple function classes." Advances in Neural Information Processing Systems 35 (2022): 30583-30598.

---

> > ### Author Response · Authors · 2023-11-15
> > **Response to the Reviewer cBZm (Continued.)**
> >
> > > On training of ICRM models with maximum context length 100
> >
> > Thanks to the attention in transformer models, we can obtain predictions for all samples in a sequence of length 100 in a single forward pass. Consequently, during each iteration, we randomly select a training sequence (of length 100) and generate predictions for each example with just one forward pass through a single ICRM model.
> >
> >
> > ---
> >
> >
> > > Clarification about autoregressive loss
> >
> > We apologize for the incorrect usage of the word ‘autoregressive’ here. By autoregressive loss, we meant to refer to the loss of our autoregressive model. We have corrected this in the revised manuscript. As discussed in Section 4, we minimize the expected loss over predictions using all the context lengths. Denote context of length $j$ as $c^e_{j+1} = (x^e_1, \ldots, x^e_{j})$, for all $j = 1, \ldots, t-1$, and $c^e_1 = \emptyset$. Then mathematically, the loss function is given by $\sum_{j=1}^t \ell(h(x^e_j; c^e_j), y^e_j)$.
> >
> > ---
> >
> >
> >  > Clarification on attention scores in Figure 2
> >
> > We clarify the rationale behind highlighting specific images in Figure 2:
> > - A threshold of 0.07 is chosen across sequences to distinguish high and low scores.
> > - Images following the query have scores of 0.0 due to causal attention, where the model ignores future tokens. Thus, highlighting such inputs doesn’t add value to the observations.
> > - Rather than categorizing all image scores as high/low, we aim to uncover non-trivial patterns learned by ICRM. We intentionally avoid highlighting images we expect the model to pay little attention to, like an image with a score of 0.05 in row one. Instead, we emphasize non-trivial attention, such as the model paying low attention to images with a atmost one curved arc in row one (marked in red), considering the query image contains two curved arcs.
> >
> > ---
> >
> >
> > > Same average and worst group test accuracy on WILDS Camelyon17
> >
> > As mentioned in the original submission, WILDS Camelyon17 comprises data from 5 hospitals, with 3 for training, 1 for validation, and 1 for testing. With only one testing environment, the average performance is equivalent to the worst-case performance across testing domains.
> >
> > ---
> >
> >
> > >  Clarifying the line on swapping two components
> >
> > We mean it swaps parameter tuples corresponding to $y=0$ and $y=1$.
> >
> > ---
> >
> >
> > > Computational costs of ICRM
> >
> > Training an ICRM model on FEMNIST and Rotated MNIST typically consumes around 6 GB and 10 GB of GPU memory, respectively, and takes roughly 1-2 hours for training. The exact time varies depending on the context length and batch size for our model architecture with 12 layers, 128 dimensions, and 4 heads.
> > In contrast, the ERM baseline on these datasets uses approximately 4 GB and 5 GB, respectively, and takes around 30 minutes - 1 hour to complete.  Time and space complexity of other baselines may vary.
> >
> > *Reason for time efficiency of ICRM.*  ICRM makes predictions for all samples in a sequence of length $K$ in one forward pass. Thanks to the efficient PyTorch implementation of transformers, caching intermediate hidden states, typically the outputs of attention layers, significantly accelerates inference.
> >
> > ---
> >
> >
> > > Clarification of the kind of error is being minimized, the worst group or average error?
> >
> > ICRM is trained to minimize the average error across training environments. Equation 2 refers to the min-max formulation of the optimal predictor in the domain generalization setting.
> >
> > ---
> >
> >
> > > Minor clarifications and typographical errors
> >
> > 1. _Regarding the poem example._ We have replaced "we" in the sentence with “LLMs”.
> > 2. _Typographical error in Figure 1b._ We thank the reviewer for their attention to detail. We have made the correction in the revised manuscript.
> > 3. _Sentence before equation 5._ We have now corrected this in the revised manuscript by moving the sentence to after equation

---

> > > ### Comment · Reviewer_cBZm · 2023-11-16
> > > **Reviewer Response to Rebuttal**
> > >
> > > I thank the authors for their detailed response to my review, which have clarified a number of my concerns. I will increase my score to a 5 but will consider a further score increase if they can clear up my remaining concerns.
> > >
> > >
> > > Before moving on to technical points, I would just like to note that I find your response hard to parse. It would have been great if you could have used the letters (a-z) that I used to enumerate the points in my original review.
> > >
> > > > On Neural processes and attentive neural processes
> > >
> > > Thanks for your clarifications regarding domain generalization and for adding a discussion of neural processes to the paper.
> > >
> > >
> > > > On training of ICRM models with maximum context length 100
> > >
> > > Thanks for your reply here!
> > >
> > > Can you provide ICRMs performance at all context sizes between 1 – 25 for FEMNIST, Rotated MNIST, and ImageNet-C? Given that you obtain predictions at all context sizes automatically, I would hope this is computationally doable for you.
> > >
> > > ICRM performance often stays relatively constant between 25-100 examples but there is a big jump when going from 0-25 examples. I would like to see the behavior in that region in more detail.
> > >
> > > Apologies for asking for another experiment. If you can provide these numbers, I will consider an increase in score.
> > >
> > >
> > > > Consequently, the ICRM model has to learn to ignore additional queries provided during test time. In some cases, these additional queries may have acted as distractors, causing a 1.2 percent drop in performance.
> > >
> > > If ignoring queries works at training time, why would it cause a drop of performance at test time?
> > >
> > > > On the definition of in-context learning
> > >
> > > I nevertheless think it would be good to highlight this difference in between your setup and conventional in-context learning in LLMs.
> > > This really is a bit tangential, but the results of [2] do not really seem to hold up to scrutiny, see https://arxiv.org/abs/2307.12375.
> > >
> > >
> > > I also still think your poem example is not representative of in-context learning in LLMs.
> > >
> > >
> > > > On the role of natural order
> > >
> > >
> > > Thanks for the additional experiments.
> > >
> > > Why are the numbers for ICRM (with pos) not the same as the ICRM numbers in the main paper? Is this variance from different random seed?
> > >
> > > I find the results for rotated MNIST particularly interesting. Do the inputs have a natural order here?
> > >
> > > Do inputs have a natural order for any of your experiments?
> > >
> > > If not, then I find your repeated highlighting of this somewhat inappropriate/misleading.
> > >
> > >
> > > > Images following the query have scores of 0.0 due to causal attention, where the model ignores future tokens. Thus, highlighting such inputs doesn’t add value to the observations.
> > >
> > > That makes sense. Thanks for explaining. I think you should make clear that attention scores here are zero due to causal attention. Or maybe just not show images following the query at all. (Or maybe add arrows to illustrate the direction of attention.)
> > >
> > >
> > > > A threshold of 0.07 is chosen across sequences to distinguish high and low scores.
> > >
> > > I think you should mention this. Also, why 0.07?
> > >
> > > > Rather than categorizing all image scores as high/low, we aim to uncover non-trivial patterns learned by ICRM.
> > >
> > > For the 'volleyball' query, why is high attention to 'academic gown' more interesting than high attention to 'hog'?
> > >
> > > I find your labelling here arbitrary and you should _at least_ clarify that you are making a decisions which high/low points you highlight.
> > >
> > > > However, there is a partial order imposed by the environments, which we have explained in greater detail in the revised manuscript.
> > >
> > >
> > > I cannot find any information on 'partial order' in the draft. Could you point me to the relevant section?

---

> > > > ### Author Response · Authors · 2023-11-16
> > > > **Response to the additional review by Reviewer cBZm**
> > > >
> > > > We appreciate the reviewer's prompt response and their decision to raise their score to a 5. We are glad that our initial response addressed most of their concerns. We apologize if our previous response was difficult to understand. Below, we have endeavored to address their remaining concerns chronologically, as concretely as possible.
> > > >
> > > > ---
> > > >
> > > > > Performance of ICRM models between context lengths 1 to 25.
> > > >
> > > > Thank you for this valuable suggestion. In Figure 5 of Section D.1 in the Appendix, we present the performance of ICRM, ARM, ERM, and TENT across various context lengths ranging from 0 to 50. The evaluation includes both the worst group accuracy and the average accuracy across testing environments for datasets FEMNIST, Rotated MNIST, WILDS Camelyon17, and Tiny Imagenet-C.
> > > >
> > > > ---
> > > >
> > > >
> > > > > If ignoring queries for Camelyon17 works at training time, why would it cause a drop in performance at test time?
> > > >
> > > > Ignoring examples may work well during training, but it can still result in the model inadvertently attending to some examples when presented with out-of-distribution data.
> > > >
> > > > ---
> > > >
> > > > > On the definition of in-context learning
> > > >
> > > > - We appreciate the reviewer's feedback and would like to mention that we have already included a discussion on this topic in Section D.3 of the revised manuscript. We also added additional clarification to Section 3, and the related works in Section B.
> > > >
> > > > - Regarding the poem’s example, we have now clarified that it illustrates in-context learning (ICL) operating on _unlabeled_ contexts and still exhibiting amortization capabilities that achieve significant OOD generalization. Additionally, we have added another example in Section 3 following the conventional definition of ICL.
> > > >
> > > > ---
> > > >
> > > >
> > > > > On the role of natural order
> > > >
> > > > **Regarding the reported numbers for with and w/o positional encodings**
> > > >
> > > > - The comparison for ICRM, with and without positional encodings, is carried out for the default set of hyperparameters (seed 0). For the results in the original manuscript, we reported mean and standard deviation computed across three independent runs of the entire sweep, wherein the model selected for evaluation is the one with hyper-parameters yielding the highest validation accuracy (Details in Section C.4).
> > > >
> > > > - None of the experiments covered in the paper have a natural order, including Rotated MNIST.
> > > >
> > > > **On the repeated highlighting of natural order**
> > > >
> > > > Based on the reviewer's initial review, we had already revised the manuscript to highlight the potential of our algorithm to exploit data in its natural order. This is mentioned only three times: once in Section 4 and twice in the Conclusion. Each mention pertains to future work, where we consider the application of our algorithm to address naturally occurring time-based shifts or domains naturally ordered by time (video etc.).
> > > >
> > > > ---
> > > >
> > > >
> > > > > Regarding Figure 2
> > > >
> > > > We fully concur with the reviewer's feedback and have incorporated the necessary revisions regarding causal attention in the main text of the revised manuscript, which is highlighted in magenta.
> > > >
> > > > **Why a threshold of 0.07**
> > > >
> > > > This is a heuristic. The aim is not to categorize images in a sequence with respect to high/low attention scores. However, we understand the concern raised by the reviewer. To address it, we have made changes in the figure by labeling only the query image and using green/red markings merely to explain our analysis.
> > > >
> > > > **Regarding high attention to a “hog”**
> > > >
> > > > High attention to a "hog" is intriguing but challenging to interpret due to limited image pixels. Not all decisions made by an imperfect classifier are interpretable. It is even difficult to definitively classify this as an error, as there could be features in the image of a hog that are relevant to predicting volleyball. On the other hand, the high attention to individuals wearing academic gowns is more interpretable and suggests a semantic understanding of the image.
> > > >
> > > >
> > > > ---
> > > >
> > > >
> > > > > On the partial ordering imposed by environments
> > > >
> > > > This is mentioned in Section 4 of the initially revised manuscript (in magenta), “The most natural way to construct contexts is to use past samples that appear in the natural order in which data was collected (e.g., video). Since existing DG datasets do not provide such a refined ordering, we build contexts using environment indices that are more readily available.”

---

> > > > > ### Comment · Reviewer_cBZm · 2023-11-17
> > > > > **Reviewer Response**
> > > > >
> > > > > Thanks for your quick answer!
> > > > >
> > > > > > In Figure 5 of Section D.1 in the Appendix, we present the performance of ICRM, ARM, ERM, and TENT across various context lengths ranging from 0 to 50.
> > > > >
> > > > > Thanks, the worst case results for FEMNIST are convincing: ICRM (a) starts off at similar performance as the other methods and then (b) improves progressively as more in-context examples are observed.
> > > > > This is the behavior I would expect to see to support the claim that ICRM improves over baselines because of its 'in-context learning' abilities.
> > > > > I am not convinced there is a reason ICRM should significantly outperform other baselines without any context, hence, my expectation to see (a) as well.
> > > > >
> > > > >
> > > > > However, I do not find the results on the other datasets sufficiently supportive of the authors claims that improving context at test time helps ICRM outperform baselines.
> > > > >
> > > > > For rotated MNIST, (a) is the case but (b) is much less clear, as there is really only one jump in performance from 0-5 examples (this jump could also be at one in-context example).
> > > > >
> > > > > For Camelyon, (a) and (b) are not the case.
> > > > > For Tiny ImageNet-C, (a) is not the case and the effect (b) is very small, perhaps insignificantly so.
> > > > >
> > > > > Do you explain somewhere how the standard deviations are calculated? Are they just over the three seeds (in which case, I think you should plot min/max instead as three is not an appropriate number of samples to compute a standard deviation).  Also, for the average test accuracy, it would also be important to consider the combined standard error over tasks seeds. I believe this would be the uncertainty to consider when evaluating if ICRM outperforms the baselines.
> > > > >
> > > > > Do you also average over different orders of including the in-context examples in each environment?  (If so, the variance over this would also be relevant.)
> > > > >
> > > > >
> > > > > I'm happy to discuss the above points further. Until now though, I would say there is only one dataset where ICRM clearly delivers on its promise, which is not enough for me to meet the bar of acceptance.
> > > > >
> > > > >
> > > > > > High attention to a "hog" is intriguing but challenging to interpret due to limited image pixels. Not all decisions made by an imperfect classifier are interpretable. It is even difficult to definitively classify this as an error, as there could be features in the image of a hog that are relevant to predicting volleyball. On the other hand, the high attention to individuals wearing academic gowns is more interpretable and suggests a semantic understanding of the image.
> > > > >
> > > > >
> > > > > I still don't buy this. Why would attention to an academic gown be more interpretable than attention to the hog, a walking stick, or a plate? I think this interpretation of your figure is cherry picked.
> > > > >
> > > > >
> > > > > > [All other points]
> > > > >
> > > > > Thanks for your answers and actions here. They alleviate my concerns.

---

> > > > > > ### Author Response · Authors · 2023-11-18
> > > > > > **Response to the latest reply by Reviewer cBZm**
> > > > > >
> > > > > > We thank you for the prompt responses and the discussion. ​​We appreciate the opportunity to clarify and assert key aspects of our work, addressing the concerns raised. We believe there are some major misunderstandings that, once clarified, will help the reviewer see the significance and robustness of our findings.
> > > > > >
> > > > > >
> > > > > > In their initial response, the reviewer stated that "the zero-shot ICRM outperforms all competing methods at any number of in-context observations." However, in their subsequent response, they mentioned that "I would say there is only one dataset where ICRM clearly delivers on its promise, which is not enough for me to meet the bar of acceptance." They also raised an issue with the fact that most of the gain in Rotated MNIST can occur even with just one in-context example.
> > > > > > These responses have led us to believe that their evaluations primarily focus on in-context gains with a sufficient number of examples while potentially overlooking zero-shot improvement and improvement with very few examples.
> > > > > >
> > > > > > We want to make it clear that a) in-context gains – gain observed after observing in-context examples, even if only one example, and b) zero-shot improvement  – improvement in performance over the counterparts when the model has seen no in-context examples, are both central to domain generalization. We provide rigorous evidence for both these abilities of ICRM. We elaborate on this below.
> > > > > >
> > > > > > ---
> > > > > >
> > > > > >
> > > > > > > **On adaptation with increasing context**
> > > > > >
> > > > > > - Learning with few (even one) in-context samples is a feature of ICRM and not a flaw. In Theorem 1 in the paper, we show the benefit of ICRM in the limit of large context lengths. In Theorem 2, we show that ICRM can benefit merely from one unlabeled in-context example! So in short, it is possible (as we observe empirically) that very few examples provide most of the gain, with minimal gains as the number of examples increase more.
> > > > > >
> > > > > > - The reviewer's evaluation seems to have missed our results from two additional datasets: ImageNet-R and CIFAR 10-C. Table 8 in Appendix explicitly shows in-context gains (difference in worst group performance at 100 context length and zero context length) of **20 percent** for FEMNIST, **6 percent** for Rotated MNIST, **0.6 percent** for TinyImagenet, **5 percent** for Imagenet R, **1.3** percent for CIFAR10 C. While FEMNIST is largest, others are useful gains as well.
> > > > > >
> > > > > >
> > > > > > ---
> > > > > >
> > > > > >
> > > > > > > **On the zero-shot gains of ICRM**
> > > > > >
> > > > > > - The most natural source of gain for our method is to utilize additional information in the context. However, it can offer zero-shot benefits. We'll elaborate on this with two related descriptions.
> > > > > >
> > > > > > - Observe that while ICRM models can predict without context at test time, these have always been trained to observe contexts during training. Therefore, when the ICRM model is trained, it has the opportunity to learn to extract features from the current image while selectively paying attention to other images in the context. By attending to images that the model finds useful for prediction, ICRM can end up learning “in weights” a higher-quality featurizer. That is, ICRM models may “depend” on context to learn environmental spurious features while burning “in-weights” the invariant features across environments.  Our current theoretical analysis is mainly geared towards explaining the benefit of contextual information. We believe it is a fruitful future work to develop a theory to explain the above selective augmentation perspective.
> > > > > >
> > > > > > - The example in equation (7) and its extensions in Appendix A.6 explicitly constructs a linear example for which _ICRM exhibits better out-of-distribution performance than ERM without any contextual information at test time_. Since the model is trained over an extended feature space, it can learn the right invariances for inputs in the original feature space.
> > > > > >
> > > > > >
> > > > > > ---
> > > > > >
> > > > > >
> > > > > > > **Regarding the reported mean and standard deviation**
> > > > > >
> > > > > > For each algorithm, within each independent run, we conduct a randomized search with 5 trials spanning the hyperparameter range (refer to Table 6). We repeat this process for a total of 3 independent runs per algorithm, resulting in a total of 15 models. The mean and standard error are calculated based on the highest-performing models from each of the three runs. Our reporting approach along with extensive hyper-parameter tuning adheres to established standards in the field, as exemplified by DomainBed [1] and numerous subsequent works that have adopted similar practices. We hope that the reviewer understands that conducting a hyperparameter sweep thrice is in itself computationally demanding and resource-intensive. Further, for each seed, we observe adaptation with in-context samples and averaging captures the overall behavior.
> > > > > >
> > > > > > [1] Gulrajani, Ishaan, and David Lopez-Paz. "In search of lost domain generalization." arXiv preprint arXiv:2007.01434 (2020).
> > > > > >
> > > > > >
> > > > > > ---

---

> > > > > > > ### Author Response · Authors · 2023-11-18
> > > > > > > **Response to the latest reply by Reviewer cBZm (continued)**
> > > > > > >
> > > > > > > > **About Figure 2 and attention**
> > > > > > >
> > > > > > > We address the confusion regarding our Figure. As stated in our previous response, the image corresponding to the "academic gown" class (let's denote it as (a)) contains individuals (with blurred faces for privacy considerations) wearing the gowns. On the other hand, the query image associated with the "volleyball" class (denoted as (b)) shows an individual holding a volleyball. Model's attending to (b), to make predictions on (a) signifies the model's ability to recognize human-related features across images within a sequence.
> > > > > > >
> > > > > > > We hope this clarifies the reviewers' concerns and we appreciate their understanding.

---

> > > > > > > > ### Comment · Reviewer_cBZm · 2023-11-20
> > > > > > > > **Reviewer Response**
> > > > > > > >
> > > > > > > > Thanks for your response!
> > > > > > > >
> > > > > > > > If you say that zero-shot/(k=small)-shot gains are also important to you, it would potentially be okay with me, if the (k=large)-shot gains are not always impressive with ICRM.
> > > > > > > > However, I have to say that (a) I find this to be misleading given the current motivation and presentation of the paper focuses heavily on few-shot in-context learning gains, and (b) I remain not convinced why 0-shot predictions with ICRM should be better (see below).
> > > > > > > > Again, I appreciate your continued discussion on this topic.
> > > > > > > >
> > > > > > > >
> > > > > > > > > ImageNet-R and CIFAR 10-C. Table 8 in Appendix explicitly shows in-context gains
> > > > > > > >
> > > > > > > > Thanks for the pointer. I think the ImageNetR and CIFAR-10C results also mostly demonstrate (k=small)-shot gains, i.e. gains when going from zero-shot to few-shot, especially when considering the standard errors of the results.
> > > > > > > > The performance at 25 samples with ImageNetR is better than the performance at 75 samples. This makes me think that, either ICRM is not working as intended or the true variance is higher than you report, where I think the latter is more likely. Similarly, for CIFAR-10C you obtain the best performance at 25 samples. Importantly, performance at k=25 might be the same as performance at k=1,2,5,10,... which are not reported!
> > > > > > > >
> > > > > > > >
> > > > > > > > > The example in equation (7) and its extensions in Appendix A.6 explicitly constructs a linear example for which ICRM exhibits better out-of-distribution performance than ERM without any contextual information at test time. Since the model is trained over an extended feature space, it can learn the right invariances for inputs in the original feature space.
> > > > > > > >
> > > > > > > >
> > > > > > > > I think this example might be misleading. It _assumes_ ICRM already has access to the improved feature space. In a more realistic setup, you would actually need to show that ICRM can infer $\mu_\epsilon^2$ from the context set. For many context-points it makes sense to me that ICRM could infer this term from the context. However, how would ICRM improve zero-shot predictions here? Without any context points, it cannot infer $\mu_\epsilon^2$!
> > > > > > > >
> > > > > > > > One may even expect that there is some trade-off here where ICRM has to find weights that work well for all possible k-shot predictions. Where the optimum for the zero-shot prediction should still be the ERM optimum.
> > > > > > > > In other words, if you optimize ICRM over 100 context points, the loss is an average over all the 1--100 predictions the model makes. If the model can rely on the existence of context points for most of these points, why would it learn a model that works well when there are no context points?
> > > > > > > >
> > > > > > > > Also, would this example not be solvable by ERM if you would add, as is standard, a learnable constant term to the regression model? (I'm just saying this feels a bit artificial, even for a pedagogic example.)
> > > > > > > >
> > > > > > > >
> > > > > > > > > By attending to images that the model finds useful for prediction, ICRM can end up learning “in weights” a higher-quality featurizer. That is, ICRM models may “depend” on context to learn environmental spurious features while burning “in-weights” the invariant features across environments.
> > > > > > > >
> > > > > > > > I get the intuitive idea of 'the environment varies per context so this is what ICRM will use attention for; the invariant features do not vary per context, so this is what ICRM will use the weights for'.
> > > > > > > >
> > > > > > > > However, I do find this a little bit vague. Perhaps we can discuss this with an example?
> > > > > > > >
> > > > > > > > Why should ICRM improve 0-shot performance over related work (in particular the ERM+ method) for RotatedMNIST?
> > > > > > > >
> > > > > > > > It makes sense to me that, for multiple inputs, ICRM can infer the rotation of the environment and improve predictions. However, for zero-shot predictions, I do not see what 'higher-quality' 'invariant' features ICRM could learn that standard methods cannot learn.

---

> ### Author Response · Authors · 2023-11-20
> **Response to the latest reply by Reviewer cBZm**
>
> We thank the reviewer for their continued discussion. We hope that our clarifications below can help with the concerns they raised and help see our work in a positive light.
>
> ---
>
>
> > Regarding misleading motivations
>
> In their response, they state “I find this to be misleading given the current motivation and presentation of the paper focuses heavily on few-shot in-context learning gains”. We are not clear what they mean here. Both zero-shot and few-shot gains are at the heart of domain generalization, which is the primary focus of our paper.  We do not see how the paper is misleading. Nowhere in our paper do we assert that gains are exclusively expected with large context lengths and not attainable with small context lengths. In fact, achieving gains at small context lengths is practically significant. Our theoretical results focus on in-context gains in the limit of both large and small context lengths (even one context length), as demonstrated in Theorems 1 and 2, respectively.
>
> ---
>
>
> > Regarding the issue of performance drop in CIFAR10-C
>
> We use a toy example to explain the source of the performance drop. Consider a sequence $x_1, \cdots, x_n$ and corresponding labels $y_1, \cdots, y_n$.  Suppose the label are generated as  $y_i = f(x_{i},x_{i-1})$, where $x_i$ is the current query, $x_{i-1}$ is the previous query and $y_i$ is the label for the current query. We contrast learning a function with context length 1 and context length 2. The functions with context length 1 take as input $x_i, x_{i-1}$ and the functions with context length 2 take as input $x_i, x_{i-1}, x_{i-2}$.  Observe that the function class of context length 2 contains the function class of context length 1. As a result, the optimal predictor (in terms of the prediction error over the training distribution) in length 2 class is at least as good as that in length 1 class. However, when it comes to learning the ideal function $f(x_{i},x_{i-1})$ with a finite number of samples, learning the ideal function from length 1 class should require fewer samples than learning the ideal function from length 2 class. In other words, with a fixed number of samples, the generalization error of the length 1 class should be smaller than length 2 class.  In our experiments, these finite sample-driven learning-based issues could lead to performance drops.
>
> ---
>
>
> > Regarding the performance of CIFAR 10 C at intermediate points
>
> We would be happy to add performance at all intermediate points for CIFAR 10 C and Imagenet R. And at the risk of repetition even if the performance was the same for all $k=1, 2, …, 25$, it is acceptable and not in contradiction with desiderata or the theory.
>
> ---
>
>
> > Regarding zero shot gains
>
>  You stated, “I remain not convinced why zero-shot predictions with ICRM should be better”. We have provided empirical evidence that zero-shot gains are achievable. We have provided a solid theoretical analysis for context lengths 1 and beyond. We also provide initial insights into zero-shot gains through examples in equation (7) and the selective augmentation perspective. Transforming these initial insights into a solid full-blown theory to explain “why ICRM improves zero-shot predictions” is an important future work, which we also stated in our previous response.  We hope that the reviewer appreciates that it is not possible to explain all aspects of the experiments through the lens of theory in one paper.
>
> ---
>
>
> > Regarding the comparison of ERM vs ICRM at zero shot in the example from equation (7)
>
> In Proposition 3 in the Appendix, we showed that for the example under consideration, the error of ERM grows in variance of the first feature. In the paragraph that follows Proposition 3, we show that the error of ICRM without any context is independent of the variance of the first feature. In the absence of any context, we assume ICRM uses a fixed value of zero as an estimate for the mean.  Zero is an arbitrary choice and qualitatively the result stays the same even if some other fixed value is used. If the variance of the first feature is sufficiently high in the test environment, then the error of ERM is larger than the error of ICRM (without context).
> On the point about adding a fixed constant to ERM, this will also not work as the mean $mu_e^2$ varies across environments.
>
> ---

---

> > ### Author Response · Authors · 2023-11-20
> > **Response to the latest reply by Reviewer cBZm (continued)**
> >
> > > Regarding an example supporting a better featurizer for ICRM
> >
> > At the risk of repetition, we want to emphasize that when ICRM is trained, the weights of the featurizer are learned to minimize prediction error at different context lengths. Even though ICRM at test time may see no context, it uses the weights of the featurizer that have been influenced by contextual examples provided at training time. Let us consider the following example. Consider a certain training step, where the current query is a car and the attention layer is attending to images of other automobiles in the context. The weights of the featurizer (both Resnet-50 backbone and transformer weights) are updated taking into account the features of the current query (car) and its alignment to the other automobiles in the context. In contrast, the weight update for standard ERM will only update the weights of the model based on the current query and not consider the contextual queries. Therefore, ERM and ICRM learn different featurizer as a result. Due to attention to contextual queries, the featurizer of ICRM can be better. As stated earlier, we believe a solid theoretical analysis to show this is the case is an exciting future work.
> >
> > ---

---

> > > ### Comment · Reviewer_cBZm · 2023-11-20
> > > **Reviewer Response**
> > >
> > > Thanks for your continued engagement. I am sufficiently satisfied by your responses and will increase my score to a 6.
> > >
> > > I would encourage the authors to include some of this discussion in the updated version of the paper.
> > >
> > > For example, I think the current abstract strongly implies that the gains of ICRM are due to in-context learning ("We argue that context is environment, and posit that in-context learning holds the key to better domain generalization.")
> > > This suggests that test-time observations of environment are the reason why ICRM improves over baselines.
> > > However, half the datasets in Figure 5 show that zero-shot gains dominate the improvements obtained from ICRM.
> > > The authors convincingly explain why learned features would be different for ICRM, but not why they would be better.

---

> > > > ### Author Response · Authors · 2023-11-20
> > > > **Response to the latest reply by Reviewer cBZm**
> > > >
> > > > We sincerely appreciate the reviewer’s continued engagement with our work and their thoughtful review. Their feedback has been very valuable in refining our paper. We also thank them for their decision to raise their score.
> > > >
> > > > Based on their suggestion, we would ensure that the revised manuscript better reflects the diverse aspects of ICRM's performance, both in terms of in-context gains and zero-shot gains. We thank them again for their time and consideration

---

### Official Review · Reviewer_NswY · 2023-10-30

**Soundness:** 3 good
**Presentation:** 3 good
**Contribution:** 4 excellent
**Rating:** 6
**Confidence:** 3

**Summary:**

This paper proposes a new method for domain generalization called In-Context Risk Minimization (ICRM), a method which generalizes to a new domain given a sequence of unlabeled examples from that domain. The authors motivate ICRM by referring to the LLM literature, in particular their emergent ability to perform well OOD. They prove that ICRM can "zoom-in" on good/optimal risk minimizers given a sample/infinite data. Empirically, they demonstrate that ICRM outperforms baselines on several DG datasets and perform supplemental experiments to ablate and interpret these results.

**Strengths:**

- I think this is an ambitious paper and connects two interesting areas within OOD-type research - there are definitely interesting insights gained from the new setup and the idea of receiving a sequence of new-domain examples
- empirical results are strong, demonstrate a good use of unlabelled data
- experiments are pretty thoroughly done, I appreciate the supplementary studies in Table 3, Fig 2, and Fig 5
- the new method is described clearly and the theoretical results seem useful: in particular Theorem 3 and its extension seem important in terms of plugging this work into the iVAE work. I also like the specification of the limitations in terms of Voronoi cells of the training environments as a concept

**Weaknesses:**

- I find myself somewhat confused by the analogy between LLMs and ICRM - the paper makes it seem as those these should map 1:1 but I can't quite make it clear to myself, perhaps the authors can clarify. It's not clear what the sequence of Xs that arrive correspond to in LLMs, since they are listed as being selected at random at training time: this means they can't be language tokens, and if they are unrelated sequences I don't see how they correspond with the notion of context laid out in the "gravitational fields poem" example (these are also correlated - context sequences in the LLM space are usually not thought of as random wrt what follows them). I think ICRM is interesting nonetheless but I have trouble getting the intuition straight because of this
- I think that the relationship to various types of distribution shift should be highlighted a little bit more here: for instance, it seems like ICRM will be vulnerable to shifts in P(Y | X) across environments, since it only receives unlabelled examples from the new domain at test time. I think this is true of many DG methods, and I think this is what the "Voronoi cells" specification in Thm 3 is constraining, which I think is nice. However, I think this weakness is somewhat obscured by the discussion of context and the analogy to LLMs (e.g. gravitation fields poem example): in LLMs the "context" as it is generally considered is quite rich and can help with generalization to compound ie Y | X shifts, but ICRM as I understand it does not have that capability.
- in Sec 5, I don't find the example particularly clarifying: I may be missing something but it seems unfair to give ICRM access to the \mu variables, and not give them to ERM. It's possible this is a reasonable comparison to make but I think it needs to be explained a bit more in that case.
- I don't understand how ICRM allows models to take advantage of "natural order", since the examples are drawn randomly at training time, correct?
- I find it a little puzzling that the performance of ICRM decreases with added context in Camelyon17 - is there an explanation for this?
- It's strange to me that ICRM would outperform other methods even at 0 context: isn't the advantage of the method the ability to use context?


Smaller questions:
- Intro: Minor point but important I think - the Angwin et al paper (Propublica COMPAS investigation) is not an example of an investigation of the impact of out of distribution model failures, but rather the impact of unbalanced model error rates. There are many studies of societal impacts of OOD failures and I think it would be better to cite one of those
- Thm 3: "there exists an ICL algorithm" - what is an ICL algorithm? I don't think this is defined as a mathematical object
- end of Sec 5: I think you mean that lung cancer is "deterministic" in the single smoker case, rather than it "invariably follows"

**Questions:**

- Some people would argue that LLMs don't actually perform well OOD - rather, they are trained on a large enough dataset that instances where they have to perform truly OOD are much rarer. This might be good to address in the motivation/intro
- In general, I find the two concrete examples in the paper more confusing that clarifying (gravitational fields poem, ICRM example in Sec 5): I think clarifying these would go a long way to my understanding of the method and paper

---

> ### Author Response · Authors · 2023-11-15
> **Response to the Reviewer NswY**
>
> We thank the reviewer for their thought provoking review and encouraging words of appreciation for our work. We have incorporated the changes which can be found in magenta in the main body and the Appendix.
>
> ---
>
>
> > On the analogy between LLMs and ICRM
>
> We agree that there is not an exact one-to-one correspondence between all aspects of an LLM and ICRM. However, a few important aspects bear a strong parallel in our implementation. Firstly, both LLM and ICRM condition on the entire sequence of experience so far. Secondly, both typically exploit attention mechanisms to select the examples to pay attention to. However, there are differences, as rightly pointed out in the review. For us, the examples in a context are unlabeled inputs sampled at random from the environment, and thus the order of the examples does not matter. This is a facet specific to datasets we consider and, hence is different from standard language modeling. However, the proposal can handle datasets with some natural ordering based on time (e.g., video) and space (e.g. text). In such a scenario, we preserve the natural ordering instead of shuffling it.
>
>
> ---
>
>
>
> > On the nature of distribution shifts
>
> We agree that ICRM cannot tackle arbitrary shifts in $P(Y|X)$, and the Voronoi cells are precisely meant to constrain that. However, there is an interesting side to it. The constraint imposed by the Voronoi cell is more relaxed than the standard covariate shift assumption where $P(Y|X)$ does not vary. This is afforded to ICRM by the fact that it operates on an extended feature space that takes both current input $X$ and distribution $p(X)$ from the test environment as input, while standard learners (based on ERM, IRM) operate solely on $X$.
>
> ---
>
>
> > On the fairness of comparison between ICRM and ERM in Section 5
>
> This example is meant to bring to light how a key difference between the nature of ICRM and ERM translates to an important difference in the learned model. ERM operates on the current query $x$, and ICRM in its standard form operates on $x$ and $p(x)$. In this example, we make an additional assumption that ICRM operates on $x$ and $\mu(p(x))$, which we have stated. Under this assumption, we show that ICRM learns the right invariances. It is a pedagogical example to show that “if one could use distributional features as additional inputs, one could reveal the invariance of interest”. Further, our experiments are empirical support for ICRM having such capability for distributional feature discovery.
>
> ---
>
>
> > On the issue of ICRM exploiting “natural order”.
>
> This point is absolutely correct. In the current setup, we randomly sequence data and do not exploit natural order. We were alluding to possible future work where, if the data had some natural order, namely, say a video where you classify each frame, our ICRM  proposal extends without change and exploits natural order. In sum, our remark about “natural order” is made as an attempt to motivate future data collectors to retain this precious time-metadata, so methods such as ICRM can be deployed in full force.
>
> ---
>
>
> > On the drop in Camelyon-17.
>
> In the Camelyon dataset, standard ERM often encounters a failure mode where the model exploits shortcut features such as luminosity. Consider a model that only takes the current query as input but has an effective featurizer that eliminates luminosity. Such a model can perform well on Camelyon-17. We posit that ICRM learns a similar featurizer by observing context during training. Consequently, the ICRM model has to learn to ignore additional queries provided during test time.  In some cases, these additional queries may have acted as distractors, causing a 1.2 percent drop in performance.
>
> ---

---

> > ### Author Response · Authors · 2023-11-15
> > **Response to the Reviewer NswY (Continued)**
> >
> > > On the zero-shot gains
> >
> > The most natural source of gain for our method is to utilize additional information in the context. However, it can offer zero-shot benefits. We'll elaborate on this with two related descriptions.
> >
> > - Observe that while ICRM models can predict without context at test time, these have always been trained to observe contexts during training. Therefore, when the ICRM model is trained, it has the opportunity to learn to extract features from the current image while paying attention to other images in the context. By attending to images that the model finds useful for prediction, ICRM can end up learning “in weights” a higher-quality featurizer. That is, ICRM models may “depend” on context to learn environmental spurious features while burning “in-weights” the invariant features across environments. Our current theoretical analysis is mainly geared towards explaining the benefit of contextual information. We believe it is a fruitful future work to develop a theory to explain the selective augmentation perspective.
> > - The example in equation (7) and its extensions in Appendix A.6 show that the extended feature space of ICRM can lead to better out-of-distribution performance even when it does not have any contextual information at test time. Since the model is trained over an extended feature space, it can learn the right invariances for inputs in the original feature space.
> >
> > ---
> >
> >
> > > On Propublica and COMPAS
> >
> > We are happy to cite other studies regarding societal impacts of OOD failures
> > - Angwin, Julia, et al. "Machine bias." Ethics of data and analytics. Auerbach Publications, 2022. 254-264.
> > - Barocas, Hardt, Narayanan: Fairness and Machine Learning: Limitations and Opportunities (MIT Press, 2023)
> > - Hazirbas, Caner, et al. "Towards measuring fairness in ai: the casual conversations dataset." IEEE Transactions on Biometrics, Behavior, and Identity Science 4.3 (2021): 324-332.
> >
> > ---
> >
> >
> > > Definition of the ICL algorithm
> >
> > An ICL algorithm receives the data from multiple environments as input and outputs a predictor that takes the current query and the context prepended to it as input. We have clarified it in the manuscript.
> >
> > ---
> >
> >
> > > On lung cancer, deterministic vs invariably follows
> >
> > That’s correct, for one given smoker, the outcome of disease given its entire vector of characteristics is deterministic.
> >
> > ---
> >
> >
> > > On OOD claims for LLMs
> >
> > In the part, where we referred to OOD generalization capabilities of  LLMs, we have prefixed the claims with “may” as it is a hypothesis that is still being debated.
> >
> > ---
> >
> >
> > > On clarification on gravitation fields example
> >
> > In this example, we only meant to show that the model can adapt its output based on the context (e.g., a teenager vs a physics graduate). In standard examples of capital prediction tasks of in-context learning shown in the introduction, the model also adapts its behavior based on contextual information. We think the confusion stems from the fact that in-context learning is often associated with (input, label) pair demonstrations. However, a model can use other sources of information in the context and not just the labels. As shown in [1]. labels do not play a significant role in several in-context learning tasks.
> >
> > [1] Work, What Makes In-Context Learning. "Rethinking the Role of Demonstrations: What Makes In-Context Learning Work?."

---

> > > ### Comment · Reviewer_NswY · 2023-11-20
> > > **Response**
> > >
> > > Thanks for this rebuttal - mostly my opinion on the paper has remained stable although I appreciate some clarifications. Some thoughts:
> > >
> > > - I think the claim of the paper seems to imply that the analogy is pretty tight between ICRM to LLMs - I'd recommend backing off some of this language if LLMs are mostly inspiration. It's actually quite different to be operating on p(x) as an input than it is to do sequence prediction. I think this is where my confusion comes from - I think it would be much better to have examples which are *not* sequence prediction but nonetheless show the desired behavior
> > > - On that note, I think describing this model as "operating on p(x) as an input" is nice language and would find that clarifying in the paper
> > > - citations for societal impacts: I haven't read the top two, they seem fairly general. The bottom seems closer to what we're talking about here - not sure if you're interested in distinguishing between a subgroup failure and an OOD failure.

---

> > > > ### Author Response · Authors · 2023-11-20
> > > > **Response to the latest comment by the Reviewer NswY**
> > > >
> > > > We thank the reviewer for their feedback and continued support!
> > > >
> > > > We want to clarify that the model is trained via minimizing a loss that makes predictions at different sequence lengths from $1$ to $t$. Similarly, at inference, the model makes predictions at different sequence lengths. Hence, we believe it may not be appropriate to claim that the model operates on p(x).
> > > >
> > > > We will be happy to further clarify our position on LLMs and our work in the paper.
> > > >
> > > > In LLMs, models operate on sequences of the form
> > > >
> > > > $\boldsymbol{z_1}$ $z_2$
> > > >
> > > > $z_1$ $\boldsymbol{z_2}$ $z_3$
> > > >
> > > > $z_1$ $z_2$ $\boldsymbol{z_3}$ $z_4$
> > > >
> > > > $z_1$ $z_2$ $z_3$ $\boldsymbol{z_4}$ $z_5$
> > > >
> > > > where the model predicts the last token in the above sequence conditioned on all tokens appearing before it.
> > > >
> > > > Similarly in ICRM, we operate on sequences of the form
> > > >
> > > > $\boldsymbol{x_1}$ $y_1$
> > > >
> > > > $x_1$ $\boldsymbol{x_2}$ $y_2$
> > > >
> > > > $x_1$ $x_2$ $\boldsymbol{x_3}$ $y_3$
> > > >
> > > > $x_1$ $x_2$ $x_3$ $\boldsymbol{x_4}$ $y_4$
> > > >
> > > > In both cases, using transformers, we predict the last term in the sequence conditioned on all that follows before it.

---

### Official Review · Reviewer_1GJP · 2023-11-01

**Soundness:** 2 fair
**Presentation:** 3 good
**Contribution:** 2 fair
**Rating:** 6
**Confidence:** 2

**Summary:**

This paper introduces a new method for domain generalization that utilizes data previously seen in the test set as contextual information to improve generalization accuracy.

**Strengths:**

The writing in the article flows smoothly, and the experiments seem to yield very promising results.

**Weaknesses:**

The combination of domain generalization and in-context learning is an ambitious idea. However, the theoretical and experimental discussions in this paper are not sufficient.

I find that the motivation for integrating these two concepts does not fully convince me. It appears that this paper only utilizes information from observed data samples x, which I believe is not entirely consistent with the current concept of in-context learning in LLM because a sequence formed solely from observed training data x may not be sufficient as a demonstration to the model. Authors could consider providing sequences of x, y pairs as context, rather than just a sequence of x.

While this paper seems to get promising results, its practical applicability may be limited because it does not account for the possibility of the testing set data samples being a mixture of unknown domains. To address this issue, it might be worth exploring the identification of domain-specific information or introducing additional information as prompts (context).

In fact, the method proposed in this paper is quite similar to the example presented in Figure 1.b, with the key difference being that this paper's method does not involve averaging.

**Questions:**

* The method proposed in this paper was inspired by some ideas in in-context learning, but its relationship with in-context learning in LLM is not significant. The description in the third paragraph of the introduction can easily lead to misunderstandings.

* A concern regarding Theorem 1: Why does the inequality hold? Even though we assume that data can be independently sampled from x, it doesn't necessarily imply that the distribution of the sequence formed by the samples is independent of the variables. Therefore, The assumption in Theorem 1 seems not correct. Are there any relevant references that can confirm the existence of this independence?

* Does function h have different parameters for different environments? Theorem 1 indicates that h has different parameters depending on the environment. If this is the case, which set of parameters should be used during testing?

* The current motivation of this article lies in the idea that applying context information in domain generalization can get better generalization results. Furthermore, this paper argues that previous methods did not consider this aspect. However, there is an issue in assuming that testing data comes from the same domain (or we know the domain partition of test data) when discussing the generalization problem. If it samples from different domains, this approach may potentially mislead the model to produce incorrect results, as the function h learned during training somehow combines information from c and x to make decisions.

* Although the experiments have yielded many promising results, they are only compared to ERM and two baselines from three years ago. The baseline's performance is even lower than ERM, although the author initially acknowledges that most methods may not necessarily outperform ERM. However, the choice of baselines appears insufficiently comprehensive.

---

> ### Author Response · Authors · 2023-11-15
> **Response to the Reviewer 1GJP**
>
> We thank the reviewer for their critical feedback that helps us improve the work and also for appreciating the ambitious idea that we undertake in this work. We have highlighted the changes in magenta in both the main body and the Appendix of the revised manuscript.
>
> ---
>
> > On integration of the two concepts of domain generalization and in-context learning:
>
> We will clarify that, in order to benefit from in-context learning in domain generalization, the context itself must be a sequence of unlabeled inputs. The first and most popular conception of In-context learning [1]  involves providing the model with contextual information, typically a few sample $(x, y)$ pairs that represent a specific “task”. In contrast, we introduce an alternative perspective on in-context learning, where unlabeled $x$ inputs act as the contextual backdrop for a task, also known as an 'environment.'  Below, we emphasize the significance of this approach along two key dimensions.
>
> - **Why unlabelled $x$’s:** In domain generalization, at test-time, the learner only has access to the unlabeled x’s from the test environment and not both $(x,y)$ pairs. We embrace this challenge and develop a proposal inspired by in-context learning that shows we can continue benefiting from information in the unlabeled $x$’s. Our theory qualifies the conditions under which the information from unlabelled $x$’s only can help improve upon standard ERM in both in-distribution and out-of-distribution settings.
>
> - **Example of when only $x$’s can provide useful context:** Consider the following example – there are two types of users, one who has a more cursive handwriting and the other type uses straight edges.  At test time, the classifier may be faced with samples from cursive users. By observing the images ($x$’s) only of the samples drawn by the user without the label, the model can quickly realize that it is dealing with a cursive type user. These intuitions are made formal by our theory. Our method capitalizes on these intuitions to show the empirical benefits.
>
> Further, our method can be easily adapted to deal with $(x,y)$ pairs as input. We refer to this approach as Supervised ICRM. As shown in the table below and Section D.3 and as anticipated, Supervised ICRM outperforms ICRM but is not suitable for domain generalization settings where data labels are unavailable at inference.
>
> Dataset |  |  | Avg.* |  |  |  |  | WG.*  |  |  |
> |----------|----------|----------|----------|----------|----------|----------|----------|----------|----------|----------|
> **FEMNIST** | 0 | 25 | 50 | 75 | 100 | 0 | 25 | 50 | 75 | 100 |
> ICRM       |  78.7  | 87.2| 87.4 | 87.5|  87.8|  59.8| 69.3 | 70.6 | 70.6| 70.6
> Supervised ICRM | **79.0** | **87.8** | **87.7** | **88.2** | **87.9** |**61.2** | **72.2** | **73.5** | **74.5** | **74.9** |
> | | | | | | | | | |
> **Rotated MNIST** | 0 | 25 | 50 | 75 | 100 | 0 | 25 | 50 | 75 | 100 |
> ICRM       |  **93.6** |   96.1|  96.2| 96.2 | 96.2|  **82.5**|  88.5|  88.5| 88.8| 88.8
> Supervised ICRM | 93.3 | **96.3** | **96.3** |**96.3** | **96.3** | 82.0 | **89.0** | **89.0** | **89.1** | **89.3** |
>
> To further the evidence that our proposal of using unlabeled $x$’s is powerful, we provide experiments on two more datasets Imagenet-R (consisting of renditions of Imagenet classes) and CIFAR10-C dataset. Please see the table below or refer to Section D.2.
>
> Dataset |  |  | Avg.* |  |  |  |  |  | WG.*  |  |  |
> |----------|----------|----------|----------|----------|----------|----------|----------|----------|----------|----------|----------|
> CIFAR10-C | 0 | 25 | 50 | 75 | 100 | | 0 | 25 | 50 | 75 | 100 |
> ARM   | 65.9 | 66.0 | 66.0 | 66.0 | 66.0  | | 39.3 | 39.3 | 39.4 | 39.3 | 39.4
> ERM   | 66.1 | 66.1 | 66.1 | 66.1 | 66.1 | | 39.8 | 39.8  | 39.8  | 39.8  | 39.8
> ICRM  | **70.6** | **71.9** |  **71.9** | **71.9** | **71.9** | | **54.6** | **56.0** | **55.8** | **55.8** | **55.9**
> |  |  |  |  |  |  |  |  |  |  |
> ImageNet R | 0 | 25 | 50 | 75 | 100 | | 0 | 25 | 50 | 75 | 100 |
> ARM   | 56.3 | 58.1 | 58.8 | **59.8** | 59.0  | | 47.4 | 45.3 | 47.2 | **49.8** | 47.4
> ERM   | **58.9** | 58.9 | 58.9 | 58.9 | 58.9 | |**48.0** | **48.0**  | **48.0**  | 48.0  | 48.0
> ICRM   | 57.4 | **59.7** |  **59.6** | 59.4 | **60.5** | | 45.4 | **48.0** | 47.2 | 46.9 | **50.6**
>
> *Here, “Avg.” and “WG.” respectively denote the average and the worst group accuracy across test environments.
>
> [1] Brown, Tom, et al. "Language models are few-shot learners." Advances in neural information processing systems 33 (2020): 1877-1901.

---

> ### Author Response · Authors · 2023-11-15
> **Response to the Reviewer 1GJP (Continued)**
>
> > On the assumption that testing data sequences are sampled from the same domain
>
> While we did mention this point in the previous draft, we acknowledge that it may not have been sufficiently emphasized. In the revised manuscript, we have reiterated this aspect multiple times. We agree that this is an assumption, if true, shows the success of the proposal. Additionally, we consider extending our approach to settings where this assumption does not hold as an exciting avenue for future research. Nevertheless, it's important to note that the current assumption is practical for many scenarios. In particular, all work in domain generalization (see DomainBed [1], WILDS [2]) follows the same fixed-test-domain protocol.
>
> [1] Gulrajani, Ishaan, and David Lopez-Paz. "In search of lost domain generalization." arXiv preprint arXiv:2007.01434 (2020).
>
> [2] Koh, Pang Wei, et al. "Wilds: A benchmark of in-the-wild distribution shifts." International Conference on Machine Learning. PMLR, 2021.
>
> ---
>
>
> > On the difference w.r.t averaging based methods
>
> Our experiments show that a long line of work starting from the seminal works on marginal transfer learning that first appeared in NeurIPS 2011, later improved by adaptive risk minimization (NeurIPS 2021) have adopted a suboptimal choice for capturing contextual information.  Our proposal simplifies such literature in light of recently developed transformer and attention architectures. In a nutshell, we show that state-of-the-art performance is obtained by presenting all of the data as a sequence directly to the model, which can later decide to average in various non-linear ways. Further, as discussed in the paper, the size of the representation $\phi$ would have to grow linearly with the size of the training data to ensure storing all of the information available in the data
>
> ---
>
>
> > On the relationship to in-context learning in LLMs
>
> Our whole journey and thought process in this paper was inspired by LLMs and their ability to tackle OOD tasks through in-context learning. Our approach is inspired by it and is not an exact emulation of the workings of a LLM. We have added additional clarifications on this in the paper.
>
> ---
>
> > On the concern regarding Theorem 1
>
> We are uncertain about which inequality is being referred to here."
> - If the reference is to $I(Y; E|X)>0$, then we emphasize that this simply states that X does not carry all the relevant information to predict Y. There is some extra information to be gained from knowing the environment one is operating in. For instance, if one is driving, the decision to speed or not should depend on the current image $X$ in front of the driver and the operating conditions (crowded city, weather, etc., that form $E$). This assumption is made in all marginal transfer learning papers.
>
> - Or perhaps the reference is to the inequality in equation (10). Perhaps your concern refers to the condition above equation 10 in the Appendix, which states $I(Y; C_t | X, E)=0$. Below, we show how to derive it. Equation 10 rests on the assumption that $X_i, Y_i$’s are independent conditional on $E$. This is a fairly standard data generation assumption in most domain generalization papers (e.g., invariant risk minimization, its precursors, and many works that follow up on it). The assumption just means that within an environment, $X_i, Y_i$ samples are independent of one another. Conditional on the user, if the first digit drawn by a user is 9, that does not impact the choice of the digit it draws next.
>
> Consider two samples $(X_1,Y_1,X_2,Y_2)$ from the environment $E=e$. From conditional independence of the pair it follows that
> $P(X_1=x_1, Y_1=y_1, X_2=x_2, Y_2=y_2 | E=e) = P(X_1=x_1, Y=y_1|E=e) P(X_2=x_2, Y_2=y_2 |E=e)$
>
> We now take the sum over $y_1$’s on both sides of the above equality to obtain.
> $\sum_{y_1} P(X_1=x_1, Y_1=y_1, X_2=x_2, Y_2=y_2 | E=e) = \sum_{y_1}P(X_1=x_1,Y=y_1|E=e) P(X_2=x_2,Y_2=y_2 |E=e)$
>
> The LHS simplifies to $P(X_1=x_1,X_2=x_2, Y_2=y_2|E=e)$ while the RHS simplifies to $P(X_1=x_1|E=e) P(X_2=x_2,Y_2=y_2|E=e)$. Thus we obtain
> $P(X_1=x_1,X_2=x_2, Y_2=y_2|E=e) = P(X_1=x_1|E=e) P(X_2=x_2, Y_2=y_2|E=e)$
>
> The RHS can be simplified further and we obtain
> $P(X_1=x_1,X_2=x_2, Y_2=y_2|E=e) =  P(X_1=x_1|E=e)P(X_2=x_2|E=e)P(Y_2=y_2|X_2=x_2,E=e)$
>
> From the above equality, we obtain that $Y_2$ is independent of $X_1$  conditional on $X_2, E$. Since $X_1$ alone forms the context, $Y_2$ is independent of the context conditional on $X_2, E$. Hence, in this example $I(Y_2;C| X_2,E)=0$. The above argument readily extends to longer sequences, and thus we get $I(Y; C_t | X, E)=0.$

---

> > ### Author Response · Authors · 2023-11-15
> > **Response to the Reviewer 1GJP (Continued)**
> >
> > > Regarding $h(.)$ and parameters for different environments
> >
> > $h(.)$ possesses different parameters for  different environments. Theorem 1 deals with in-distribution  evaluations. Hence, the parameter of the test environment for Theorem 1 is set to be one of the training environments. In Theorem 3, we deal with out-of-distribution environments. In this case, the parameters of the test environment are constrained to be in the Voronoi cell of the training environment. Note that Voronoi cells do not constrain the test parameters to be arbitrarily close ,and quite remarkably, the parameter vector can be quite far. In our experiments, since we conduct out-of-distribution evaluations, in some cases these parameters can be quite different from those seen at train time.
> >
> > ---
> >
> >
> > > On additional baselines
> >
> > We have incorporated six additional recent and popular baselines into our study. These baselines either extend the invariance-based paradigm (Fish, IB-IRM), leverage contextual information differently (BN-adapt, Bayesian BN-adapt), or apply classic regularizations (Mixup, IB-ERM). You can find the detailed comparisons in the table below and Section D.2.
> >
> > Dataset |  |  | Avg.* |  |  |  |  | WG.*  |  |  |
> > |----------|----------|----------|----------|----------|----------|----------|----------|----------|----------|----------|
> > **FEMNIST** | 0 | 25 | 50 | 75 | 100 | 0 | 25 | 50 | 75 | 100 |
> > BN Adapt  | 78.3 | 76.9 | 80.3 | 81.5 | 82.4 | 54.3 | 56.2 | 61.9 | 64.7 | 65.3
> > Bayesian BN Adapt  | 78.3 | 79.6 | 81.3 | 82.2 | 82.9 | 54.3 | 60.4 | 64.7 | 65.5 | 66.3 |
> > Fish | 74.2 | 74.2| 74.2| 74.2| 74.2| 47.3| 47.3|47.3 | 47.3 |47.3|
> > IB-ERM | 74.3 |74.3 |74.3 |74.3 |74.3 |53.0  |53.0  |53.0  | 53.0 |53.0 |
> > IB-IRM | 68.9 |68.9 |68.9 |68.9 |68.9 |46.5 | 46.5|46.5 | 46.5 |46.5
> > Mixup | 80.5 |80.5 |80.5 |80.5 |80.5 | 59.0 |59.0 |59.0 | 59.0| 59.0|
> > ICRM       |  **78.7**  | **87.2**| **87.4** | **87.5**|  **87.8**|  **59.8**| **69.3** | **70.6** | **70.6**| **70.6**
> > | | | | | | | | | |
> > **Rotated MNIST** | 0 | 25 | 50 | 75 | 100 | 0 | 25 | 50 | 75 | 100 |
> > BN Adapt  | **94.6** | 87.0 | 91.5 | 93.7 | 94.3 | 80.5 | 70.9 | 76.9 | 79.8 | 80.9
> > Bayesian BN Adapt  | **94.6** |  91.2 | 93.4 | 94.3 | 94.7 | 80.5 | 75.4 | 79.2 | 80.7 | 81.3
> > Fish | 93.5 | 93.5| 93.5|93.5 |93.5 | 76.1| 76.1| 76.1| 76.1 | 76.1 |
> > IB-ERM | 92.4 | 92.4| 92.4| 92.4| 92.4|77.7 |77.7 |77.7 |77.7| 77.7 |
> > IB-IRM | 84.8 |84.8 |84.8 |84.8 |84.8 | 65.1| 65.1| 65.1| 65.1|65.1|
> > Mixup | 93.2 | 93.2|93.2 |93.2 |93.2 |76.7 | 76.7 |76.7 |76.7 |76.7|
> > ICRM       |  93.6 |   **96.1**|  **96.2**| **96.2** | **96.2**|  **82.5**|  **88.5**|  **88.5**| **88.8**| **88.8**
> > | | | | | | | | | |
> > **WILDS Camelyon17** | 0 | 25 | 50 | 75 | 100 | 0 | 25 | 50 | 75 | 100
> > BN Adapt | 67.5 | 82.0 | 87.4 | 89.7 | 89.9 | - | - | - | - | - |
> > Bayesian BN Adapt  | 67.5 | 82.0 | 87.3 | 89.6 | 89.7 |- | - | - | - | - |
> > Fish | 53.8 |53.8 | 53.8| 53.8| 53.8| - | - | - | - | - |
> > IB-ERM | 50.3 | 50.3| 50.3| 50.3|50.3 |- | - | - | - | - |
> > IB-IRM | 51.1 | 51.1| 51.1| 51.1| 51.1|- | - | - | - | - |
> > Mixup | 36.2 |36.2  |36.2  |36.2  |36.2  |- | - | - | - | - |
> > ICRM |  **92.0** | **90.7**  | **90.8**  | **90.8** | **90.8** | - | - | - | - | - |
> > | | | | | | | | | |
> > **Tiny ImageNet-C** | 0 | 25 | 50 | 75 | 100 | 0 | 25 | 50 | 75 | 100
> > BN Adapt  | 31.7 | 1.7 | 1.7 | 1.9 | 2.1 | 9.4 | 1.3 | 1.4 | 1.6 | 1.7
> > Bayesian BN Adapt  | 31.7  | 2.2 | 2.1 | 2.3 | 2.4 | 9.4 | 1.6 | 1.6 | 1.8 | 1.8
> > Fish | 32.5 |32.5 |32.5 |32.5 |32.5 | 11.5 |11.5  | 11.5 |11.5  | 11.5
> > IB-ERM | 34.7 | 34.7| 34.7| 34.7| 34.7| 14.7|14.7 |14.7 |14.7| 14.7 |
> > IB-IRM | 35.2 |35.2 |35.2 | 35.2| 35.2| 16.0|16.0 |16.0 | 16.0 |16.0|
> > Mixup | 36.2 |36.2 | 36.2| 36.2| 36.2 |11.6 |11.6 |11.6 | 11.6| 11.6 |
> > ICRM       |  **38.3** | **39.2**  | **39.2**  | **39.2** | **39.2** | **18.8**  |  **19.2** | **19.5**  | **19.5** | **19.4** |

---

> > > ### Author Response · Authors · 2023-11-20
> > > **Gentle Reminder to the Reviewer 1GJP**
> > >
> > > We hope that you found our responses along with the additional experiments that introduce new baselines and datasets convincing.  Please let us know if you have any further questions that you want clarified. The rebuttal interaction period is drawing to a close, so we look forward to hearing from you soon. Thank you for your time and consideration.

---

> > > > ### Author Response · Authors · 2023-11-21
> > > > **Gentle reminder to the Reviewer 1GJP**
> > > >
> > > > With one day left to the closing of the interaction period, we wanted to take a last opportunity to send a gentle reminder. We hope our new baselines, new datasets, and additional clarifications have been sufficiently convincing. Please let us know if you have any questions. We thank you for your time and consideration.

---

### Official Review · Reviewer_7Bph · 2023-11-09

**Soundness:** 3 good
**Presentation:** 4 excellent
**Contribution:** 4 excellent
**Rating:** 8
**Confidence:** 3

**Summary:**

The authors introduce In-Context Risk Minimization (ICRM): an algorithm that leverages techniques from next-token predictors (i.e., "in-context learning") to learn a transformer-based prediction model that generalizes at test time to new unseen domains.  The authors present several theoretical results that provide intuition on the similarities and differences between their proposed approach and existing approaches in domain generalization. They also demonstrate the value of ICRM by benchmarking their algorithm against other domain generalization approaches on several datasets.

More abstractly, the authors argue that the success of their approach provides evidence that "context is environment": the intuitive idea considering examples from a similar environment as "context" can encourage the learned model to "zoom in" to control for environment-specific features, while also learning environment-agnostic invariant features.

**Strengths:**

This paper is extremely timely and would be of great interest to researchers on domain generalization and group robustness. I strongly recommend that this paper is accepted.

* The authors clearly motivate, present from first principles, and connect modern research in domain generalization and in-context learning.  Their notation and exposition in Sections 2 and 3 is thoughtful and clear.  Abstract concepts are made clear using examples, such as the self-driving car in Section 2 and sentence examples in Section 3.
* The authors' proposed algorithm (ICRM) is intuitive to implement, and described clearly.
* The authors' theoretical results in Section 3 (with the exception of the domain generalization result Theorem 3) are relatively intuitive, and also demonstrate the conditions under which ICRM is equivalent (Prop. 1) vs. beneficial (Theorem 1) over global ERM.
* The authors provide detailed information about their experimental design and instantiation of each algorithm being benchmarked so that their experiments are clearly reproducible.
* The authors' proposed method appears to result in significant performance gains over comparable domain generalization algorithms, which suggests its ability to have positive impact in practice.
* I appreciate the authors' efforts to thoughtfully connect more abstract interpretations of how "environment is context" to existing literature from causality that studies similar phenomena (at the end of Section 5).  I found the smoking example to be an especially clear analogy that helped me better understand the motivation behind the approach.

**Weaknesses:**

I am happy to consider raising my score if the authors address the below concerns.
1. **Choice of datasets for evaluation + including experiments on "harder" datasets with spurious correlations that vary across environments**.
  * My basic sense of the datasets that the authors chose to benchmark the value of their domain generalization method on is that there is arguably not *that* high variance in what semantic features are present across environments (e.g., simple rotations or corruptions are somewhat artificial shifts that are detached from reality). To really buy the author's claim that this approach is relevant for domain generalization, I would love to see experimental results on prominent spurious correlation benchmarks, such as those used in [1] (e.g., CivilComments, WaterBirds, CelebA).  I understand that several of these benchmarks only have 2 total test environments (so are perhaps unsuitable for domain generalization specifically), but can you attempt to include additional results on a spurious correlation benchmark (or justify why it's not appropriate)? I'm curious if on more difficult benchmarks, the model learned by ICRM may undesirably "zoom-in on toxic spurious correlations".
2. **Weakening claims about "what ICRM has learned"**.
  * Unless the authors attempt to peek under the hood (using post-hoc explanation methods or by designing controlled experiments) to probe at what features the ICRM models have "learned", I suggest that the authors make clear that some of their claims that their models learn "invariant" or "contextual features", are hypotheses/intuition (rather than assertions of the truth). ex: in 6.1, you state that the gains are "because ICRM training still benefits from contexts as to find contextual features that ERM ignores".  I think this is a bit too strong (what are the "contextual" invariant features for your experiments using real data, for example?).  Perhaps soften to "we hypothesize that this is because…"
3. **More discussion of how to use ICRM in practice**, i.e., the nuances of what test set data to present as belonging to the same "environment" (/context) vs. different environments.  IMO this approach is perfect for domain generalization settings where the "environment" is literally the institution in which the algorithm is being deployed (ex: I begin using it in a new hospital), but has way more challenges when we consider "environments" as naturally occurring subpopulations or time-based shifts.  Can you please include additional discussion within your paper about how one would decide what examples to group together as being in the same "environment" in practice? I also thought it was a really cool result that you don't need to observe labels in order to include examples in the same "context" that should be emphasized further in such a practical discussion.


[1] https://proceedings.mlr.press/v177/idrissi22a/idrissi22a.pdf

**Questions:**

In addition to the above 3 weaknesses which are most important to my score, I also had several comments, questions, and suggestions about places I got confused when reading the paper that can be made more clear.

* nit: Why did you choose the title "context is environment"? I wonder if "environment is context" is more appropriate, as you're aiming to solve domain generalization using in-context learning, and treating environment as context.  "Context is environment" to me implies that you are labeling "context" as environment to feed as input to domain generalization algorithms.
* nit, Intro: Is the expectation here that we have environment labels available during training time? – when you state that "simple empirical risk minimization" is a strong baseline, do you mean just training using ERM on the entire dataset in an environment-agnostic way? (As a reader, I would need to dig into the references you've cited here to clarify). EDIT:  I see in Section 2 that you mean a global ERM that "pools the data together".  Can you clarify this earlier in the intro?
* nit, Intro: Can you make more clear, perhaps using an example, what it means to "reveal" an "invariance"?  By an "invariance", do you mean some semantic feature that is predictive of the true label $y_i^e$ across all environments $e$?  By "reveal", do you mean that you discover that your in-context learner models actually "use" the invariant features to make predictions, or something else?
* Section 2: "the size of the representation would have to grow linearly with the size of the training data to describe aspects corresponding to a small group of examples, such as extreme value statistics". I had trouble interpreting this sentence.  What do you mean by "size" here – do you mean the dimensionality of each $\phi_i^e$?  Why are more dimensions necessary for the representations to encode relevant information?  Why do we care about "describing aspects corresponding to a small group of examples" – is it because we have fewer samples from some environments relative to others?  Is the problem that we must use the same function $\phi$ across all the environments?
* Section 5: "define $\delta_e$ to be a permutation of $\gamma_e$ that swaps its two components".  What do you mean by "two components" – do you mean swaps the parameters for $y = 0$ and $y = 1$?
* Section 5: I may be missing something, but I'm not sure if I understand the linear least-squares example given in Equation 7.  Is it fair in this case to compare ICRM to "global ERM", which in many cases will fail to learn to add environment-specific averages $\mu_e^i$? Does this phenom (that the true model can be learned) not hold for simple per-environment ERM too?  I am unsure of how illustrative this example (that ICRM tends to learn the true $\alpha$ vs. some $\alpha'$) is for the overarching claim that "ICRM learns invariant features".
* Section 5, last paragraph: I don't really understand the value of including the last sentence here ("when constraining the environment to only one smoker, the outcome of lung cancer disease invariably follows").  Are you trying to suggest that when we have fewer environments (a "smaller diameter"), that invariance is easier to achieve? Can you provide more practical context around this point?
* Section 6: I am confused by why in Figure 2, you show images that appear (from my eye) to be identical to the query, and calculate the attention score (e.g., between the two rotated 2s or Js).  Can you clarify exactly how the images on the right of the line are "augmented"?  Are you certain that these augmentations were significant enough?  Does it make sense for the model to see two almost identical images in the set of test examples?
* Section 6: nit: In Figure 2, can you write (in small font) what the true labels are for each image, rather than the integer indices? (or at least for the query images?)
* Section 7: "we enable learning machines to fully exploit data in natural order". I question if you should emphasize this point in your conclusion, as you didn't actually present any of the data "in natural order" (i.e., chronologically from its creation date) in any of your experiments.

---

> ### Author Response · Authors · 2023-11-15
> **Response to the Reviewer 7Bph**
>
> First of all, we would like to thank the reviewer for their diligent and insightful review and their very encouraging words. The changes in the manuscript can be found highlighted in magenta.
> We share our thoughts on the questions asked below.
>
> ---
>
> > On the choice of evaluation and including experiments on “harder” datasets with spurious correlations
>
> - In the literature, distribution shifts are often divided into two categories: diversity shift and correlation shift [1]. Our work primarily focuses on diversity shift, deferring the exploration of correlation shift to future research. Handling datasets like Waterbirds is challenging, as the typical evaluation assumes test data sampled i.i.d from all environments (subpopulations), making a 3:1 training-to-testing split nonstandard. Further, adapting ICRM to such a setting requires significant modifications to the training pipeline and additional analysis. Therefore, our focus is on domain generalization benchmarks, where all environments except one are used for training, aligning with our assumption of _only one environment during testing_.
> - To test the model in other hard settings, we have included the ImageNet R dataset [2], which consists of challenging renditions of Imagenet R classes, and the CIFAR10-C dataset. In both datasets, we observe consistent improvements offered by ICRM (see table below and also in the Appendix of the edited draft).
>
>
> Dataset |  |  | Avg.* |  |  |  |  |  | WG.*  |  |  |
> |----------|----------|----------|----------|----------|----------|----------|----------|----------|----------|----------|----------|
> CIFAR10-C | 0 | 25 | 50 | 75 | 100 | | 0 | 25 | 50 | 75 | 100 |
> ARM   | 65.9 | 66.0 | 66.0 | 66.0 | 66.0  | | 39.3 | 39.3 | 39.4 | 39.3 | 39.4
> ERM   | 66.1 | 66.1 | 66.1 | 66.1 | 66.1 | | 39.8 | 39.8  | 39.8  | 39.8  | 39.8
> ICRM  | **70.6** | **71.9** |  **71.9** | **71.9** | **71.9** | | **54.6** | **56.0** | **55.8** | **55.8** | **55.9**
> |  |  |  |  |  |  |  |  |  |  |
> ImageNet R | 0 | 25 | 50 | 75 | 100 | | 0 | 25 | 50 | 75 | 100 |
> ARM   | 56.3 | 58.1 | 58.8 | **59.8** | 59.0  | | 47.4 | 45.3 | 47.2 | **49.8** | 47.4
> ERM   | **58.9** | 58.9 | 58.9 | 58.9 | 58.9 | |**48.0** | **48.0**  | **48.0**  | 48.0  | 48.0
> ICRM   | 57.4 | **59.7** |  **59.6** | 59.4 | **60.5** | | 45.4 | **48.0** | 47.2 | 46.9 | **50.6**
>
> *Here “Avg.” and “WG.” respectively denote the average and the worst group accuracy across test environments. \
>
> [1] Ye, Nanyang, et al. "Ood-bench: Quantifying and understanding two dimensions of out-of-distribution generalization." Proceedings of the IEEE/CVF Conference on Computer Vision and Pattern Recognition. 2022.
>
> [2] Hendrycks, Dan, et al. "The many faces of robustness: A critical analysis of out-of-distribution generalization." Proceedings of the IEEE/CVF International Conference on Computer Vision. 2021.
>
> ---
>
> > On weakening claims about “What ICRM has learned”
>
> This is indeed a great point—we have weakened our exposition to explicitly emphasize that it is our hypothesis that ICRM exhibits a propensity to learn more invariant predictors. The post-hoc analysis of attention maps in Section 6.3 complements it empirically.
>
> To substantiate this further,  we extract embeddings from the penultimate layer of the model trained using our approach of the data pooled together from training environments. Then, we use these embeddings to train a linear classifier that predicts the corresponding environment index. The results (details in Section D.4) demonstrate our approach's success, achieving accuracy rates exceeding 75% on FEMNIST and 98% on Rotated MNIST. This shows that context can help us infer the environment and hence, reinforces the notion of _context is environment_.
>
> ---
>
> > More discussion on how to use ICRM in practice
>
> You make an excellent point on the type of distribution shifts our method is suited for. In datasets such as Waterbirds, the test domain is a mixture of hidden groups/domains and not separated into domains based on the groups. Extending our method to tackle distribution shifts in slowly-changing test environments and changes in subpopulation such as Waterbirds, CelebA type dataset is an exciting future work. We have added this discussion in the the revised manuscript
>
> ---
>
> > On “Context is Environment” vs. “Environment is Context''
>
> We consider the environment to be a more fundamental primitive. We believe that the data generation process in nature begins with sampling of an environment, followed by the generation of samples from this environment, forming the context. Due to this, it is the context that helps us infer the environment and hence context is environment.
>
> ---
>
> > On clarification of ERM
>
> In our paper, Empirical Risk Minimization (ERM) refers to the training of a single model by minimizing the error across the entire pooled dataset. We have incorporated the recommended clarification in the manuscript.
>
>
> ---

---

> ### Author Response · Authors · 2023-11-15
> **Response to the Reviewer 7Bph (Continued)**
>
> > Clarification on “reveal” an “invariance”
>
> By “invariance” we mean a feature representation inducing a classifier that performs well across environments. By “revealing” we mean that the learner can find and use such an invariance..
>
> ---
>
> > On the linear growth in the size of the representation
>
> In this particular case, we aimed to highlight the limitations of marginal transfer learning proposals that use non-linear averaging on the inputs. By size, we mean the dimensionality of the representation $\phi(x)$. Appendix Section A.7 presents a family of examples where such marginal transfer learning proposals fail. In particular, we give an example of function classes for which the size of $\phi(x)$ has to grow in the context length to perfectly match the function class. The overarching intuition here is that one would need an $\theta(n)$-dimensional $\phi(x)$ to ensure storing all of the information available in $n$ examples.
>
> ---
>
> > Clarifying the line on swapping two components
>
> Yes, we mean it swaps parameter tuples corresponding to $y=0$ and $y=1$.
>
> ---
>
> > On linear least squares example
>
> We assume in this example that ICRM directly operates on the features summarizing the means $\mu$. ERM in the standard form only operates on the current query and not on an extended feature space. This example illustrates how ICRM by operating on an extended feature space can discover invariances for features in the original input space. This is as opposed to a global ERM across environments, or any ERM on a separate environment.
>
> ---
>
> > On the example of smoking and invariance
>
> The smoking example attempts to illustrate the fact that, as we consider growing collections of examples (smokers), we need to mine more and more features to find an invariant map—for instance, consider the enormous amount of inputs that we would need to invariantly predict the outcome of disease for all smokers in the world with one single model. On the other extreme, if considering one smoker only, the invariant rule is constant and equal to the observed disease outcome. This illustrates a tension between how many environments one desires to be invariant across ($|E|$) and how many features are necessary to guarantee such invariance ($|C|$).
>
> ---
>
> > Clarification about images in the input sequences in Figure 2
>
> Figure 2 displays four random sequences, each containing images randomly sampled from _real data_ in a test environment. It's important to note that none of the images in these sequences were manually augmented. The key takeaway from this visualization is that ICRM effectively learns to attend to a select few samples in an input sequence. These samples either belong to the same class or exhibit similar features, despite potentially belonging to different classes.
>
> ---
>
> > True labels for each image in Figure 2
>
> We thank the reviewer for this suggestion. Using the label mapping [here](https://gist.github.com/aaronpolhamus/964a4411c0906315deb9f4a3723aac57), we have replaced the class indices with their true labels in Figure 2. Clearly, the third row shows that the model, when presented with a query image of a "bullet train", attends not only on other trains but also on a "school bus"---indicating a semantic understanding of similarity. Further, in the last row, since the associated image from class "volleyball” in the query, also features a volleyball player, the model learns to pay attention to other images with individuals, including those wearing academic gowns.
>
> ---
>
> > On the natural order of data point in the conclusion
>
> Existing datasets in domain generalization only carry environment-level separation, and the proposal is designed to handle the partial ordering of data dictated by this separation. However, if the data had more refined information (e.g., time-based ordering in video), the proposal can exploit that too. We have clarified this in the revised manuscript. In sum, our remark about “natural order” is made to motivate future data collectors to retain this precious time metadata, so methods such as ICRM can be deployed in full force.
>
> ---

---

> ### Comment · Reviewer_7Bph · 2023-11-19
> **Response to the rebuttal**
>
> Thank you for your prompt and detailed response to each of my questions and concerns!  Thank you also for updating your draft for clarity, and for also including the additional experimental results on ImageNet-R and CIFAR10-C.  The authors' responses have sufficiently addressed my concerns, and after reviewing the larger discussion at this time I still believe this paper should be accepted.

---

> > ### Author Response · Authors · 2023-11-20
> > **Response to the latest reply by Reviewer 7Bph**
> >
> > We are delighted that our responses have addressed the reviewer’s concerns and that they continue to support the acceptance of the paper. We thank them again for their time and consideration

---

### Author Response · Authors · 2023-11-15
**Summary of key revisions**

We thank the reviewers for dedicating their time and expertise to assess our paper. Their perceptive remarks and constructive feedback have been valuable in refining our work. In response, we have endeavored to address the primary concerns raised by the reviewers while also conducting additional experiments to augment the support for our claims. Below is a brief summary of the key revisions:

* Inclusion of 6 additional baselines, invariance-based approaches, models utilizing contextual information via batch statistics, and classic regularization techniques (Section D.2.1).
* Addition of two challenging datasets - ImageNet-R and CIFAR10-C (Section D.2.1).
* Additional ablation of the method with and without positional encodings and labeled context (Section D.3).
* Clarification of key aspects of our algorithm, such as zero-shot gains across benchmarks, unlabeled data driven ICL for domain generalization, extension to exploit data in _natural order_, and many more.

---

### Meta-Review · Area_Chair_UPg2 · 2023-12-06

**Metareview:**

This paper proposes In-Context Risk Minimization (ICRM), a new method for domain generalization that generalizes to a new domain using a sequence of unlabeled examples from that domain.

The paper is well-written and well-motivated, offering useful theoretical results and promising experimental findings, as agreed upon by most reviewers.

However, reviewers still express concerns regarding the weak connection between in-context learning in LLMs and domain generalization, the need for citations on societal impacts, and the comparison of zero-shot gains vs ICRM.

Nevertheless, given its novelty and promising results, I recommend accepting this paper with a poster rating.

**Justification For Why Not Higher Score:**

As mentioned above, reviewers continue to express concerns regarding the weak connection between in-context learning in LLMs and domain generalization, the need for citations addressing societal impacts, and the comparison of zero-shot gains to ICRM. Until these points are fully addressed, I believe that acceptance with a poster rating is appropriate

**Justification For Why Not Lower Score:**

The paper demonstrates a clear impact in the field, receiving overall positive ratings from all reviewers. Therefore, acceptance as a poster is an appropriate rating

---

### Decision · Program_Chairs · 2024-01-16

Accept (poster)